# A μ-opioid receptor superagonist analgesic with minimal adverse effects

Juan L. Gomez[1], Emilya N. Ventriglia[1], Zachary J. Frangos[1], Agnieszka Sulima[2], Michael J. Robertson[3,4], Michael D. Sacco[3,4], Reece C. Budinich[1], Ilinca M. Giosan[5], Tongzhen Xie[5], Oscar Solis[1], Anna E. Tischer[1], Jennifer M. Bossert[6], Kiera E. Caldwell[6], Hannah Bonbrest[6], Amelie Essmann[7,8], Zelai M. Garçon-Poca[7,8], Shinbe Choi[9], Michael R. Noya[9], Feonil Limiac[9], Ali Arce[9], Grant C. Glatfelter[10], Margaret Robinson[11], Li Chen[11], Angelina A. Mullarkey[3], Dain R. Brademan[3], Garrett Enten[12], William Dunne[1], César Quiroz[12], Ingrid Schoenborn[1], Chae Bin Lee[13,14], Rana Rais[13,14], Daniel P. Holt[15], Robert F. Dannals[15], Lei Shi[11], Ruth Hüttenhain[3], Sergi Ferré[12], Eugene Kiyatkin[9], Jordi Bonaventura[7,8], Yavin Shaham[6], Venetia Zachariou[5], Michael H. Baumann[10], Georgios Skiniotis[3,4,16,17 ✉], Kenner C. Rice[2 ✉] & Michael Michaelides[1 ✉]

Developing safe and effective pain medications is an ongoing challenge for human health. Agonists for the μ-opioid receptor (MOR) are essential pain medications, but their high intrinsic efficacy also induces adverse side effects, including respiratory depression, constipation, tolerance, dependence, withdrawal and addiction[1-7]. Strategies to limit adverse effects traditionally include developing MOR agonists that have low intrinsic efficacy or that preferentially activate G-protein signalling over β-arrestin signalling[8]. Here we identify a novel MOR agonist with supramaximal intrinsic efficacy and a unique pharmacological profile that produced effective analgesia in rodents with minimal adverse effects. N-desethyl-fluornitrazene (DFNZ) was derived from a class of synthetic benzimidazole opioids called nitazenes. DFNZ has impaired brain penetrance, a unique spatiotemporal MOR cellular signalling profile, and diminished efficacy at the MOR–galanin 1 receptor (GAL1) heteromer. DFNZ does not induce respiratory depression, tolerance or MOR downregulation after repeated exposure. Compared with other MOR agonists, DFNZ has limited effects on dopamine neurotransmission in nucleus accumbens and weaker reinforcing effects in the drug self-administration procedure. These results provide novel insights about MOR and nitazene pharmacology, have important implications for pain and addiction treatment, and challenge the prevailing dogma that high-efficacy MOR agonists cannot constitute safe and effective therapeutic agents.

The opioid crisis, initially sparked by the overprescription of opioid analgesics and exacerbated by the rise of potent synthetic opioids such as fentanyl, is a public health crisis that has led to many deaths over recent decades[6]. Recently, a class of synthetic opioids—the benzimidazole opioids (nitazenes)—has entered the recreational drug supply[9]. Nitazenes comprise a class of selective MOR agonists with high potency and efficacy[10]. Etonitazene, the most potent compound of this class, and related analogues were synthesized in the 1950s as potential analgesics, but clinical development was abandoned because of their extreme potency and overdose risk. Several nitazene analogues have been associated with human overdose fatalities, prompting their placement into Schedule I by the US Drug Enforcement Administration. Nitazenes show complex structure–activity relationships, with antinociceptive potency ranging from levels comparable to morphine to 1,000-fold higher[10], yet their pharmacological mechanisms are not well understood. Here we report the identification

[1]Biobehavioral Imaging and Molecular Neuropsychopharmacology Section, National Institute on Drug Abuse Intramural Research Program, Baltimore, MD, USA. [2]Drug Design and Synthesis Section, Molecular Targets and Medication Discovery Branch, National Institute on Drug Abuse Intramural Research Program, Baltimore, MD, USA. [3]Department of Molecular and Cellular Physiology, Stanford University School of Medicine, Stanford, CA, USA. [4]Department of Structural Biology, Stanford University School of Medicine, Stanford, CA, USA. [5]Department of Pharmacology, Physiology and Biophysics, Boston University Chobanian and Avedisian School of Medicine, Boston, MA, USA. [6]Neurobiology of Relapse Section, National Institute on Drug Abuse Intramural Research Program, Baltimore, MD, USA. [7]Departament de Patologia i Terapèutica Experimental, Institut de Neurociències, Universitat de Barcelona, L'Hospitalet de Llobregat, Spain. [8]Neuropharmacology and Pain Group, Neuroscience Program, Bellvitge Institute for Biomedical Research (IDIBELL), L'Hospitalet de Llobregat, Spain. [9]Behavioral Neuroscience Research Branch, National Institute on Drug Abuse Intramural Research Program, Baltimore, MD, USA. [10]Designer Drug Research Unit, National Institute on Drug Abuse Intramural Research Program, Baltimore, MD, USA. [11]Computational Chemistry and Molecular Biophysics Section, National Institute on Drug Abuse Intramural Research Program, Baltimore, MD, USA. [12]Integrative Neurobiology Section, National Institute on Drug Abuse Intramural Research Program, Baltimore, MD, USA. [13]Johns Hopkins Drug Discovery, Johns Hopkins School of Medicine, Baltimore, MD, USA. [14]Department of Neurology, Johns Hopkins School of Medicine, Baltimore, MD, USA. [15]Department of Radiology, Johns Hopkins School of Medicine, Baltimore, MD, USA. [16]Department of Structural Biology, St Jude Children's Research Hospital, Memphis, TN, USA. [17]Center of Excellence for Structural Cell Biology, St Jude Children's Research Hospital, Memphis, TN, USA. ✉e-mail: georgios.skiniotis@stjude.org; kennerr@nida.nih.gov; mike.michaelides@nih.gov

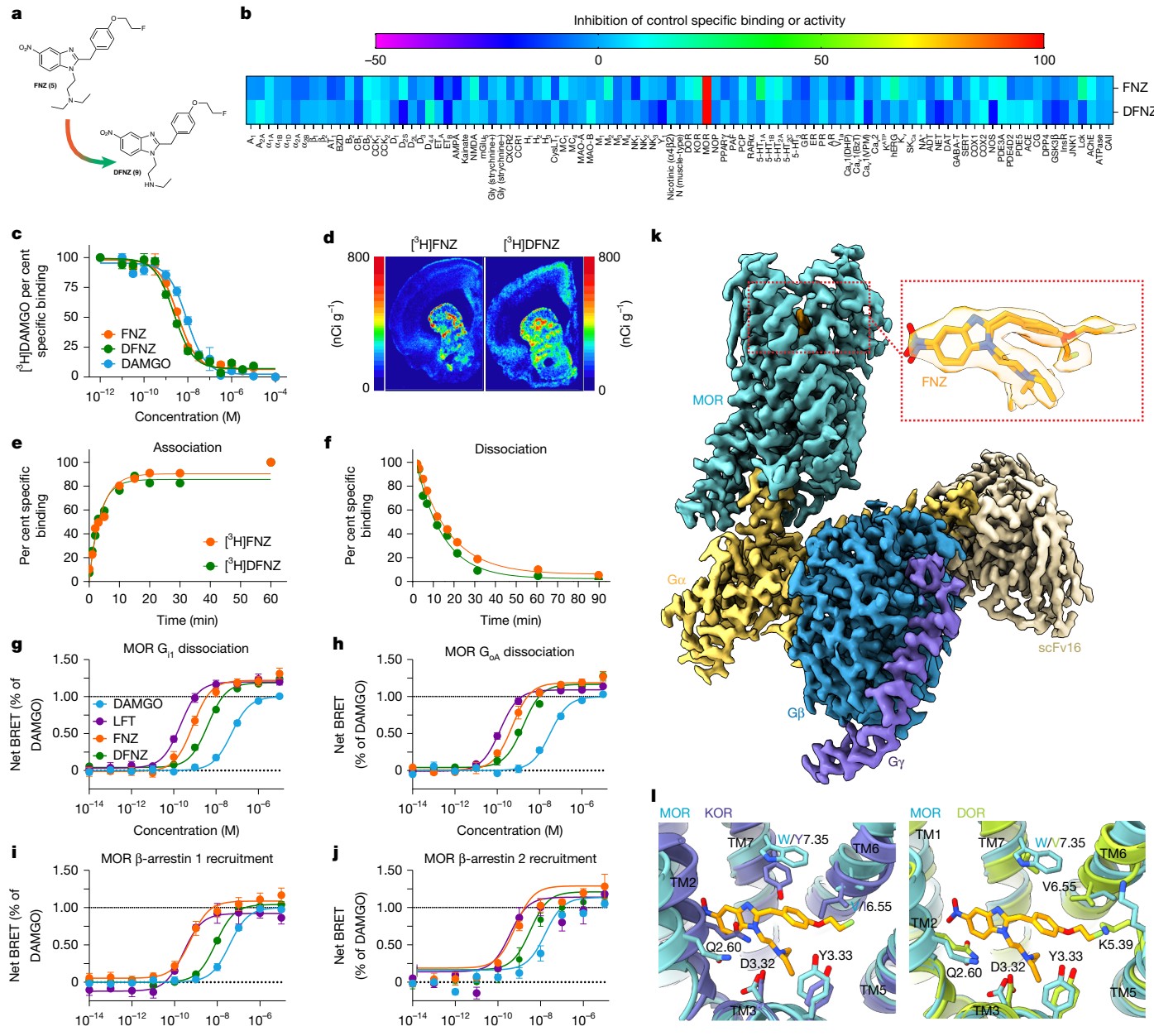

**Fig. 1 | FNZ and DFNZ are selective and potent MOR superagonists.**
**a**, Structures of FNZ and DFNZ. **b**, Competitive binding screens of 100 nM FNZ ($n$ = 2 independent experiments) and DFNZ ($n$ = 2 independent experiments). **c**, Competitive binding of FNZ and DFNZ against [$^3$H]DAMGO in rat brain membranes (three independent experiments). **d**, [$^3$H]FNZ and [$^3$H]DFNZ autoradiography in rat brain. **e,f**, Association and dissociation kinetic binding of [$^3$H]FNZ and [$^3$H]DFNZ to human MOR (two independent experiments). **g–j**, BRET assays showing human MOR G-protein activation and $G_{i1}$ (**g**), $G_{oA}$ (**h**), β-arrestin 1 (**i**) and β-arrestin 2 (**j**) recruitment with DAMGO, FNZ, DFNZ and lofentanil (LFT) (three independent experiments). **k**, Cryo-EM structure of FNZ bound to mouse MOR–$G_i$ protein complex at 2.3 Å resolution. **l**, Overlays of FNZ-bound MOR structure showing residues in Ballesteros–Weinstein notation

of a novel nitazene as a potential therapeutic agent for pain and opioid addiction.

## FNZ and DFNZ are selective MOR superagonists

Etonitazene has 1,000-fold greater antinociceptive potency than morphine[10], rendering it unsuitable for clinical use. However, modifications to its alkoxy chain length can alter MOR potency[11,12].

with active-state structures of κ- and δ-opioid receptors (KOR and DOR). For KOR, the W7.35Y and V6.55I substitutions together with a closer positioning of TM6 produced several clashes with the phenyl and $o$-fluoroethyl moieties. A shift in TM2 of KOR brings both the backbone and side chain of Q2.60 overlapping with the nitro-benzimidazole of FNZ. DOR has fewer clashes that all result from minor conformational differences. TM6 is pushed closer to the binding pocket, bringing V6.55 inward towards the $o$-fluoroethyl, and K5.39 and Q2.60 occupy different rotamers or positions that may result in additional mild clashes. Some of these differences for DOR (and to a lesser extent KOR) might be overcome with conformational rearrangements, albeit with an associated energetic penalty to compound binding. Data are shown as mean ± s.e.m.

We explored whether fluorine substitution at the end of the ethoxy chain would reduce potency and afford radiolabelling with $^{18}$F for positron emission tomography (PET) studies. We synthesized fluor-nitrazene (FNZ) (Fig. 1a and Supplementary Fig. 1) and confirmed its high MOR selectivity (Fig. 1b and Extended Data Fig. 1). FNZ had high affinity for MOR (inhibition constant ($K_i$) = 1.36 ± 0.11 nM (Fig. 1c,d)), relatively slow MOR binding kinetics[13] (on rate ($k_{on}$) = 6.35 × 10$^8$ M$^{-1}$ min$^{-1}$; off rate ($k_{off}$) = 0.062 min$^{-1}$; $T_{1/2}$ = 11.05 min (Fig. 1e,f)),

and activated MOR with high potency and supramaximal efficacy (expressed as percentage of DAMGO activity): $G_{i1}$ (half-maximal effective concentration ($EC_{50}$) = 0.79 ± 0.15 nM, maximum effect ($E_{max}$) = 122.0 ± 3.0%), $G_{oA}$ ($EC_{50}$ = 0.51 ± 0.09 nM, $E_{max}$ = 118.5 ± 2.6%), β-arrestin 1 ($EC_{50}$ = 0.57 ± 0.16 nM, $E_{max}$ = 108.8 ± 3.6%), β-arrestin 2 ($EC_{50}$ = 0.54 ± 0.15 nM, $E_{max}$ = 116.3 ± 3.9%) (Fig. 1g–j and Extended Data Fig. 1).

FNZ was rapidly metabolized in mouse liver microsomes, becoming undetectable within 30 min post incubation (Extended Data Fig. 2). Metabolite identification studies confirmed its primary metabolite as $N$-desethyl-FNZ (DFNZ) (Fig. 1a), which accounted for more than 50% of FNZ metabolism, and other metabolites each constituted less than 10% FNZ metabolism (Extended Data Fig. 2). We synthesized DFNZ (Supplementary Fig. 2) which retained the MOR selectivity (Fig. 1b and Extended Data Fig. 1), affinity ($K_i$ = 1.00 ± 0.06 nM) (Fig. 1c,d) and MOR binding kinetics ($k_{on}$ = 7.18 × 10$^8$ M$^{-1}$ min$^{-1}$; $k_{off}$ = 0.075 min$^{-1}$; $T^{1/2}$ = 9.22 min (Fig. 1e,f)) of FNZ. DFNZ activated MOR with high potency and supramaximal efficacy for $G_{i1}$ dissociation ($EC_{50}$ = 4.07 ± 0.43 nM, $E_{max}$ = 118.7 ± 1.6%), $G_{oA}$ dissociation ($EC_{50}$ = 1.66 ± 0.29 nM, $E_{max}$ = 116.5 ± 2.6%), β-arrestin 1 recruitment ($EC_{50}$ = 8.49 ± 0.89 nM, $E_{max}$ = 104.2 ± 1.6%) and β-arrestin 2 recruitment ($EC_{50}$ = 2.78 ± 0.78 nM, $E_{max}$ = 108.2 ± 3.9%), but showed divergent G-protein versus β-arrestin preference compared with FNZ (Fig. 1g–j and Extended Data Fig. 1) and was moderately stable in mouse liver microsomes (Extended Data Fig. 2).

To gain structural insights into the determinants of MOR activation and signalling by nitazenes, we obtained the cryo-electron microscopy (cryo-EM) structure of FNZ bound to the MOR–$G_i$ protein complex at 2.3 Å resolution (Fig. 1k and Extended Data Fig. 3). FNZ was well resolved in the orthosteric binding site, with its tertiary amine forming a salt bridge with the carboxyl side chain D3.32 (Ballesteros–Weinstein numbering[14]) (Fig. 1l), and the fluoroethyl group extending primarily towards transmembrane helix 6 (TM6). Both the fluoroethyl group and the diethyl amine may occupy more than one conformation, as suggested by the presence of adjacent map features near these groups and consistent with the flexible nature of these moieties (Extended Data Fig. 3). The benzimidazole ring is sandwiched between I7.39 and Q2.60, with its nitro group packing against Y1.39 and potentially forming a hydrogen bond with the side chain OH (Extended Data Fig. 3). Overlaying the FNZ-bound MOR structure with the active-state structures of κ- and δ-opioid receptors revealed clashes arising from sequence and conformational differences, providing insight into the basis of selectivity of FNZ for MOR (Fig. 1l).

## Unique spatiotemporal MOR activation by DFNZ

Assessing in vitro functional kinetics of agonists can provide insights into their mechanism of action, temporal signalling profiles and therapeutic potential[15]. We performed in vitro kinetic bioluminescence resonance energy transfer (BRET) assays measuring $G_o$ activation and β-arrestin 2 recruitment (Fig. 2a). At $G_o$, FNZ and DFNZ showed efficacies at 2 min of 120.8% and 123.2%, respectively compared with DAMGO but these plateaued to 111.2% (9.6% decrease) and 98.4% (24.8% decrease) after 24 min, with the effect of DFNZ being significantly different from those of DAMGO and FNZ (Fig. 2b and Extended Data Fig. 4). At β-arrestin 2, FNZ showed significantly greater efficacy than DAMGO for the first 30 min, whereas DFNZ showed significantly higher efficacies than both DAMGO and FNZ during this time, and significantly greater efficacies than DAMGO over the full time course (Fig. 2c and Extended Data Fig. 4).

At $G_o$, potencies for all compounds increased over time, with DFNZ showing the highest potency (Fig. 2d). By contrast, at β-arrestin 2, FNZ potency was significantly higher than those of the other compounds (Fig. 2e). The higher potency at $G_o$ and lower potency at β-arrestin 2 of DFNZ resulted in significantly different G-protein versus β-arrestin preference between DFNZ and FNZ (Fig. 2f). Notably, the G-protein versus β-arrestin preference of DFNZ was driven by its high potency on the G-protein signalling pathway (Fig. 2d) rather than the low potency of its effect on β-arrestin recruitment (Fig. 2e).

MOR function is dependent on the spatiotemporal profile of the activated receptor[16], including its cellular localization and trafficking and its interactions with proteins downstream of its main signal transducers[17]. We performed proximity labelling of MOR using ascorbic acid peroxidase (APEX) followed by quantitative proteomics to examine the cellular localization and trafficking profile and proximal MOR protein interaction networks of FNZ, DFNZ and DAMGO (Fig. 2g). DFNZ led to slower MOR internalization and endosomal trafficking compared with DAMGO and FNZ (Fig. 2h and Supplementary Table 1), which correlated with its weaker labelling of β-arrestin 2, a regulator of MOR endocytosis, and extended labelling of EYA4, a modulator of $G_i$ signalling downstream of MOR activation[17] (Fig. 2i,j, Extended Data Fig. 4 and Supplementary Table 1). Overall, DFNZ showed a distinct MOR kinetic profile and a relative functional bias for G-protein signalling, suggesting altered receptor regulation and signalling dynamics compared with FNZ and DAMGO.

## Analgesic effects in acute and chronic pain models

In rats, subcutaneous FNZ injections produced potent and immediate hot plate antinociception (Fig. 3a), which lasted for at least 2 h (median effective dose ($ED_{50}$) ≈ 0.005 mg kg$^{-1}$) (Fig. 3b). The $ED_{50}$ for catalepsy (approximately 0.01 mg kg$^{-1}$) was significantly higher (around two-fold), and no significant decrease in body temperature was observed (Extended Data Fig. 5). DFNZ produced antinociception but with around 60-fold lower potency than FNZ ($ED_{50}$ ≈ 0.3 mg kg$^{-1}$) (Fig. 3c). The $ED_{50}$ for catalepsy (approximately 1 mg kg$^{-1}$) was significantly higher (around threefold), and a significant decrease in body temperature was observed only at the highest DFNZ dose, tenfold higher than its analgesic $ED_{50}$ (Extended Data Fig. 5).

In contrast to rats, MOR agonists produce hyperlocomotion in mice, which depends on their intrinsic efficacy, with full agonists producing maximal effects[18]. Morphine produced maximal and sustained effects at 30 mg kg$^{-1}$ whereas FNZ and DFNZ produced shorter effects that peaked at 0.1 mg kg$^{-1}$ (300-fold lower) and 3 mg kg$^{-1}$ (10-fold lower) respectively (Extended Data Fig. 6). None of the compounds produced hyperlocomotion in *Oprm1*-knockout mice (Extended Data Fig. 6).

To test whether FNZ produced antiallodynic responses in a model of peripheral inflammation, we injected complete Freund's adjuvant (CFA) into the left hind paw of mice and began daily subcutaneous injections of vehicle, FNZ (0.03 mg kg$^{-1}$) or DFNZ (1 mg kg$^{-1}$) starting 3 days after CFA (Fig. 3d). Antiallodynic responses were assessed 1 h after drug injection when hyperlocomotion had subsided (Extended Data Fig. 6). CFA significantly decreased paw withdrawal latency (Hargreaves) and threshold (von Frey) post injection on day 3, whereas FNZ and DFNZ treatment significantly reversed these responses (Fig. 3e,f). We tested additional doses of FNZ and DFNZ, confirming dose-dependent antinociception (Extended Data Fig. 6). Neither FNZ nor DFNZ triggered tolerance despite repeated daily administration of maximal analgesic doses, and neither produced antinociception in *Oprm1*-knockout mice (Extended Data Fig. 6).

## DFNZ exhibits impaired brain penetrance

To further study the in vivo properties of FNZ and DFNZ, we synthesized [$^{18}$F]FNZ (Fig. 4a and Extended Data Fig. 7), injected it intravenously into rats pretreated 15 min previously with saline or the MOR orthosteric antagonist naltrexone (1 mg kg$^{-1}$), and performed PET. [$^{18}$F]FNZ showed rapid brain entry and quick washout, and was detected in the brain only for the first 10 min after its injection in areas with high MOR density, an effect that was blocked by naltrexone (Fig. 4b and Extended Data Fig. 7).

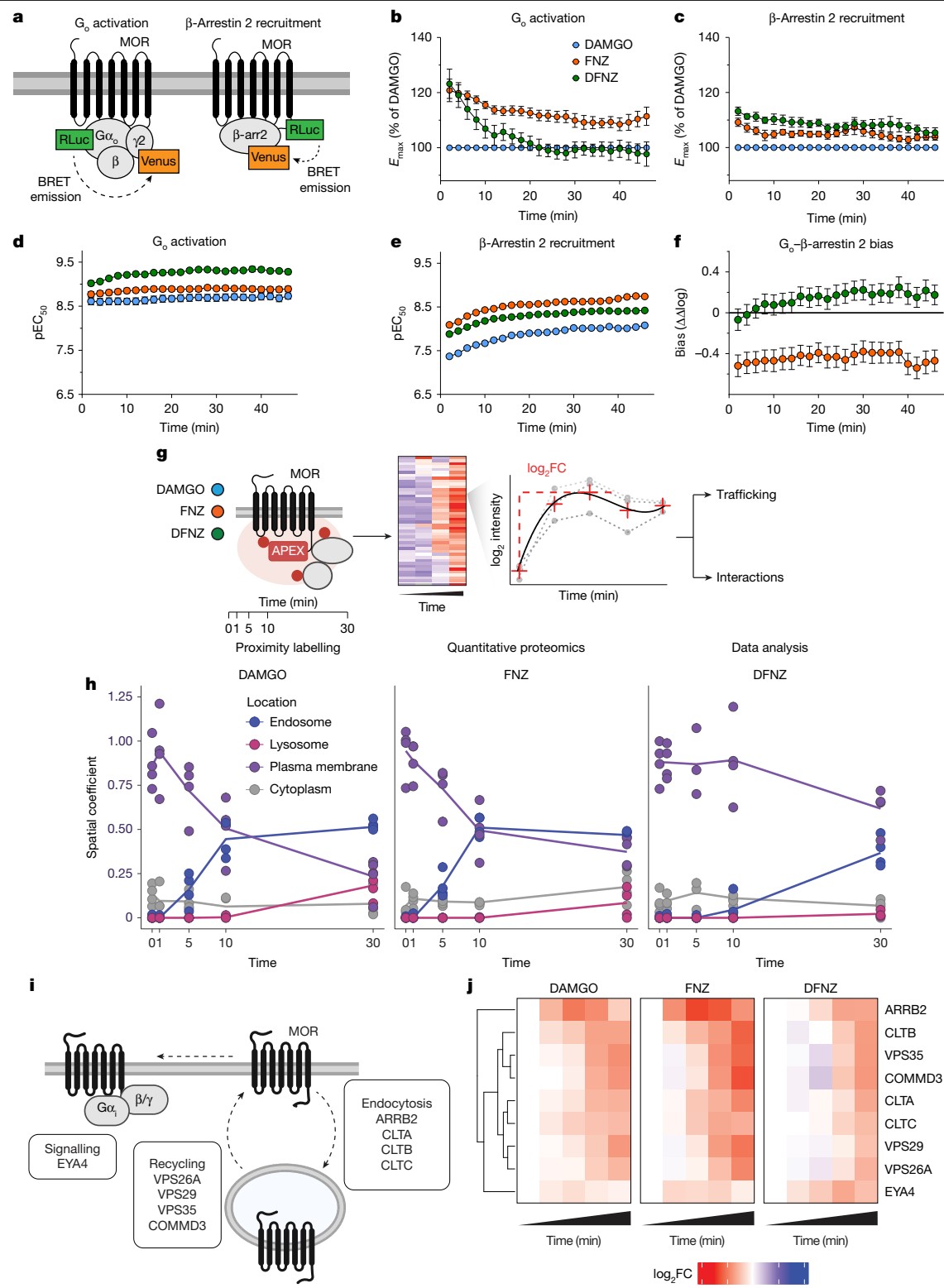

**Fig. 2 | FNZ and DFNZ have divergent spatiotemporal signalling profiles.**
**a**, Schematic of in vitro BRET kinetic functional assays to show MOR $G_o$ activation and β-arrestin 2 (β-arr2) recruitment. **b**–**e**, Efficacy as percentage of the DAMGO effect (% DAMGO) (**b**,**c**) or potency (pEC$_{50}$) (**d**,**e**) of FNZ ($n = 9$ independent experiments) and DFNZ ($n = 9$ independent experiments) over time. **b**, $G_o$ efficacy: two-way analysis of variance (ANOVA), drug–time interaction, $F(44, 902) = 3.46$, $P < 0.001$. **d**, $G_o$ potency: two-way ANOVA, drug effect, $F(2, 902) = 422.2$, $P < 0.001$. **c**, β-Arrestin efficacy: two-way ANOVA, drug effect, $F(2, 483) = 247.8$, $P < 0.001$. **e**, β-Arrestin potency: two-way ANOVA, drug effect, $F(2, 483) = 2,803$, $P < 0.001$. **f**, $G_o$–β-arrestin 2 bias of FNZ ($n = 8$ independent

experiments) and DFNZ ($n = 8$ independent experiments) (two-way ANOVA, drug effect, $F(1, 322) = 390.2$, $P < 0.001$). **g**, Schematic of the MOR–APEX approach following activation with DAMGO, FNZ and DFNZ. log$_2$FC, log$_2$-transformed fold change. **h**, Location-specific coefficients for each ligand and each time point to model receptor trafficking ($n = 4$ replicates per drug and time point). **i**, Schematic depicting known regulators of MOR signalling, endocytosis and trafficking. **j**, Heat map comparing log$_2$-transformed fold change in proximal labelling for the known regulators of MOR following receptor activation. Data are shown as mean ± s.e.m.

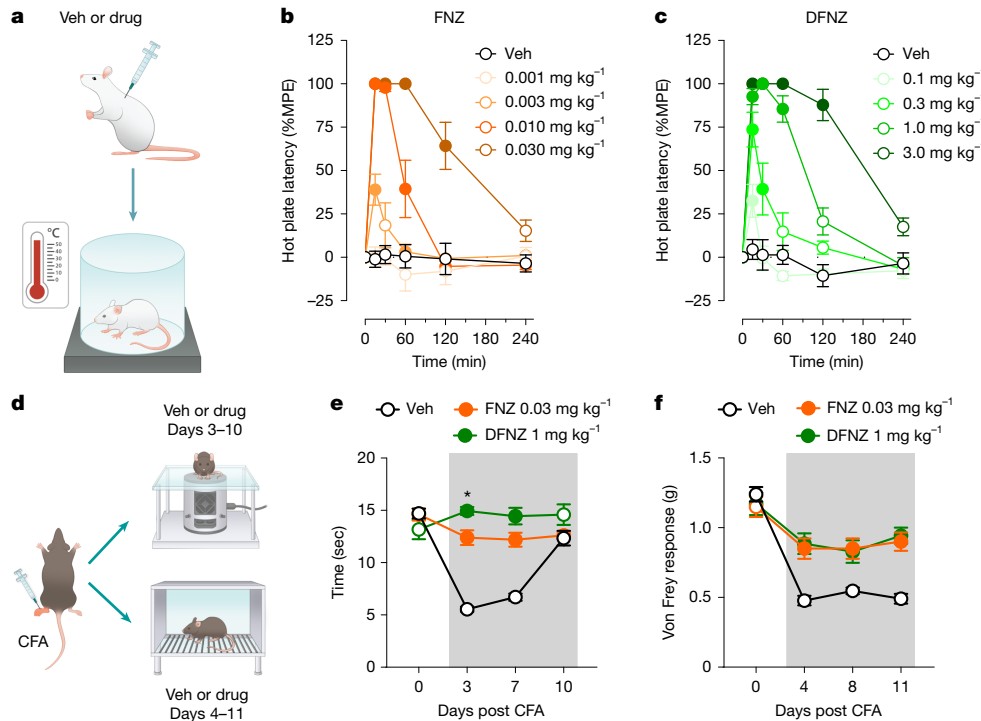

**Fig. 3 | FNZ and DFNZ are effective in antinociceptive assays. a**, Schematic of hot plate procedure. Veh, vehicle. **b,c**, Antinociceptive effects of FNZ (**b**; $n = 6$ rats per dose; two-way repeated measures ANOVA, dose–time interaction, $F_{(20,150)} = 16.63$, $P < 0.001$) and DFNZ (**c**; $n = 6$ rats per dose; two-way repeated measures ANOVA, dose–time interaction, $F_{(20,144)} = 15.33$, $P < 0.001$) on the hot plate. Closed circles indicate statistically significant difference compared with vehicle. MPE, maximum possible effect. **d**, Schematic showing Hargreaves and von Frey assessments in mice to repeated vehicle ($n = 15$ mice), 0.03 mg kg⁻¹ FNZ ($n = 8$ mice) or 1 mg kg⁻¹ DFNZ ($n = 7$ mice) in the CFA model of peripheral inflammation 1 h after drug administration. **e**, Hargreaves assessment as illustrated in **d**. Two-way repeated measures ANOVA, treatment–time interaction, $F_{(6, 81)} = 21.79$, $P < 0.001$. In the Hargreaves assessment, 1 mg kg⁻¹ DFNZ reversed responses to pre-CFA levels and induced significantly greater antinociception compared with 0.03 mg kg⁻¹ FNZ on day 3 after CFA treatment (Tukey post hoc test, two-sided, $P = 0.02$). **f**, von Frey assessment as illustrated in **d**. Two-way repeated measures ANOVA, treatment–time interaction, $F_{(6,810)} = 7.42$, $P < 0.001$. In **e**,**f**, the shaded area shows drug treatment days; closed circles indicate statistically significant difference compared with vehicle. Data are shown as mean ± s.e.m. *$P < 0.05$.

We performed similar experiments using [³H]FNZ, which accumulated in MOR-rich regions and was also blocked by naltrexone, confirming MOR occupancy (Fig. 4b and Extended Data Fig. 7).

Since the fluoroethyl group of FNZ is retained after N-dealkylation, the ¹⁸F radiolabel is retained for the metabolite ([¹⁸F]FNZ to [¹⁸F]DFNZ). Therefore, if DFNZ has high brain penetrance, metabolized [¹⁸F]DFNZ should be detected in the brain after [¹⁸F]FNZ injection[19]. However, no specific uptake was observed in the brain starting 10 min after [¹⁸F]FNZ injection, suggesting that DFNZ has impaired brain penetrance.

Efflux transporters at the blood–brain barrier have a critical role in actively preventing potentially harmful substances, including certain opioids, from accumulating in the brain[20,21], which influences in vivo drug effects and analgesic efficacy[5]. We first screened the drugs using a transporter inhibition panel and then performed targeted efflux transporter substrate assays. FNZ was an inhibitor of multidrug and toxin extrusion protein 2-K (MATE2-K) and organic cation transporter 2 (OCT2) whereas DFNZ was an inhibitor of MATE1 (Extended Data Fig. 7). Since MATE1 is expressed in the brain, albeit at low levels[22], we tested whether the drugs were transported by MATE1, but neither drug was a MATE1 substrate (Extended Data Fig. 7). Finally, we tested whether FNZ and DFNZ were substrates of P-glycoprotein (PGP) and breast cancer resistance protein (BCRP), the two major efflux transporters of the blood–brain barrier[23]. DFNZ was a substrate of both PGP and BCRP, whereas FNZ was not (Fig. 4c). To confirm whether DFNZ is a PGP and BCRP substrate in vivo, we performed [³H]DFNZ uptake studies in wild-type mice and mice lacking both PGP and BCRP (PGP/BCRP-knockout mice). [³H]DFNZ showed weak and exclusive accumulation in the ventricles of wild-type mice, whereas PGP/BCRP-knockout mice showed strong and widespread [³H]DFNZ accumulation in the brain, confirming in vivo efflux of DFNZ by PGP and BCRP (Fig. 4d and Extended Data Fig. 7).

To further assess the pharmacokinetics of FNZ and DFNZ, we calculated their concentrations in brain and plasma. A brain-to-plasma ratio below 1 is indicative of active brain uptake whereas a ratio of greater than 1 is indicative of active brain clearance and low accumulation[24]. Opioids that produce effects on the central nervous system, such as fentanyl, oxycodone, buprenorphine and methadone[25–28] have ratios above 2.8, whereas the PGP substrate loperamide[29], which is inactive in the central nervous system, has a ratio below 1. FNZ and DFNZ showed plasma protein binding of 94.6% and 91.5%, respectively. In mice, treatment with 0.03 mg kg⁻¹ FNZ led to higher unbound concentration in the brain (0.91 ± 0.05 nM) than in plasma (0.51 ± 0.05 nM). Peak brain FNZ concentrations were observed at 10–15 min after injection, and FNZ was no longer detected in the brain after 60 min (Fig. 4e). By contrast, a tenfold higher dose of DFNZ (0.3 mg kg⁻¹) led to lower unbound concentrations in brain (12.24 ± 1.62 nM) than in plasma (27.4 ± 2.2 nM) at 15 min (Fig. 4f,g). In rats, at 15 min after treatment with 0.01 mg kg⁻¹ FNZ, unbound drug was detected in brain at 0.37 ± 0.05 nM and in plasma at 0.15 ± 0.02 nM (Fig. 4h,i). At a tenfold higher dose of FNZ (0.1 mg kg⁻¹), unbound drug was detected at 6.45 ± 0.76 nM in brain and 2.23 ± 0.22 nM in plasma. At the same 0.1 mg kg⁻¹ dose of DFNZ, unbound drug was detected at only 2.74 ± 0.18 nM in brain and 1.94 ± 0.2 nM in plasma. At 1 mg kg⁻¹ DFNZ, unbound drug in brain was detected at 35.4 ± 2.0 nM and in plasma at 40.4 ± 6.7 nM. The rat dose–response brain/plasma ratios ranged from 2.4–3.2 for FNZ and 0.9–1.5 for DFNZ (Fig. 4j).

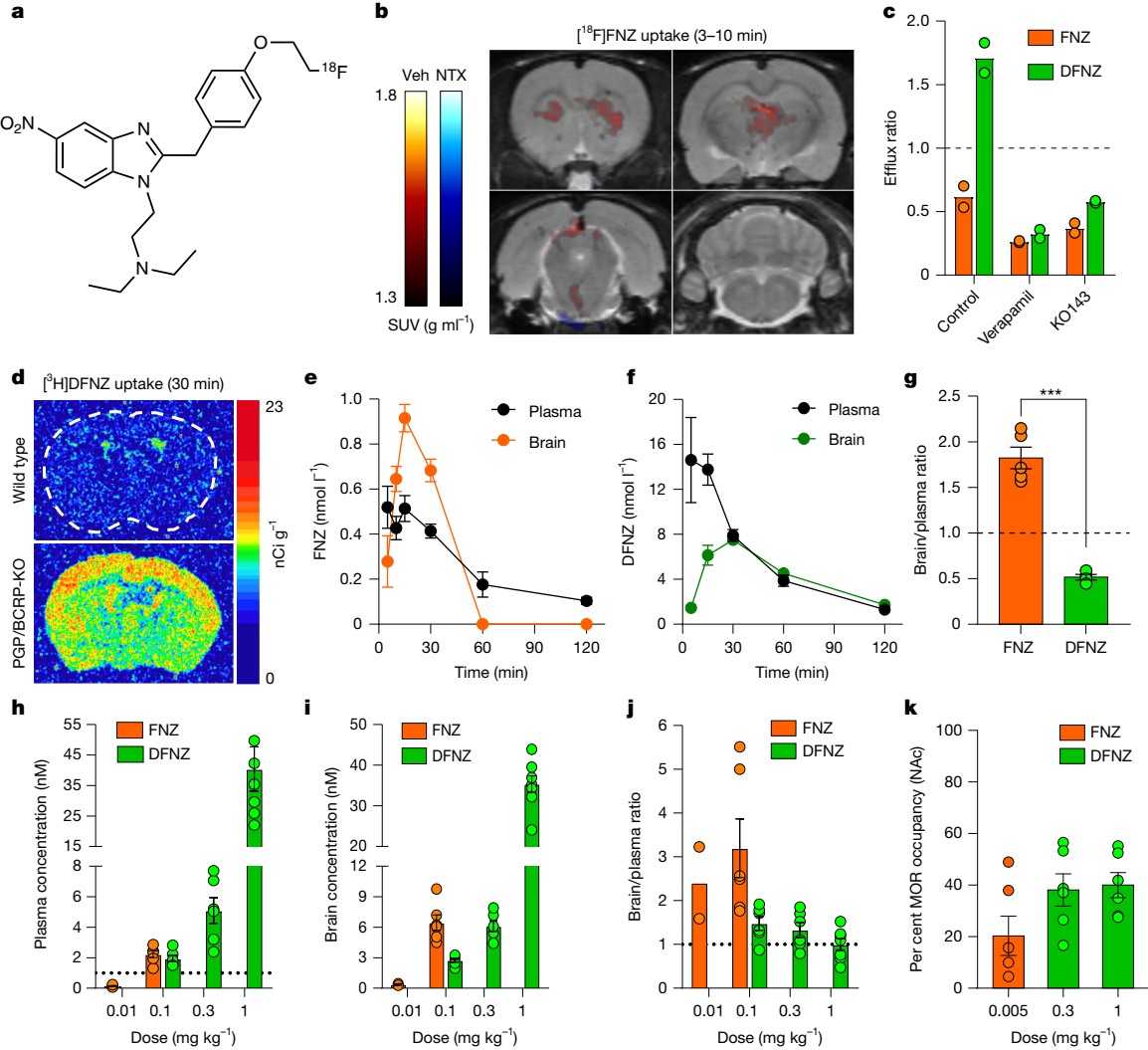

**Fig. 4 | FNZ and DFNZ show divergent brain penetrance and pharmacokinetics. a**, [18F]FNZ structure. **b**, PET standard uptake value (SUV) images of [18F]FNZ brain uptake in rats pretreated with vehicle (*n* = 3 rats) or naltrexone (NTX) (*n* = 3 rats). **c**, Efflux ratios of FNZ (*n* = 2 replicates) and DFNZ (*n* = 2 replicates) in the Caco-2 permeability assay with and without verapamil (a PGP inhibitor) and KO143 (BCRP inhibitor). **d**, [3H]DFNZ uptake in brain 30 min after injection in wild-type (*n* = 2 mice) or PGP/BCRP-knockout (KO) (*n* = 2 mice) mice. **e**–**g**, Unbound brain and plasma FNZ (**e**) and DFNZ (**f**) concentrations and brain/plasma ratios of FNZ and DFNZ (**g**) in male mice (*n* = 5 mice per drug). FNZ had a significantly higher brain/plasma ratio (approximately 1.8) than DFNZ (approximately 0.5) (unpaired *t*-test, two-sided, *t*(7) = 9.49, *P* < 0.001).

**h**–**j**, Estimated unbound plasma (**h**: 0.01 mg kg⁻¹ FNZ, *n* = 6 rats; 0.1 mg kg⁻¹ FNZ, *n* = 6 rats; 0.1 mg kg⁻¹ DFNZ, *n* = 6 rats; 0.3 mg kg⁻¹ DFNZ, *n* = 6 rats; 3 mg kg⁻¹ DFNZ, *n* = 7 rats) and brain (**i**: 0.01 mg kg⁻¹ FNZ, *n* = 2 rats; 0.1 mg kg⁻¹ FNZ, *n* = 6 rats; 0.1 mg kg⁻¹ DFNZ, *n* = 6 rats; 0.3 mg kg⁻¹ DFNZ, *n* = 6 rats; 3 mg kg⁻¹ DFNZ, *n* = 8 rats) concentrations and brain/plasma ratios (**j**: 0.01 mg kg⁻¹ FNZ, *n* = 2 rats; 0.1 mg kg⁻¹ FNZ, *n* = 6 rats; 0.1 mg kg⁻¹ DFNZ, *n* = 6 rats; 0.3 mg kg⁻¹ DFNZ, *n* = 6 rats; 3 mg kg⁻¹ DFNZ, *n* = 7 rats) across different concentrations (FNZ was not tested above 0.1 mg kg⁻¹). **k**, MOR occupancy of FNZ and DFNZ in rats (*n* = 6 rats per group) using [3H]DAMGO in NAc. Data are shown as mean ± s.e.m. ****P* < 0.001.

To confirm brain MOR occupancy, we injected rats with analgesic doses of FNZ or DFNZ and performed ex vivo MOR occupancy using [3H]DAMGO autoradiography. Despite the 60- to 200-fold higher doses, and 16- to 90-fold higher brain concentrations, 0.3 and 1 mg kg⁻¹ DFNZ produced MOR occupancy resembling that produced by 5 µg kg⁻¹ FNZ treatment, indicating that DFNZ exhibits limited MOR occupancy in the brain (Fig. 4k and Extended Data Fig. 7).

## DFNZ does not cause brain hypoxia at analgesic doses

We next used implantable sensors coupled with amperometry to measure oxygen levels in the nucleus accumbens (NAc) of freely moving rats injected intravenously with FNZ, DFNZ or fentanyl. We previously showed that intravenous fentanyl showed more than tenfold greater potency in promoting brain hypoxia compared with intraperitoneal fentanyl[30]. At equipotent doses, 0.03 mg kg⁻¹ intravenous fentanyl

(1.5-fold greater than the analgesic ED₅₀ (ref. 31)) produced significantly greater hypoxia than 0.01 mg kg⁻¹ intravenous FNZ (twofold greater than the analgesic ED₅₀) (Fig. 5a and Extended Data Fig. 8), which produced comparable hypoxia to fentanyl only at a dose 6 times greater than its subcutaneous analgesic ED₅₀ (that is, 0.03 mg kg⁻¹). DFNZ did not cause a significant decrease in NAc oxygen levels at an intravenous dose of 0.1 mg kg⁻¹, a dose comparable to the subcutaneous dose that produced maximal analgesia (1 mg kg⁻¹) (Fig. 5a and Extended Data Fig. 8), but produced a small decrease in NAc oxygen levels at the very high intravenous dose of 0.3 mg kg⁻¹ (equivalent to a 3 mg kg⁻¹ subcutaneous injection). However, this effect was significantly lower than that of both 0.03 mg kg⁻¹ intravenous fentanyl (1.5-fold greater than the analgesic ED₅₀) and 0.03 mg kg⁻¹ intravenous FNZ (sixfold greater than the analgesic ED₅₀).

To further assess brain oxygen responses at the therapeutic (maximal analgesic) dose, we injected rats subcutaneously with saline or 1 mg kg⁻¹

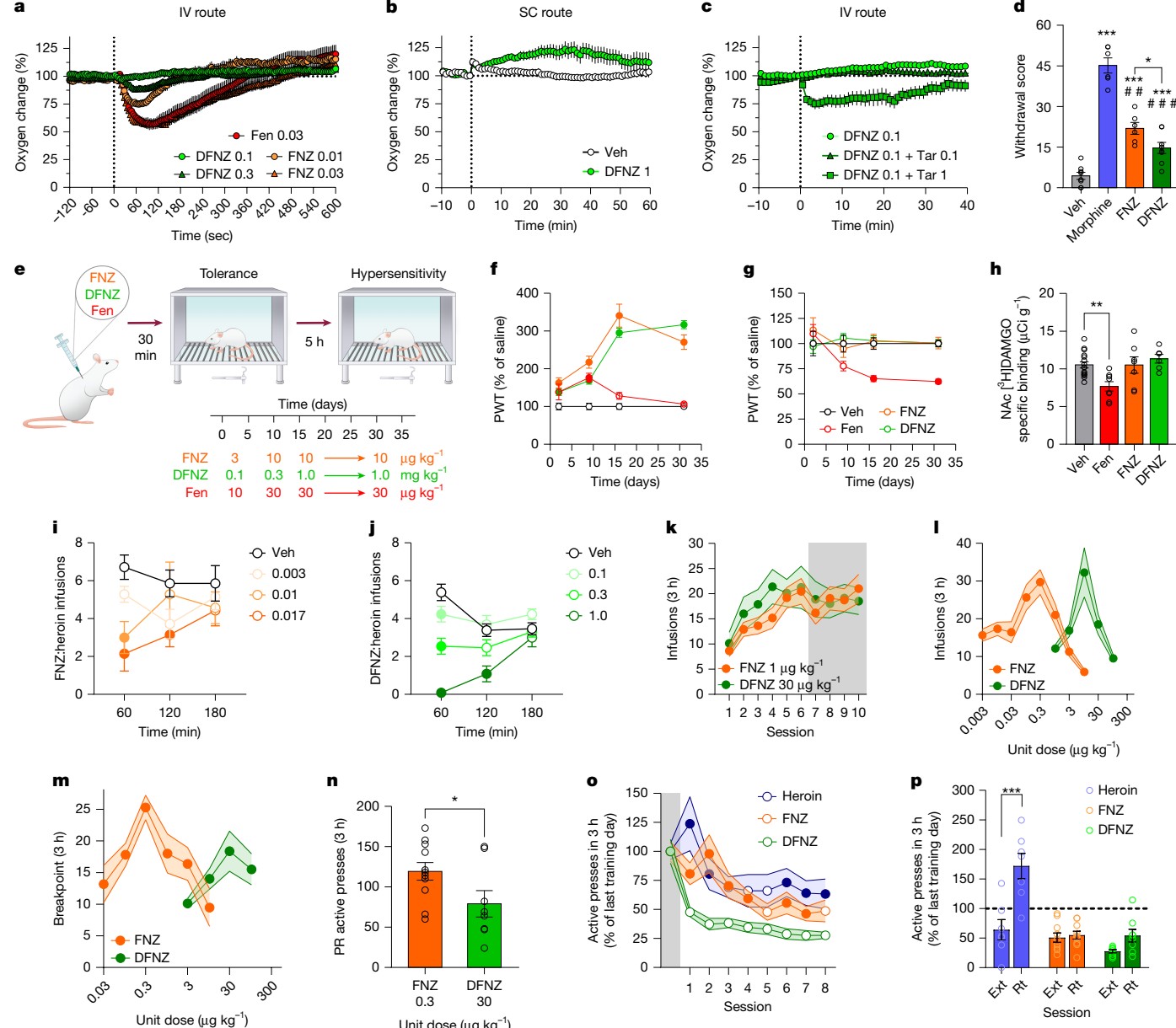

**Fig. 5 | DFNZ has a strong safety profile. a–c**, Hypoxia after intravenous (IV) FNZ ($n$ = 14 independent experiments), DFNZ ($n$ = 14 independent experiments) or fentanyl (Fen) ($n$ = 8 independent experiments) (**a**), subcutaneous (SC) vehicle ($n$ = 9 independent experiments) or DFNZ ($n$ = 9 independent experiments; two-way repeated measures ANOVA, treatment–time interaction, $F_{(61, 915)}$ = 4.24, $P$ < 0.001) (**b**), and (IV) tariquidar (Tar) ($n$ = 13 independent experiments) and DFNZ ($n$ = 7 independent experiments; two-way ANOVA, treatment–time interaction, $F_{(100, 1071)}$ = 2.37, $P$ < 0.001) (**c**). Drug concentrations are in mg kg$^{-1}$. **d**, Withdrawal after vehicle ($n$ = 8 rats), morphine ($n$ = 6 rats), FNZ ($n$ = 6 rats) or DFNZ ($n$ = 8 rats) (ANOVA, $F_{(3, 24)}$ = 67.01, $P$ < 0.001; Holm–Sidak post hoc tests, $P$ < 0.02). Asterisks (*) indicate significant differences compared with vehicle and pound (#) signs indicate significant differences compared with morphine (##$P$ < 0.01, ###$P$ < 0.001). **e**, Schematic of von Frey assessments in **f**,**g**. **f**,**g**, Paw withdrawal threshold (PWT) at 30 min (**f**; two-way repeated measures ANOVA, drug–dose interaction, $F_{(9, 84)}$ = 19.29, $P$ < 0.001) or 5 h (**g**; two-way repeated measures ANOVA, drug–dose interaction, $F_{(9, 84)}$ = 3.07, $P$ = 0.003) after vehicle ($n$ = 16 rats), FNZ ($n$ = 8 rats), DFNZ ($n$ = 8 rats) or fentanyl ($n$ = 8 rats). Closed circles indicate significant differences compared with vehicle. **h**, MOR density

after vehicle ($n$ = 16 rats), FNZ ($n$ = 7 rats), DFNZ ($n$ = 6 rats) or fentanyl ($n$ = 8 rats) (ANOVA, $F_{(3, 33)}$ = 5.47, $P$ = 0.003; Dunnett's post hoc test, $P$ = 0.004). **i**,**j**, FNZ (**i**; $n$ = 7 rats; two-way repeated measures ANOVA, dose effect, $F_{(3, 18)}$ = 9.25, $P$ < 0.001) and DFNZ (**j**; $n$ = 13 rats; treatment–time interaction, $F_{(6, 69)}$ = 9.64, $P$ < 0.001) decrease heroin self-administration. Drug concentrations are in mg kg$^{-1}$. Closed circles indicate significant differences compared with vehicle. **k–n**, Acquisition (**k**: white background, fixed ratio 1 (FR1); grey background, FR3). FR3 dose–response (**l**), breakpoint (**m**; progressive ratio (PR)) and progressive ratio response (**n**: unpaired $t$-test, two-sided, $t_{(17)}$ = 2.13, $P$ = 0.04) during FNZ ($n$ = 11 rats) or DFNZ ($n$ = 8 rats) self-administration. **o**, FNZ ($n$ = 11 rats), DFNZ ($n$ = 8 rats) or heroin ($n$ = 10 rats) extinction (two-way repeated measures ANOVA, drug–time interaction, $F_{(16, 208)}$ = 2.46, $P$ = 0.002). Closed circles indicate significant differences compared with DFNZ. **p**, FNZ ($n$ = 10 rats), DFNZ ($n$ = 8 rats) or heroin ($n$ = 7 rats) reinstatement (two-way repeated measures ANOVA, drug–session interaction, $F_{(2, 22)}$ = 18.55, $P$ < 0.001; Sidak's post hoc test, $P$ < 0.001). Ext, extinction; Rt, reinstatement. Data are shown as mean ± s.e.m. *$P$ < 0.05, **$P$ < 0.01, ***$P$ < 0.001.

DFNZ and repeated the experiments. Unexpectedly, DFNZ induced a tonic, moderate and significant increase in brain oxygen levels that lasted for at least 60 min (Fig. 5b).

Finally, to test whether the interaction of DFNZ with PGP contributes to its respiratory profile, we repeated the above experiments using 0.1 mg kg$^{-1}$ intravenous DFNZ in rats pretreated with the PGP inhibitor

tariquidar. As in prior experiments, 0.1 mg kg$^{-1}$ intravenous DFNZ did not produce brain hypoxia (Fig. 5c). However, tariquidar pretreatment followed by 0.1 mg kg$^{-1}$ intravenous DFNZ significantly decreased NAc oxygen levels for over 30 min (Fig. 5c), confirming that the safe respiratory profile of DFNZ is due to its interaction with PGP.

## DFNZ has a weak effect on inducing withdrawal

To assess opioid withdrawal, we injected rats subcutaneously with FNZ or DFNZ twice per day for 7 days, at doses threefold higher than their analgesic ED$_{50}$ (0.03 and 1 mg kg$^{-1}$, respectively). On day 8, we injected each drug again followed by the MOR antagonist naloxone (1 mg kg$^{-1}$) to precipitate withdrawal. For comparison, we also tested morphine[32]. As expected, morphine produced significant precipitated withdrawal, whereas DFNZ, and to a lesser degree FNZ produced significantly lower withdrawal (approximately 70% lower) than morphine (Fig. 5d). Specifically, morphine significantly induced 8 out of 14 withdrawal symptoms, whereas FNZ and DFNZ treatment significantly induced 3 out of 14 and 1 out of 14 withdrawal symptoms respectively, indicating that FNZ, and especially DFNZ, have weak withdrawal effects (Extended Data Fig. 8).

## DFNZ does not cause tolerance or MOR density changes

To test whether repeated exposure to FNZ or DFNZ produces mechanical sensitivity changes characteristic of tolerance and allodynia[33], we injected rats for 5 days per week with each drug or with fentanyl, escalating the doses over 1 month. We measured changes in mechanical sensitivity using von Frey assay twice per day at various days after the start of injections (Fig. 5e). We then euthanized the rats to assess MOR density. As previously described[34], repeated fentanyl at the analgesic ED$_{50}$[31] caused tolerance and mechanical hypersensitivity (Fig. 5f,g and Extended Data Fig. 9) and these effects were associated with decreased NAc MOR density (Fig. 5h). By contrast, repeated injections of FNZ or DFNZ did not cause tolerance, mechanical hypersensitivity or MOR density changes (Fig. 5f–h and Extended Data Fig. 9).

## DFNZ decreases heroin self-administration

We used intravenous self-administration (IVSA), the gold standard procedure for predicting abuse liability of drugs in humans[35–37] to assess whether FNZ and DFNZ would decrease heroin intake. Rats were trained for heroin IVSA and then pretreated subcutaneously with FNZ, DFNZ or fentanyl immediately prior to each session. We also tested the effects of the drugs on palatable food self-administration. Similar to fentanyl (Extended Data Fig. 9), analgesic doses of FNZ and DFNZ significantly decreased heroin self-administration (Fig. 5i,j). At the highest dose, the drugs also decreased food self-administration (Extended Data Fig. 9), suggesting that the inhibitory effect of the highest doses on heroin IVSA is due to motor deficits. Overall, FNZ and DFNZ substituted for heroin IVSA, implying that both drugs may serve as operant reinforcers in this procedure.

## DFNZ has weak reinforcing effects

Rats self-administered heroin at 0.1 mg kg$^{-1}$ per infusion (Extended Data Fig. 9) and FNZ and DFNZ were self-administered at 0.001 mg kg$^{-1}$ per infusion (100-fold lower than heroin) and 0.03 mg kg$^{-1}$ per infusion (threefold lower than heroin), respectively (Fig. 5k and Extended Data Fig. 9). In dose–response assessments, FNZ produced stronger reinforcement than DFNZ under both fixed ratio 3 (FR3) and progressive ratio schedules (Fig. 5l,m). Under the progressive ratio schedule, rats pressing for DFNZ reached a significantly lower breakpoint than with FNZ (Fig. 5n).

Extinction responding in the IVSA procedure is used to study relapse-related behaviour during abstinence[38,39]. Rats showed the typical 'extinction burst' for the heroin-paired lever and maintained pressing for the FNZ-paired lever for several days (Fig. 5o and Extended Data Fig. 9). By contrast, rats immediately extinguished lever pressing for the DFNZ-paired lever (Fig. 5o and Extended Data Fig. 9). At the last extinction session, the rats had decreased lever pressing for heroin and FNZ, but the rate of pressing for heroin was still higher compared with DFNZ. After the last session, the rats were injected intravenously with reinforcing doses of each drug to examine reinstatement of drug seeking, a classical measure of drug relapse[40]. Heroin priming significantly increased drug-paired lever pressing, surpassing the pre-extinction pressing rate. By contrast, rats injected with either FNZ or DFNZ did not reinstate drug seeking (Fig. 5p). Together, unlike heroin and other MOR agonists, FNZ and DFNZ did not produce the typical extinction burst or reinstatement of drug seeking. Additionally, DFNZ specifically showed very rapid extinction of drug-reinforced responding.

## DFNZ has limited efficacy on dopamine activity

The rate at which MOR agonists enter the brain, along with their intrinsic efficacy influences their pharmacodynamics and ability to increase dopamine neurotransmission in NAc and other regions involved in the rewarding effects of addictive drugs[41]. Phasic dopamine responses are hypothesized to strengthen associations between drug cues and drug delivery, facilitating reinforcement learning and increasing risk of drug craving, relapse and addiction[42,43]. Tonic dopamine responses are hypothesized to modulate motivation and motor responses without enabling reinforcement learning[42,43].

To examine dopamine neuronal activity, mice expressing Cre recombinase in tyrosine hydroxylase (TH)-expressing neurons (TH-Cre mice) received injections of a Cre-dependent adeno-associated virus (AAV) encoding an axonal-targeted fluorescent calcium indicator[44] (AAV-hSyn-FLEx-axon-GCaMP6s) into the ventral tegmental area (VTA). We then implanted a fibre optic cannula into NAc to perform fibre photometry of GCaMP6s activity in VTA→NAc dopamine terminals (Fig. 6a,b).

FNZ produced rapid, dose-dependent, statistically significant increases in dopamine neuron activity (Fig. 6c). DFNZ also produced a significant increase in dopamine neuron activity, but with significantly lower magnitude and a delayed temporal pattern (Fig. 6c). Unlike FNZ, increasing the dose of DFNZ had limited effects on dopamine neuron activation (Fig. 6c). Peak locomotor doses of morphine and fentanyl produced a significant increase in dopamine neuron activity, with morphine producing the strongest and most sustained response, whereas fentanyl showed a comparable effect to DFNZ (Extended Data Fig. 10).

At 0.1 mg kg$^{-1}$, FNZ produced a significant increase in the frequency of GCaMP6s spikes in dopamine neurons compared with a lower analgesic dose of FNZ (0.03 mg kg$^{-1}$) and all three doses of DFNZ (Fig. 6d). DFNZ did not significantly increase the frequency of GCaMP6s spikes at any tested dose (Fig. 6d). Peak locomotor doses of morphine and fentanyl produced significant increases in the frequency of GCaMP6s spikes, similar to FNZ but not DFNZ (Extended Data Fig. 10). No differences in dopamine neuron spike amplitude (Fig. 6e) or spike duration (Fig. 6f) were observed.

Next, we tested the effects of FNZ, DFNZ, fentanyl and morphine on NAc dopamine responses using fibre photometry and the genetically encoded dLight1.3b sensor (AAV-hSyn-dLight1.3b), which measures fast (phasic) and slow (tonic) dopamine responses[45] (Fig. 6g,h). FNZ caused rapid, dose-dependent, statistically significant increases in slow (tonic) NAc dopamine responses at both analgesic (0.03 mg kg$^{-1}$) and peak locomotor doses (0.1 mg kg$^{-1}$) (Fig. 6i). DFNZ also produced significant increases in slow (tonic) dopamine, but these effects were delayed compared with 0.1 mg kg$^{-1}$ FNZ and were significantly different from saline only at the highest DFNZ dose (3 mg kg$^{-1}$) (Fig. 6i).

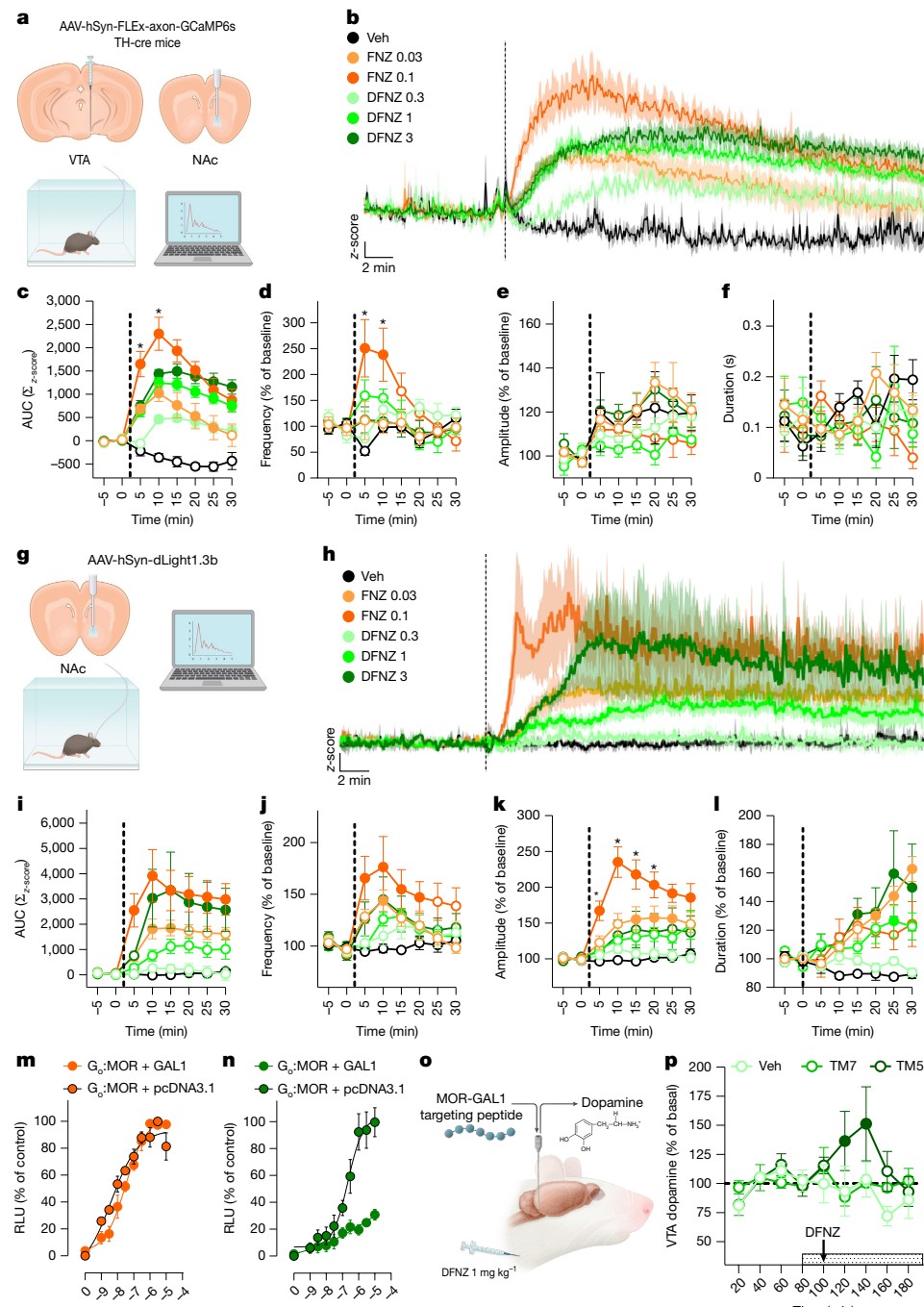

**Fig. 6 | DFNZ has limited effects on dopamine neurotransmission.**
**a**, Axon-GCaMP6s fibre photometry in NAc dopamine neurons. **b**, Averaged axon-GcaMP6s traces ($n = 5$ mice per drug, concentrations in mg kg$^{-1}$). The vertical line indicates time of injection. **c–f**, Area under the curve (AUC) analysis (**c**: $n = 5$ mice per drug; mixed-effects ANOVA, treatment–time interaction, $F(35, 138) = 10.97$, $P < 0.001$; Tukey post hoc tests, $P < 0.05$), spike frequency (**d**: $n = 5$ mice per drug; mixed-effects ANOVA, treatment–time interaction, $F(35, 138) = 3.17$, $P < 0.001$; Tukey post hoc tests, $P < 0.05$), spike amplitude (**e**: $n = 5$ mice per drug; mixed-effects ANOVA, treatment, $P = 0.19$; treatment–time interaction, $P = 0.5$) and spike duration (**f**: $n = 5$ mice per drug; mixed-effects ANOVA, treatment, $P = 0.93$; treatment–time interaction, $P = 0.08$). **g**, dLight1.3b photometry in NAc. **h**, Averaged dLight traces (vehicle, $n = 5$ mice; 0.03 mg kg$^{-1}$ FNZ, $n = 6$ mice; 0.1 mg kg$^{-1}$ FNZ, $n = 6$ mice; 0.3 mg kg$^{-1}$ DFNZ, $n = 5$ mice; 1 mg kg$^{-1}$ DFNZ, $n = 9$ mice; 3 mg kg$^{-1}$ DFNZ, $n = 4$ mice, mg kg$^{-1}$). The vertical line indicates time of injection. **i–l**, AUC (**i**; mixed-effects ANOVA, treatment–time interaction, $F(35, 202) = 5.32$, $P < 0.001$), transient frequency (**j**; two-way repeated measures ANOVA, treatment–time interaction,

$F(35, 196) = 2.68$, $P < 0.001$), amplitude (**k**; two-way repeated measures ANOVA, treatment–time interaction, $F(35, 194) = 4.60$, $P < 0.001$; Tukey post hoc tests, $P < 0.05$) and duration (**l**; two-way repeated measures ANOVA, treatment–time interaction, $F(35, 196) = 3.32$, $P = 0.001$). Number of mice as in **h**. Closed circles indicate significant differences compared with vehicle. Asterisks indicate significant differences compared with 3 mg kg$^{-1}$ DFNZ. **m,n**, NanoBiT-based $G_o$ protein engagement assay with FNZ (**m**) or DFNZ (**n**) in cells co-transfected with MOR and control vector ($n = 2$ independent experiments) or GAL1 ($n = 2$ independent experiments). RLU, relative light units. **o**, Schematic of microdialysis experiments. **p**, Infusion of the MOR–GAL1 heteromer-disrupting TM5 peptide (horizontal dashed line) increased somatodendritic dopamine response to DFNZ in VTA (vehicle, $n = 5$ rats; TM5, $n = 5$ rats; TM7 (control), $n = 8$ rats). Two-way repeated measures ANOVA, time main effect, $F(8, 120) = 2.06$, $P = 0.04$. Closed circles indicate significant differences compared with TM7. The arrow indicates DFNZ injection. All statistical tests were corrected for multiple comparisons. Data are shown as mean ± s.e.m. *$P < 0.05$.

Peak locomotor doses of morphine and fentanyl produced weaker slow (tonic) dopamine responses compared with FNZ and DFNZ (Extended Data Fig. 10).

FNZ produced significant increases in the frequency and amplitude of fast (phasic) dopamine transients, whereas DFNZ did not (Fig. 6j,k). Fentanyl and FNZ produced comparable and significant increases in fast (phasic) dopamine transient frequency, whereas morphine resembled DFNZ in this respect (Extended Data Fig. 10). FNZ produced fast (phasic) dopamine transients of greater amplitude than all other drugs, whereas morphine and fentanyl were comparable to DFNZ (Extended Data Fig. 10). Finally, 0.03 mg kg$^{-1}$ FNZ and both doses of DFNZ produced significant increases in dopamine transient duration over time but 0.1 mg kg$^{-1}$ FNZ did not (Fig. 6l), and resulted in significantly smaller transient duration than DFNZ, morphine or fentanyl (Extended Data Fig. 10). Overall, DFNZ produced divergent and limited effects on dopamine neurotransmission compared with FNZ, morphine and fentanyl, showing a relative preference for promoting slow (tonic) dopamine over fast (phasic) dopamine transients and with limited capacity to stimulate dopamine neurotransmission.

## DFNZ has lower efficacy at the MOR–GAL1 heteromer

We previously found that MORs that heteromerize with GAL1 constitute a significant population of MORs localized in the VTA and that these MOR–GAL1 heteromers mediate the dopaminergic effects of MOR agonists[46]. Weak efficacy or loss of efficacy at MOR–GAL1 heteromers are associated with the lower reinforcing effects of racemic methadone compared with other opioids[46] and the lack of (S)-methadone reinforcement and abuse liability[47,48]. We therefore explored whether the qualitative differences in dopamine neurotransmission between FNZ and DFNZ are due to pharmacodynamic differences at MOR–GAL1 heteromers.

(S)-methadone, which we previously showed acts as an agonist at MOR but a competitive antagonist at VTA MOR–GAL1 heteromers[47], showed a significant MOR concentration–response curve, but lost its efficacy when cells were co-transfected with GAL1 (Extended Data Fig. 10). In the same assay, FNZ showed about the same potency and efficacy in cells with and without co-transfected GAL1 (Fig. 6m). By contrast, DFNZ showed a much lower efficacy when cells were co-transfected with GAL1 (Fig. 6n), indicating that DFNZ, but not FNZ, is less efficacious in cells co-expressing MOR and GAL1.

Using microdialysis, we previously showed that MOR–GAL1 heteromers are critical to MOR agonist-induced increases in extracellular dopamine levels in VTA[46,47]. To test whether MOR–GAL1 influences the dopaminergic effects of DFNZ, we implanted rats with a microdialysis cannula targeted to the VTA and measured extracellular dopamine before and after intraperitoneal injections of a maximal analgesic dose (1 mg kg$^{-1}$) of DFNZ alone, combined with intra-VTA infusion of a synthetic control peptide targeting TM7, or a peptide targeting the TM5 portion of MOR that selectively disrupts MOR–GAL1 heteromerization[46] (Fig. 6o). As a positive control, we injected FNZ (0.1 mg kg$^{-1}$), which significantly increased VTA dopamine levels (Extended Data Fig. 10). At 1 mg kg$^{-1}$, DFNZ did not increase extracellular dopamine levels in VTA, whereas infusion of the MOR–GAL1 heteromer-destabilizing TM5 peptide, but not the control TM7 peptide, produced a significant increase in extracellular dopamine (Fig. 6p), indicating that VTA MOR–GAL1 heteromers are critical to the weak dopaminergic effects of DFNZ.

## Discussion

Despite their nearly identical structures, DFNZ exhibited reduced in vivo potency and distinct pharmacological properties compared with FNZ and other structurally unrelated MOR agonists. Of note, DFNZ showed an unexpectedly strong safety profile for a nitazene MOR superagonist.

The drug produced potent but transient MOR superagonism in vitro and induced a unique MOR cellular spatiotemporal activation profile compared with FNZ and DAMGO. Despite its supraphysiological efficacy at both G-protein activation and β-arrestin recruitment, the MOR spatiotemporal activation profile of DFNZ resembled those of agonists with lower efficacy such as morphine and PZM21[17]. One potential explanation for this is that similar to morphine and PZM21[49], DFNZ showed relative functional bias for G-protein activation over β-arrestin recruitment.

At therapeutic doses, DFNZ produced a moderate and sustained increase in brain oxygen, indicating that at this dose it did not activate MORs in respiratory nuclei. Moreover, repeated therapeutic doses of DFNZ did not produce tolerance or withdrawal effects related to metabolic, autonomic, behavioural or motor systems, and only produced a significant increase in 1 out of the 14 withdrawal symptoms (irritability). From the perspective of pain medication, these are strong therapeutic advantages.

Following DFNZ self-administration, rats showed immediate extinction of operant responding in the presence of drug-paired cues, a key component of addiction[50], suggesting that DFNZ has weak reinforcement. Immediate extinction of drug self-administration has been previously observed with ketamine[51], a weak MOR agonist that is self-administered by rats[51] and monkeys[52] with low abuse liability[53,54] and short-lasting effects on dopamine neurotransmission[54,55]. We speculate that the rapid extinction of DFNZ-reinforced responding is due to its unique effects on NAc dopamine neurotransmission and specifically its inability to induce phasic dopamine neuron activity and dopamine release. Indeed, cues and contexts associated with drug self-administration typically drive lever pressing during extinction sessions, and phasic dopamine in NAc has a critical role in reinforcement learning, conditioned drug effects, and cue- and context-induced relapse of opioid and psychostimulant seeking[42,43,50]. Phasic dopamine is hypothesized to represent the high frequency 'signal' component driving learning aspects of behaviour, motivation and value, whereas tonic dopamine is hypothesized to modulate the background 'noise' level, regulating behavioural exploration[56,57]. The balance between tonic and phasic dopamine responses influences the signal-to-noise ratio of dopamine neurotransmission, which is hypothesized to determine the strength of reinforcement learning, the motivation to pursue reinforcers, and/or their incentive value[56]. We propose that the unique physiological profile of DFNZ, which promotes slow (tonic) dopamine over fast (phasic) dopamine responses, combined with its decreased efficacy at MOR–GAL1 heteromers, leads to a low dopamine signal-to-noise ratio, explaining both its weak reinforcement in rats and capacity to promote hyperlocomotion in mice.

Together, the lack of tolerance, lack of MOR density changes, weak withdrawal effects, immediate extinction of drug self-administration, lack of reinstatement and limited effects of DFNZ on NAc dopamine neurotransmission suggest that DFNZ has a low potential for abuse liability in humans. Supporting this idea, anecdotal reports from recreational users suggest that certain fluorinated nitazenes lack rewarding effects[58].

The unique pharmacokinetic and pharmacodynamic characteristics of DFNZ are likely to contribute to its superior safety profile. Indeed, factors such as brain penetrance, duration and magnitude of MOR activation, cellular signalling dynamics and receptor trafficking profiles are critical determinants of the analgesic efficacy and adverse effects of opioid agonists[1,7,59]. Given these cumulative and unique pharmacological properties, we propose that the ability of DFNZ to provide effective analgesia with minimal adverse effects is a consequence of its distinct pharmacology.

Opioid-induced analgesia is primarily mediated by central MORs, but peripheral MORs also contribute to pain relief[60]. Indeed, peripherally restricted MOR superagonists such as the strong and selective PGP substrate loperamide are safe and effective antidiarrheal agents, which

induce modest analgesic effects under some conditions[60,61]. Similar to loperamide, DFNZ is a PGP substrate with MOR superagonist effects. However, unlike loperamide, DFNZ produces strong analgesia, is a substrate for BCRP and is not peripherally restricted, but has impaired brain penetrance and limited brain MOR occupancy at therapeutic doses. To our knowledge, a MOR superagonist with limited brain MOR occupancy, such as DFNZ, has not been previously reported, but theoretical models suggest that such a compound could provide analgesia with a lower risk of adverse effects[62]. Nevertheless, it remains unclear from our study to what extent the analgesic effects of DFNZ are mediated by central versus peripheral mechanisms. Further studies using MOR antagonist derivatives that do not cross the blood–brain barrier can address this question.

We included both male and female rodents in most procedures, but one limitation of our study is that we did not include enough subjects to detect possible sex-dependent differences in some of the procedures. The extent to which MOR agonists, especially synthetic agonists, have sex-dependent effects on opioid reward, relapse and other behavioural effects is unclear[63]. Nonetheless, future studies should evaluate the contribution of sex to the unique behavioural and physiological effects of DFNZ.

DFNZ produced weak reinforcement in pain-naive rats, but we did not assess the effects of pain on DFNZ self-administration. With rare exceptions, prior studies have failed to show pain-induced increases in opioid self-administration[64–68]. Nevertheless, further studies are needed to determine whether pain exposure enhances DFNZ reinforcement and abuse liability.

In conclusion, our findings challenge the conventional dogma that high-efficacy MOR agonists are unsuitable for development as safe analgesics. Our results indicate that DFNZ, and potentially other nitazenes with similar safety profiles, warrant further investigation as potential therapeutic agents for pain. Additionally, given the favourable safety profile of DFNZ and its ability to reduce heroin self-administration, sustained-release formulations of DFNZ should be explored as novel opioid maintenance treatments. DFNZ may offer therapeutic benefits comparable to those of the full MOR agonist methadone, but without the associated risk of respiratory depression.

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

# Methods

## Binding screen

These experiments were performed by a commercial vendor (Eurofins) under contract with the National Institutes of Health (NIH). Membrane homogenates from stable cell lines or animal tissues expressing each protein were incubated with the respective radioligand in the absence or presence of FNZ or DFNZ (100 nM and 10 µM). In each experiment, the respective reference compound was tested concurrently with the test compound to assess the assay reliability. Nonspecific binding was determined in the presence of a specific agonist or antagonist at the target. Following incubation, the samples were filtered rapidly under vacuum through glass fibre filters presoaked in buffer and rinsed several times with an ice-cold buffer using a 48-sample or 96-sample cell harvester. The filters were counted for radioactivity in a scintillation counter using a scintillation cocktail.

## Animals

Sprague Dawley rats (6 to 8 weeks old) were purchased from Charles River. C57BL/6 J mice (8 weeks old) were obtained from Charles River or Jackson Laboratory. Adult TH-Cre mice (B6.Cg-7630403G23RikT g(Th-cre)1Tmd/J) (10 to 12 weeks old) were bred at the National Institute on Drug Abuse (NIDA). *Oprm1*-knockout mice (B6.129S2-*Oprm1*$^{tm1Kff/J}$) were obtained from Jackson Laboratory (strain 007559) (6 to 8 weeks old). PGP/BCRP-knockout mice (FVB.129P2-*Abcb1a*$^{tm1Bor}$*Abcb1b*$^{tm1Bor}$ *Abcg2*$^{tm1Ahs}$) (6 to 8 weeks old) were obtained from Taconic Biosciences. Animals were single or group housed in an environment with stable temperature (21–23 °C) and humidity (35–55%) and maintained on a 12 h:12 h light:dark reverse cycle with ad libitum access to food and water. Experiments and procedures complied with ethical regulations for animal testing and research, followed the NIH guidelines and were conducted according to the guidelines and approved by the Institutional Animal Care and Use Committees at the National Institute on Drug Abuse, the Chobanian and the Avedisian School of Medicine at Boston University and the University of Barcelona. Animals were randomly assigned to experimental groups and treatment conditions. Sample sizes were estimated based on experience from past work. The experimenters were blinded to the group allocation and treatment when applicable and as noted below.

## Competition binding assays

Dissected rat brains (minus cerebellum) were suspended in Tris-HCl 50 mM buffer supplemented with protease inhibitor cocktail (1:1,000) and disrupted with a Polytron homogenizer (Kinematica). Homogenates were centrifuged at 48,000$g$ (50 min, 4 °C) and washed twice in the same conditions to isolate the membrane fraction. Protein was quantified by the bicinchoninic acid method (Pierce). Membrane preparations were incubated with 50 mM Tris-HCl (pH 7.4) containing [$^3$H]DAMGO (~5 nM, 46 Ci mmol$^{-1}$, NIDA Drug Supply) and increasing concentrations (10 nM to 1 mM) of FNZ, DFNZ or DAMGO. Nonspecific binding was determined in the presence of 100 µM of unlabelled DAMGO. Free and membrane-bound radioligand were separated by rapid filtration of 500-µl aliquots in a 96-well plate harvester (PerkinElmer) and washed with 2 ml of ice-cold Tris-HCl buffer. Microscint-20 scintillation liquid (65 µl per well, PerkinElmer) was added to the filter plates, plates were incubated overnight at room temperature, and radioactivity counts were determined in a MicroBeta2 plate counter with an efficiency of 41%. One-site competition curves from three independent experiments with three replicates per experiment were fitted using GraphPad Prism 10 (GraphPad Software). $K_i$ values were calculated using the Cheng–Prusoff equation.

## Kinetic binding assays

**Association.** To each well was added 150 µl membrane preparation from cells expressing human MOR (hMOR) (Revvity, ES-542-M400UA) and 50 µl of naloxone (Tocris) or buffer (50 mM Tris, 10 mM MgCl$_2$, 0.1 mM EDTA (pH 7.4)). The plate was pre-incubated for approximately 30 min. [$^3$H]FNZ (~0.3 nM, 43 Ci mmol$^{-1}$, Novandi Chemistry) or [$^3$H]DFNZ (~0.3 nM, 54 Ci mmol$^{-1}$, Novandi Chemistry) solution (50 µl in buffer) was then added over fixed time intervals between 0–120 min.

**Disassociation.** To each well was added 150 µl hMOR membranes and 50 µl radioligand solution in buffer. In three wells, 50 µl of naloxone was added at the start of the assay to serve as nonspecific binding for all dissociation timepoints. The plate was then incubated for 90 min with gentle agitation to allow equilibrium to be reached. Dissociation was initiated by adding 50 µl of naloxone (10 µM final assay concentration) at various intervals between 0–120 min.

**Filtration.** All incubations were stopped by vacuum filtration onto presoaked (buffer with 0.1% polyethyleneimine (PEI)) GF/C filters using a 96-well FilterMate harvester, followed by 5 washes with ice-cold wash buffer. Filters were then dried under a warm air stream, sealed in polyethylene, scintillation cocktail added, and the radioactivity counted in a Wallac TriLux 1450 MicroBeta counter.

**Data analysis.** For each association and dissociation time point, nonspecific binding was subtracted from total binding to give specific binding. Data from two independent experiments with two replicates per experiment was fitted to standard ligand association and dissociation models using the nonlinear curve-fitting routines in GraphPad Prism 10. Dissociation curves were plotted initially to obtain $k_{off}$ and dissociation half-life ($t_{1/2}$). To plot the association curves, the $k_{off}$ calculated from the dissociation curves was used as an input parameter for the curve-fitting routine.

## [$^3$H]FNZ and [$^3$H]DFNZ autoradiography

Frozen rat or mouse brain tissue was sectioned (20 µM) on a cryostat (Leica) and thaw mounted on glass slides and stored at −80 °C until day of experiment. Slides were pre-incubated (10 min, room temperature) in incubation buffer (50 mM Tris-HCl, 1 mM MgCl$_2$, 5 mM KCl, 2 mM CaCl$_2$, pH 7.4), then incubated (60 min, room temperature) in incubation buffer containing 4 nM [$^3$H]FNZ (43 Ci mmol$^{-1}$) or 4 nM [$^3$H]DFNZ (54 Ci mmol$^{-1}$). Nonspecific binding was determined in the presence of 10 µM naloxone. The sections were then washed by 2× 30 s washes in their respective Tris buffers. Finally, slides were dipped in ice-cold distilled water to remove salts. The slides were exposed to the phosphor screen for 7 days and then imaged using a Typhoon biomolecular imager (Cytiva). The digitized images were calibrated using $^{14}$C standard slides (American Radiolabeled Chemicals).

## BRET assays

Plasmids used for G-protein dissociation assays include Gα$_{i1}$-RLuc8, Gα$_{oA}$-RLuc8, Gβ and Gγ-GFP2, each procured from the Trupath library[69] and MOR with an N-terminal haemagglutinin signal sequence and Flag tag. Plasmids used in β-arrestin recruitment assays include GRK2, MOR–Rluc8 and β-arrestin 1/2–mVenus. G-protein signalling data were determined by measuring the decrease in BRET signal following agonist-induced receptor activation and subsequent dissociation of Gα–RLuc8 from Gβ/Gγ–GFP2. Initially, suspension HEK293 cells (sourced from ATCC, untested for mycoplasma contamination) were grown in FreeStyle 293 Expression Medium (ThermoFisher Scientific) to a cell density of ~1.25 × 10$^6$ and transfected using PEI and an equimolar mixture of MOR, Gα-RLuc8, Gβ and Gγ-GFP2 plasmids for a total of 600 ng DNA per ml of cells. Transfected cells were grown for 48–72 h. Serial dilutions of agonist were prepared with assay buffer (1× HBSS, 0.1% BSA, and 6 mM MgCl2) and 30 µl was transferred to a 96-well opaque plate (Corning). 1 ml of transfected cells were then centrifuged for 1 min at 700$g$ and resuspended in 7 ml assay buffer

with 5 µg ml$^{-1}$ coelenterazine 400a (Cayman Chemical). The assay was initiated by adding 60 µl of cells into each well for a final volume of 90 µl and read for 15 min using a SpectraMax iD5 plate reader with emission wavelengths of 410/515 nm and an integration of time of 1 s. Similarly, the recruitment of β-arrestin 1 and 2 was monitored via BRET. RLuc8 and mVenus were fused to the C-terminus of MOR and N-terminus of β-arrestin 1/2, respectively. In this assay, BRET increases as β-arrestin is recruited to MOR. Otherwise, methods were identical to those used to measure G-protein dissociation, except cells were transfected with a 1:1:5 ratio of MOR-RLuc8:β-arrestin-mVenus:GRK2, and cells were resuspended with assay buffer containing coelenterazine h (Cayman Chemical) instead of coelenterazine 400a. Finally, the reaction was measured with emission wavelengths of 485/535 nm. Computed BRET ratios (GFP2 or mVenus/RLuc8) at 00:03:50, 00:07:36, and 00:11:23 (hh:mm:ss) were averaged and ligand-free well readings were subtracted from the ligand-treated control well readings for net BRET values. The $E_{max}$ value of DAMGO (MedChemExpress) was determined by measuring the net BRET tangent to the plateau of the dose–response curve. Each reading was then normalized to the DAMGO $E_{max}$ value by subtracting the min asymptote of the DAMGO positive control and dividing by the span of the DAMGO positive control (maximum asymptote–minimum asymptote). Normalized values from three independent experiments were then plotted as a function of concentration for each biological triplicate and fit with a three-parameter logistic nonlinear regression model with GraphPad Prism 10.

### Kinetic BRET assays

The BRET-based G$_o$ (Gα$_{oA}$) protein activation and β-arrestin 2 recruitment assays were performed as described previously[46,70]. In brief, G$_o$ protein activation assay uses *Renilla* luciferase 8 (Rluc8)-fused Gα$_{oA}$ and mVenus-fused Gγ2 as the BRET pair, co-transfected with MYC-tagged hMOR and Gβ1 in the µg ratio 5:1:4:5 (hMOR:Gα$_{oA}$:Gβ1:Gγ2). The β-arrestin 2 recruitment assay uses RLuc8-fused hMOR with a MYC tag and mVenus-fused β-arrestin 2 as the BRET pair, co-transfected with GRK2 in the µg ratio 0.5:6.5:5 (hMOR:β-arr2:GRK2)[46]. HEK293T cells (sourced from ATCC, tested for mycoplasma contamination) were transiently transfected with the above constructs using PEI at a ratio of 2:1 (PEI:total DNA by mass). After ~48 h of transfection, cells were washed, collected and resuspended in PBS + 0.1% glucose + 200 µM sodium bisulfite buffer. Two-hundred thousand cells were then transferred to each well of the 96-well plates (White Lumitrac 200, Greiner Bio-One) followed by addition of 1 µg µl$^{-1}$ coelenterazine H, a luciferase substrate for BRET. Three minutes after addition of coelenterazine H, ligands were added to each well. For kinetic experiments, cells were incubated at 25 °C within the PHERAstar FSX plate reader with BRET signal measurements collected in 2-min cycles from 2–46 min. BRET ratio was calculated as the ratio of mVenus (535 nm) over RLuc8 (475 nm) emission. Data were collected from at least 9–10 independent experiments performed in triplicate and normalized to maximal basal-subtracted BRET signal by DAMGO (Cayman Chemical) as 100%. Data analysis and statistical testing were performed using GraphPad Prism 10. In brief, we performed three-parameter dose–response regressions for all compounds at each time point. For every tested concentration, we calculated the net BRET ratio by subtracting the minimum response derived from the fitted raw BRET curve. Representative kinetics of maximum net BRET ratios for the tested compounds in both assays are shown in Extended Data Fig. 5. The maximum net BRET ratio for each compound was then normalized to that of DAMGO at the corresponding time point (reported as %$E_{max}$) by dividing each value by the span of the DAMGO positive control (maximum asymptote–minimum asymptote). Kinetics of the averaged pEC$_{50}$ and %$E_{max}$ values were subsequently plotted in Fig. 2. To evaluate whether the test compounds exhibited Go protein versus β-arrestin 2 signalling bias, bias factors were calculated using the $E_{max}$ – EC$_{50}$ method as follows.

To calculate the bias, we first defined a proxy transduction index (TI; distinct from the transduction coefficient) as:

$$TI = \log\left(\frac{\%E_{max}}{EC_{50}}\right)_{compound}$$

Then using the index of DAMGO as the reference, we derived bias as follows:

$$\Delta TI = TI_{compound} - TI_{DAMGO}$$
$$bias_x = \Delta TI_{G_o} - \Delta TI_{\beta arr2}$$

where $x$ indicates a specific time point.

The s.e.m. of TI was calculated as follows:

$$s.e.m. = \frac{\sigma}{\sqrt{n}}$$

Where $\sigma$ is the s.d. and $n$ is the number of experiments. Similar to a previous work[71], the estimated standard error (SE) for $\Delta TI$ for each compound in a given assay is calculated as:

$$SE_{assay} = \sqrt{(SEM_{compound})^2 + (SEM_{DAMGO})^2}$$

Standard error of bias between the G$_o$ and β-arrestin 2 pathways:

$$SE_{final} = \sqrt{(SE_{G_o})^2 + (SE_{\beta arr2})^2}$$

### Agonist-stimulated [$^{35}$S]GTPγS autoradiography

Frozen rat brain tissue was sectioned (20 µM) on a cryostat (Leica) and thaw mounted on glass slides. Preincubation buffer was pipetted onto each slide and incubated for 20 min at room temperature (50 mM Tris-HCl, 1 mM EDTA, 5 mM MgCl$_2$ and 100 mM NaCl). The buffer was removed via aspiration and incubated for 60 min in preincubation buffer containing 2.7 mM GDP and 1.3 µM DPCPX. GDP buffer was removed and [$^{35}$S]GTPγS cocktail (GDP buffer, 20 mM dithiothreitol (DTT), 300 nm [$^{35}$S]GTPγS) with agonists of interest (FNZ 10 µM; DAMGO 10 µM, DFNZ 10 µM), without agonists (basal condition), or with a saturated concentration of non-radioactive GTP (for nonspecific binding) was pipetted onto each slide and incubated for 90 min. The [$^{35}$S]GTPγS cocktail was removed via aspiration and slides were washed in ice-cold buffer (50 mM Tris-HCl, 5 mM MgCl$_2$, pH 7.4) for 5 min (2×) followed by a 30 sec dip in ice-cold water. Slides were apposed to a BAS-SR2040 phosphor screen (Cytiva) for 3 days and imaged using a Typhoon biomolecular imager (Cytiva). The digitized images were calibrated using $^{14}$C standard slides (American Radiolabeled Chemicals). Regions of interest (ROIs) were hand-drawn based on anatomical landmarks and radioactivity was quantified using Multigauge v.3.0 software (Fujifilm) and expressed as per cent binding of the basal condition.

### Cryo-electron microscopy

Pellets of mouse MOR were thawed and suspended in a hypotonic lysis buffer containing 20 mM HEPES pH 7.5, 1 mM EDTA pH 8.0, 1 mM MgCl$_2$, 100 µM TCEP, 10 µM naloxone, protease inhibitor cocktail and benzonase. The solution was stirred at 100 rpm at 4 °C for 1 h before centrifugation at 100,000g. Membrane pellets were then resuspended in 500 mM NaCl, 20 mM HEPES pH 7.5, 1 mM MgCl$_2$, 10% glycerol, 10 µM Naloxone, 100 µM TCEP, 1 mM benzamidine, protease inhibitor cocktail, and stock detergent was slowly added while stirring at 100 rpm at 4 C to a final concentration of 1% lauryl maltose neopentyl glycol (LMNG), 0.2% cholesteryl hemisuccinate (CHS), 0.2% cholate. Solubilization was then allowed to proceed for 3 h, with 2 mg ml$^{-1}$ iodoacetamide added at the 30 min and 1 h points. Solution was ultracentrifuged at 100,000g to remove insoluble material, the supernatant was supplemented with

20 mM imidazole, and the solution was gravity-loaded over Ni-NTA resin at 8 °C. The sample was washed with 10 column volumes of buffer containing 500 mM NaCl, 20 mM HEPES pH 7.5, 20 mM imidazole, 0.1% LMNG, 0.01% CHS, 10 μM naloxone, 100 μM TCEP; and eluted with a buffer containing 250 mM NaCl, 20 mM HEPES pH 7.5, 250 mM imidazole, 0.05% LMNG, 0.005% CHS, 10 μM naloxone, and 10% glycerol. The eluent was supplemented with 5 mM $CaCl_2$ and loaded onto M1 Flag resin, washed with buffer containing 20 mM HEPES pH 7.5, 250 mM NaCl, 2 mM $CaCl_2$, 0.01% LMNG, 0.001% CHS, 10 μM naloxone; and eluted with buffer containing 20 mM HEPES pH 7.5, 250 mM NaCl, 1 mM EDTA, 0.01% LMNG, 0.001% CHS, 10 μM naloxone, 10% glycerol, and 0.1 mg ml$^{-1}$ Flag peptide. The eluent was supplemented with 100 μM TCEP, concentrated, and loaded on to an Enrich 650 column for size-exclusion chromatography in buffer containing 100 mM NaCl, 20 mM HEPES pH 7.5, 0.01% LMNG, 0.001% CHS, and 100 μM TCEP. Fractions containing monomeric MOR were concentrated and used immediately for G-protein complexation. G protein and scFv16 were purified as described previously[72].

**Formation and purification of MOR–$G_i$ complex.** Purified MOR was incubated with 100 μM of FNZ for 1 h on ice before addition of a molar excess of purified G-protein heterotrimer and scFv16. Complexation was incubated for 1 h on ice before addition of 2 μl apyrase solution and incubation overnight on ice to allow complexation and GDP depletion to complete fully. Solution was then diluted with buffer containing 100 mM NaCl, 20 mM HEPES pH 7.5, 0.01% LMNG, 0.001% CHS, 5 mM $CaCl_2$ and 20–100 μM agonist; then loaded over M1 Flag resin. Resin was washed with buffer containing 100 mM NaCl, 20 mM HEPES pH 7.5, 0.005% LMNG, 0.0005% CHS, 1 $CaCl_2$, and 10–50 μM agonist, and eluted into a buffer containing 100 mM NaCl, 20 mM HEPES pH 7.5, 0.002% LMNG, 0.0002% CHS, 1 EDTA, 0.1 mg ml$^{-1}$ Flag peptide, 10% glycerol, and 20–100 μM agonist. Once eluted, 100 μM TCEP was added and complex was concentrated and then loaded onto a size-exclusion chromatography Enrich 650 column in buffer of 100 mM NaCl, 20 mM HEPES pH 7.5, 0.001% LMNG, 0.0001% CHS, 0.00033% glyco-diosgenin, 100 μM TCEP, 2 mM $MgCl_2$ and 1–100 μM agonist. Fractions containing complex were then concentrated to 10 mg ml$^{-1}$ for cryo-EM sample preparation and also evaluated by negative stain EM[73].

**Cryo-EM sample preparation and data collection.** All samples were prepared on glow-discharged holey gold grids with gold support (ultrAuFoil R1.2/1.3). Three microlitres of MOR–$G_i$ sample was loaded into an FEI Vitrobot Mark IV with the chamber held at 4 °C and 100% humidity. Vitrification was performed with blot time set to 3 s. Cryo-EM data were collected on a Titan Krios electron microscope at an accelerating voltage of 300 kV with the Smart EPU software v.2.1 and a Gatan K3 direct electron detector/bioquantum energy filter. Pixel sizes, frames, and electron doses for each dataset are provided in Extended Data Fig. 4.

**Data processing.** Tiff files were imported in Relion[74] for motion correction with MotionCorr2[75], CTF estimation with CTFFIND4[76], and template-based particle picking. Binned extracted particles were then imported into CryoSPARC[77] for 2D and 3D classification. Particle stacks were then refined with the nonuniform refinement routine and transferred back to Relion for re-extraction of unbinned particles. Particles were refined in Relion and subjected to Bayesian polishing. All particle stacks were then imported back into CryoSPARC for final nonuniform refinement and local refinement.

**Model building.** Initial models were based off prior cryo-EM structures of MOR–$G_i$ with PDB 6DDE[72]. Manual model building was performed in Coot v.0.9.8.1 EL[78] with refinement in Phenix[79]. GlideEM was used to validate the ligand poses with the higher resolution maps as well as the poses[80].

## APEX reaction, biotinylated protein enrichment and preparation for mass spectrometry analysis

HEK293T cells (sourced from ATCC, tested for mycoplasma contamination) expressing the APEX2 enzyme fused to the human MOR (MOR–APEX)[17] and the spatial reference APEX constructs (for plasma membrane, early endosome, late endosome/lysosome and cytoplasm as described previously[17,81]) were cultured in 6-well plate format and incubated with 500 μM biotin-phenol at 37 °C for 30 min. The receptor was activated with 10 μM DAMGO, 100 nM FNZ, and 100 nM DFNZ (all saturating concentrations) over a time course of 30 min. The spatial reference APEX samples were not treated with the ligands. APEX labelling was initiated pre-activation (time 0) and after 1, 5, 10 and 30 min of activation by 1:3 mixing of the $H_2O_2$-containing medium (1 mM $H_2O_2$ final) with the biotin-phenol containing medium at room temperature. After 45 s of the biotinylation reaction, the cells were washed 3 times (1 min each) with ice-cold quench buffer (PBS supplemented with 10 mM sodium ascorbate, 10 mM sodium azide, and 5 mM Trolox). Cells were then collected in 1 ml of quench buffer and pelleted by centrifugation at 4 °C for 10 min at 3,000g. For cell lysis, cells were homogenized using probe sonication in RIPA buffer (50 mM Tris, 150 mM NaCl, 1% Triton X-100, 0.25% sodium deoxycholate, 0.25% SDS, pH 7.4) supplemented with 10 mM sodium ascorbate, 10 mM sodium azide, 5 mM Trolox, 1 mM DTT and protease inhibitors (Roche Complete). To remove the cell debris, cell lysate was centrifuged at 10,000g for 10 min, and the supernatant was taken for streptavidin enrichment of biotinylated proteins. The enrichment of biotinylated proteins was automated with the KingFisher Flex (Thermo Fisher Scientific) as described previously[81]. In brief, supernatants were incubated at 4 °C for 18 h with 15 μl magnetic streptavidin beads (Pierce Streptavidin Magnetic Beads, Thermo Fisher Scientific) which were pre-washed twice with RIPA buffer. Following incubation, beads were washed three times with RIPA buffer, one time with 1 M KCl, one time with 0.1 M $Na_2CO_3$, one time with 2 M urea in 50 mM Tris-HCl (pH 8) buffer, and two times with 50 mM Tris-HCl (pH 8) buffer. Beads were maintained in 100 μl of 2 M urea in 50 mM Tris-HCl (pH 8) buffer for on-bead digestion of proteins. Samples were reduced with 5 mM TCEP (Tris(2-carboxyethyl)phosphine) at 37 °C for 30 min, followed by alkylation with 5 mM IAA (Iodoacetamide) at room temperature in the dark for another 30 min, which was quenched by addition of DTT (5 mM final). For tryptic digestion, 0.5 μg trypsin and 0.25 μg LysC was added to beads and incubated with shaking at 37 °C for 6 h. Supernatants were taken and saved for desalting using NEST C18 MicroSpin columns.

## Unbiased mass spectrometric data acquisition and protein quantification for APEX samples

MOR–APEX and spatial APEX reference samples were analysed on a TimsTOF HT mass spectrometry system (Bruker) coupled to a Vanquish Neo ultra high-pressure liquid chromatography (Thermo Fisher Scientific) interfaced via a CaptiveSpray2 nanoelectrospray source (Bruker). Samples were reconstituted in 0.1% formic acid and loaded onto a 15 cm × 75 μm internal diameter Aurora CSI column (IonOpticks). Mobile phase A consisted of 0.1% formic acid, and mobile phase B consisted of 0.1% formic acid/80% acetonitrile. Peptides were separated at a flow rate of 500 nl min$^{-1}$ using a nonlinear gradient increasing buffer B from 2% to 80% over 23 min, followed by a 7 min wash at 95% B and column equilibration. The mass spectrometer acquired data in dia-PASEF mode using a 75 ms TIMS ramp, a 1/$K_0$ range of 0.72–1.5, and an $m/z$ range of 100–1,700. One TIMS precursor MS frame was followed by 9 dia-PASEF MS$^2$ frames. Fragmentation windows had a mass width of 30 Da with an $m/z$ overlap of 1 Da and were collected over an $m/z$ range of 250–1,325 and a 1/$K_0$ range of 0.68–1.40. The DIA data were analysed with Spectronaut (Biognosys, v.19.7) using direct DIA analysis default parameters for the identification and quantification of proteins. Normalization in Spectronaut was turned off. Data were

searched against the Uniprot Human database (downloaded October 2024). Transition ion intensities from Spectronaut were summarized to protein intensities using the MSstats (v.4.10.1)[82] function dataProcess with default settings only high quality were used (featureSubset = "highQuality"). All proteins with only one quantified peptide were left out of further analysis.

## Statistical analysis of APEX mass spectrometry samples

Each protein's trend over the time course after DAMGO, FNZ and DFNZ treatment was scored by fitting the $\log_2$ intensities with a continuous cubic-polynomial curve over time using the R functions lm and poly. To better fit the rapid changes, especially between time 0 and 1 min, the collected timepoints were encoded by their ranks (1, 2, 3, 4 and 5 for 0, 1, 5, 10 and 30 min). The model included an additive term for the batch; a protein's background intensity was expected and allowed to vary between batches. The time-dependent model was compared with a null model that contained only the batch term using the R function anova to compute an $F$ statistic and $P$ value. The maximum mean change between time 0 and any single later time, after imputing any missing values using the fitted model, was used as the maximum log2 fold change for that protein. Proteins were considered significant interactors with the MOR–APEX construct using thresholds a maximum log2 fold change > log2 (1.5) over the time series and $P$ value ≤ 0.05 from the ANOVA test.

## Spatial coefficients calculation for MOR–APEX2 samples

To define location-specific proteins, proteins that varied between spatial references were scored using the MSstats[82] function groupComparison to compare between each non redundant pair of spatial references. The input to MSstats was the entire set of spatial references with the MOR–APEX2 data excluded. MSstats reports pairwise differences in means as $\log_2$FC, and a pairwise $P$ value calculated from a $t$-test assuming equal variance across all spatial references. A subset of 650 location-specific proteins was selected that could reliably distinguish locations by requiring $P$ value < 0.005 and $\log_2$FC > 1.0 and observed intensity in all three replicates of at least one spatial reference greater than the 50th percentile of all observed protein intensities. For each MOR–APEX2 sample (each ligand, timepoints, and replicate), coefficients were calculated to estimate the contribution of each spatial reference to the observed protein intensity as described previously[17]. First, protein intensities were scaled linearly between 0 and 1 by setting the maximum observed intensity (across all spatial reference and MOR–APEX2 samples) for each protein to 1.0, and all other observations were set to the ratio of observed/maximum for that protein. Missing values were set to zero. A matrix representing protein intensities in the spatial references for all observed proteins ($F$) was constructed using mean (per spatial reference) scaled intensity. The location-specific subset matrix ($S$) was extracted from $F$ by using only the rows of $F$ that match the 650 location-specific proteins. $S$ was then appended with three additional columns composed of randomly sampled values from $S$ to minimize the occurrence of low but nonzero location coefficients where they were expected to be zero. Location coefficients for each MOR–APEX2 sample were then calculated using the nonnegative least-squares procedure in the R package nnls using the location-specific matrix $S$ and the vector of location-specific protein scaled intensities from each MOR–APEX2 sample as inputs. We repeated this randomization and nnls procedure 1,000 times and used the median value for final spatial reference coefficients.

## Opioid-induced hyperlocomotion

Locomotor activity was recorded immediately after wild-type ($n = 12$ mice per drug, 6 male and 6 female) or *Oprm1*-knockout mice ($n = 8$ mice per drug, 4 male and 4 female) were introduced in the activity chambers in an open field arena (Opto-varimex ATM3, Columbus Instruments).

After habituation and saline injections, they were systemically administered varying doses of morphine (1, 3, 10, 30 or 100 mg kg$^{-1}$, intraperitoneally), FNZ (1, 3, 10, 30, 100 or 300 µg kg$^{-1}$, intraperitoneally) or DFNZ (0.03, 0.1, 0.3, 1, 3, 10 mg kg$^{-1}$) and locomotor activity was recorded for a 60 min period.

## Microsomal stability and metabolite identification studies

The phase 1 metabolic stability assay was performed in mouse liver microsomes as previously described[83,84]. In brief, the reactions were conducted in 200 mM potassium phosphate buffer, NADPH regenerating system (1.3 mM NADPH, 3.3 mM glucose-6-phosphate, 3.3 mM MgCl$_2$, 0.4 U ml$^{-1}$ glucose-6-phosphate dehydrogenase, 50 µM sodium citrate), and a final test compound concentration of 10 µM. Reactions were initiated with the addition of the mouse liver microsome (0.5 mg ml$^{-1}$). Each reaction, in triplicate, was terminated with cold methanol at predetermined time points (0 min, 30 min and 60 min). Compound disappearance was monitored via high-performance liquid chromatography with tandem mass spectrometry (LC–MS). Chromatographic analysis was performed on a Dionex ultra high-performance liquid chromatography system coupled with Q Exactive Focus orbitrap mass spectrometer (Thermo Fisher). Separation was achieved using an Agilent Eclipse Plus column (100 × 2.1 mm internal diameter; maintained at 35 °C) packed with a 1.8 µm C18 stationary phase. The mobile phase was composed of 0.1% formic acid in acetonitrile and 0.1% formic acid in water with gradient elution, starting at 2.5% organic phase (from 0 to 0.25 min) linearly increasing to 99% (from 0.25 to 1.25 min), and re-equilibrating to 2.5% by 4 min. The total run time for each analysis was 5 min. Pumps were operated at a flow rate of 0.4 ml min$^{-1}$. The mass spectrometer controlled by Xcalibur software 4.1.39.9 (Thermo Scientific) was operated with a HESI ion source in positive ionization mode for all compounds. Test compounds and metabolites were identified in the full scan mode (from $m/z$ 75 to 1,125). The percentage remaining was calculated by comparing $t = 0$ min samples with $t = 30$ min and $t = 60$ min samples. Metabolites identification was performed by comparing $t = 0$ min samples with $t = 60$ min samples, and structures were proposed based on accurate mass information.

## Efflux transporter panel and substrate assessment assays

These experiments were performed by a commercial vendor (Eurofins). For the efflux transporter panel (G346), membrane homogenates from stable cell lines were pre-incubated with FNZ or DFNZ followed by incubation with appropriate substrate and the presence or absence of an appropriate reference inhibitor. Results showing an inhibition >25% at a given efflux transporter are considered to represent significant effects of the test compounds and evidence that they interact with the given efflux transporter. For the PGP and BCRP substrate assessment assay (G357), permeability of FNZ and DFNZ was determined in the A–B and B–A directions with and without the addition of the PGP-specific inhibitor verapamil or the BCRP inhibitor KO143. For the MATE1 substrate assessment assay (5092), wild-type or MATE1-overexpressing HEK293 cells were incubated with FNZ or DFNZ for 2 min or 30 min with and without the MATE1 inhibitor pyrimethamine (5 µM). In each experiment, the respective reference compound was tested concurrently with the test compound to assess the assay reliability. Fluorescein was used as the cell monolayer integrity marker. Fluorescein permeability assessment (in the A–B direction at pH 7.4 on both sides) was performed after the permeability assay for the test compound. The cell monolayer that had a fluorescein permeability of less than $1.5 × 10^{-6}$ cm s$^{-1}$ for Caco-2 was considered intact, and the permeability result of the test compound from intact cell monolayer was reported.

## Equilibrium dialysis and plasma protein binding

Procedures were adapted from prior studies[85,86]. Undiluted rat plasma was spiked with 100 nM of [$^3$H]FNZ or [$^3$H]DFNZ in Tris-HCl and

incubated at 37 °C for 1 h. Samples were loaded onto an equilibrium dialysis device (RED device, Thermofisher) and incubated at 37 °C for 4 h on a vertical shaker at 750 rpm. Samples were then counted for radioactivity using a liquid scintillation counter and plasma protein binding levels were calculated.

## Analytical assays

Optima LC−MS-grade acetonitrile was acquired from Fisher Scientific. Ammonium formate was obtained from obtained from Honeywell. Methanol was obtained from Fisher Chemical. HPLC-grade water was obtained from Avantor. LC-MS-grade 0.1% formic acid in water was obtained from ThermoFisher. A master stock of 241 µM FNZ was prepared in methanol and stored at ≤−20 °C. Working stocks were prepared from the master stock and used to spike a calibration curve in blank plasma or tissue homogenate from 0.6−121 nM for plasma and 0.241−12.1 fmol mg$^{-1}$ for tissue. For plasma measurements extraction was performed by protein precipitation with 300 µl acetonitrile to 100 µl plasma thawed from ≤−70 °C. The samples were shaken for 3 min at 2,000 rpm with a ThermoMixer C (Eppendorf) and centrifuged at 10,000 rpm for 10 min at 4 °C in a Sorvall ST 40R centrifuge (ThermoFisher). Three-hundred microliters of the supernatant were collected and evaporated to dryness at 40 °C under a stream of nitrogen gas in a Microvap Nitrogen Evaporator (Organomation). The residue was reconstituted in 50 µl of 90:10 10 mM ammonium formate in 0.1% formic acid/methanol and shaken for 3 min at 2,000 rpm. Ten microliters were injected into the UHPLC−MS system. Frozen brain tissue was thawed from ≤−70 °C and weighed. Tissue was transferred to a homogenizing tube containing ceramic beads and 2 ml of water was added per gram of tissue. Each tissue sample was homogenized using a Bead Ruptor 12 homogenizer (Omni International) for 3 cycles of 30 s each on high. Two-hundred and fifty microlitres of tissue homogenate were protein precipitated with 1 ml acetonitrile, shaken for 3 min at 2,000 rpm, and centrifuged at 10,000 rpm for 10 min at 4 °C. One millilitre of the supernatant was collected and evaporated to dryness at 40 °C under a stream of nitrogen gas. The residue was reconstituted with 75 µl of 90:10 10 mM ammonium formate/methanol and shaken for 3 min at 2,000 rpm. Ten microliters were injected into the UHPLC−MS system. Samples exceeding the upper limit of quantification were further diluted to fall within the analytical measuring range. Separation analysis was performed using a Vanquish UHPLC system (ThermoFisher) with tandem Orbitrap Exploris 120 mass spectrometer (ThermoFisher). Reverse phase chromatography was performed using an Accucore biphenyl 2.1 × 50 mm, 2.6 µm particle size column (ThermoFisher) with a gradient flow at 0.4 ml min$^{-1}$, 10 mM ammonium formate in 0.1% formic acid as mobile phase A, and methanol as mobile phase B. The run started at 5% B for 0.5 min, increased to 95% B for 2.5 min, held at 95% B for 2 min, decreased to 5% B for 1 min, then held at 5% B for 1.5 min. Analysis was performed in positive ion mode with a full scan mass range of 300−500 $m/z$ and a mass accuracy of 5 ppm. Ionization was conducted using a heated electron spray ionization (HESI) source. XCalibur v.4.4.16.14 (ThermoFisher) software was used to integrate and report peak area for the M + H ion (415.2137 $m/z$) at a retention time of 3.93 min. GraphPad Prism 10 was used to plot and fit a standard curve and to interpolate unknown values.

## MOR occupancy

Rats ($n$ = 6 per drug, 3 male and 3 female) were injected subcutaneously with saline, FNZ (5 µg kg$^{-1}$), or DFNZ (0.3 or 1 mg kg$^{-1}$) 15 min before decapitation and tissue extraction. Brains were flash-frozen and stored at −80 °C until processed. Flash-frozen brains were separated by hemisphere. One hemisphere was used for assessing concentration of FNZ or DFNZ. The other was sectioned (20 µm) on a cryostat (Leica) and thaw mounted on ethanol cleaned glass slides. For [³H]DAMGO, 50 mM Tris-HCl buffer containing 5 nM [³H]DAMGO (46 Ci mmol$^{-1}$, NIDA Drug

Supply) was pipetted onto slides and allowed to incubate for 10 min at room temperature. For nonspecific binding cold DAMGO (10 µM) was also added. The sections were then washed by 2× 30 s washes in Tris buffer. Finally, slides were dipped in ice-cold distilled water to remove salts. The slides were dried and then exposed to the phosphor screen for seven days and then imaged using a phosphor imager (Typhoon FLA 7000; GE Healthcare). The digitized images were calibrated using ¹⁴C standard slides (American Radiolabeled Chemicals). ROIs were hand-drawn based on anatomical landmarks and radioactivity was quantified using Multigauge v.3.0 software (Fujifilm) and expressed as per cent specific binding of saline-injected rats.

## [¹⁸F]FNZ synthesis

**Production of [¹⁸F]fluoride.** ¹⁸O-enriched water (98%, Huayi Isotopes, approximately 2 ml) was loaded into a niobium body, high-yield [¹⁸F] fluoride target of a General Electric Medical Systems (GEMS) PETtrace cyclotron. The target was irradiated with a proton beam of 60 µA for 15−30 min to produce and average of 68.9 GBq (1.86 Ci) ($n$ = 6) of aqueous [¹⁸F]fluoride ion.

**[¹⁸F]fluoroethyl tosylate synthesis.** The radiosynthesis was performed using an in-house custom-made radiofluorination module (RFM) using LabView control software. Microwave heating was done using a CEM Corporation PETwave microwave (Matthews). The cyclotron produced [¹⁸F]fluoride ion was collected in a 5 ml V-vial (Wheaton) inside the hot cell and assayed in a dose calibrator to obtain the initial starting radioactivity. The [¹⁸F]fluoride ion was then remotely transferred to the RFM where it was trapped on a Chromafix 30-PS-HCO3 solid-phase extraction (SPE) cartridge (ABX) earlier preconditioned using 1 ml of high-purity water (Honeywell). The [¹⁸O]water was collected for recycling. The resin cartridge was eluted using 0.15 ml of a 1:1 acetonitrile:water mixture containing 18.1 µmol potassium carbonate and 31.9 µmol of Kryptofix K222 into an empty CEM 5 ml conical borosilicate glass reaction vessel. The resin cartridge was then rinsed with 0.250 ml of acetonitrile into the same reaction vessel and the [¹⁸F] fluoride was dried at 110 °C with nitrogen flow (325 ml min$^{-1}$) for 150 s in a standard thermal heating block. Then two separate additions of 0.25 ml acetonitrile were added with 150 s and 180 s drying, respectively. After drying, the reaction vessel was remotely moved and placed into a CEM Discovery PETwave microwave cavity and cooled to 50 °C using compressed air. Ethylene ditosylate, 4 mg (10.5 µmol) in 0.5 ml acetonitrile, was added to the vessel and the solution microwaved using a dynamic method at 50 watts for a total of 5 min with a temperature limit set to 85 °C. The reaction vessel was cooled to 50 °C followed by the addition of 0.5 ml of acetonitrile and the solution was transferred to an intermediate vessel for assay and a sample was analysed by analytical HPLC. [¹⁸F]fluoroethyl tosylate yield based on gradient analytical HPLC analysis was 58% ($n$ = 3).

**[¹⁸F]FNZ.** To a second CEM 5 ml conical borosilicate glass reaction vessel containing 4.5 mg (12 µmol) of etonitazene precursor, 5.5 mg (16.8 µmol) of cesium carbonate dissolved in 0.5 ml of DMF, 0.25 ml (~364 mCi) of the crude [¹⁸F]fluoroethyl tosylate solution was added. The vessel was microwaved using a dynamic method at 100 W for a total of 4 min with a temperature limit set to 120 °C. After cooling, the solution was diluted with 3 ml of water and purified by semi-preparative HPLC (Waters XBridge 10× 150 mm, 10 mm eluted with 18/82 acetonitrile/TEA Buffer pH 3.2 at 10 ml min$^{-1}$, [¹⁸F]fluoro-etonitazene retention 15.2 min). The product fraction was collected in 50 ml water, and then eluted through a Waters tC18 Sep Pak plus. The Sep Pak was washed with 10 ml of water and then eluted with 1 ml of ethanol. An average of 98 mCi of [¹⁸F]fluoro-etonitazene was produced from the starting aliquot of crude [¹⁸F]fluoroethyl tosylate. The total synthesis time was 67 min with an overall non decay corrected radiochemical yield 5.4% from starting [¹⁸F]fluoride. The average specific activity was 831.1 GBq µmol$^{-1}$

(22,461 mCi µmol$^{-1}$) at end of synthesis with an average radiochemical purity was 99.4%.

## Positron emission tomography

Male rats were anaesthetized with isoflurane and placed in a prone position on the scanner bed of a Mediso nanoScan PET/CT and injected intravenously with [$^{18}$F]FNZ (~0.2 µg kg$^{-1}$) and dynamic scanning commenced for 90 min. When indicated, animals were pretreated (~20 min before the injection of the PET radiotracer) with vehicle ($n$ = 3 rats) or naltrexone ($n$ = 3 rats) (10 mg kg$^{-1}$, intraperitoneally). The PET data were reconstructed and corrected for dead-time and radioactive decay. All qualitative and quantitative assessments of PET images were performed using the PMOD software environment (PMOD Technologies). The dynamic PET images were coregistered to MRI templates and time–activity curves were generated using PMOD's built-in atlases and the described analyses were performed. SUV was calculated as using the equation SUV = $C$/(dose/BW) where $C$ is the tissue concentration of [$^{18}$F]FNZ (kBq ml$^{-1}$), dose is the administered dose (MBq) and BW (kg) is the animal's body mass. Statistical analyses were performed using GraphPad Prism 10. Using the SUV, data were also expressed as region/cerebellum ratios.

## [$^3$H]FNZ and [$^3$H]DFNZ brain uptake and distribution

Male rats were injected intravenously with [$^3$H]FNZ (1 µg kg$^{-1}$) with ($n$ = 2) or without ($n$ = 2) subcutaneous naloxone pretreatment 5 min before (10 mg kg$^{-1}$) and euthanized 7 min following [$^3$H]FNZ administration. For mouse uptake, [$^3$H]DFNZ (100 µg kg$^{-1}$) was injected subcutaneously 30 min prior to euthanasia in wild-type ($n$ = 2 mice, 1 male and 1 female) or PGP/BCRP-knockout mice ($n$ = 2 mice, 1 male and 1 female). The brains were flash frozen in 2-methylbutane and stored at −80 °C until use. The blood was centrifuged (13,000 rpm, 10 min at room temperature) and serum was collected. Serum samples were dissolved in scintillation cocktail (2.5 ml) and radioactivity counts were determined using a liquid scintillation counter. The brains were sectioned (20 µm) on a cryostat (Leica), mounted into glass microscope slides, and air-dried overnight at room temperature. The day after slides were placed into a Hypercassette and covered by a BAS-TR2025 phosphor screen (Cytiva). The slides were exposed to the phosphor screen for 15 days and imaged using a phosphor imager (Typhoon, Cytiva). The digitized images were calibrated using $^{14}$C standard slides (American Radiolabeled Chemicals). ROIs were hand-drawn based on anatomical landmarks and radioactivity was quantified using Multigauge v.3.0 software (Fujifilm).

## Analgesia, catalepsy and hypothermia in rats

On the day of an experiment, male rats were brought into the laboratory in their home cages and allowed 1 h to acclimate. Groups of rats ($n$ = 5 per dose group) received subcutaneous injections of vehicle (1 ml kg$^{-1}$ saline), FNZ (1, 3, 10 or 30 µg kg$^{-1}$), or DFNZ (0.1, 0.3, 1 or 3 mg kg$^{-1}$) on the lower back between the hips and were returned to their home cages. Each rat was tested twice in separate experimental sessions, with at least three days of washout between experiments, and doses were randomly assigned. Pharmacodynamic endpoints including catalepsy score, body temperature, and hot plate latency, were determined prior to injection and at 15, 30, 60, 120 and 240 min post injection. At each time point, behaviour was observed for 1 min by an experienced rater, and catalepsy was scored based on three overt symptoms: immobility, flattened body posture, and splayed limbs. Each symptom was scored as either: 1, absent; or 2, present, and catalepsy scores at each time point were summed, yielding a minimum score of 3 and a maximum score of 6. Next, body temperature was measured using a hand-held reader sensitive to signals emitted by the surgically implanted transponder. We chose to examine hypothermia as a representative adverse effect of opioid treatment, since body temperature is a physiological measure that decreases in parallel with opioid-induced bradycardia and

respiratory depression. Finally, rats were placed on a hot plate analgesia meter (IITC Life Sciences) set at 52 °C. Rats remained on the hot plate until they exhibited hind paw licking in response to the heat stimulus and were then returned to their home cages. Time spent on the hot plate was recorded using a timer triggered by a foot pedal. A 45 s cut-off was employed to prevent tissue damage. Pharmacodynamic findings were analysed using GraphPad Prism 10. Raw time-course data for hot plate latency and catalepsy score were normalized to percent maximum possible effect (%MPE), using the following equation: (experimental measure − baseline measure)/(maximum possible response − baseline measure) × 100. The maximum possible response for hot plate latency was 45 s, whereas the maximal response for catalepsy score was 6. Raw time-course data for body temperature were normalized to change from baseline, Δ temperature in °C, for each rat. Normalized time-course data were analysed by two-factor (dose × time) ANOVA followed by Tukey post hoc test to determine effects of drug doses at each time point. Mean hot plate latency and catalepsy score over the first 60 min were used to construct dose–response relationships, which were analysed by nonlinear regression (response stimulation, normalized response) to determine ED$_{50}$ (potency) values.

## Antinociceptive assays in mice

**Hot plate.** *Oprm1*-knockout mice were brought into the laboratory in their home cages and allowed 1 h to acclimate. Male and female mice received subcutaneous injections of vehicle (1 ml kg$^{-1}$ saline), FNZ (0.1 mg kg$^{-1}$), or DFNZ (3 mg kg$^{-1}$) and were returned to their home cages. Each mouse was tested three times in separate experimental sessions, with at least 3 days of washout between experiments, and doses were randomly assigned. Hot plate latency was determined prior to injection and at 15, 30, 45 and 60 min post injection. Mice were placed on a hot plate analgesia meter (IITC Life Sciences) set at 54 °C. Mice remained on the hot plate until they exhibited hind paw licking, shaking or jumping in response to the heat stimulus and were then returned to their home cages. Time spent on the hot plate was recorded using a timer triggered by a foot pedal. A 20 s cut-off was employed to prevent tissue damage. Pharmacodynamic findings were analysed using GraphPad Prism 10. Raw time-course data for hot plate latency were normalized to percent maximum possible effect (%MPE), using the following equation: (experimental measure − baseline measure)/(maximum possible response − baseline measure) × 100. The maximum possible response for hot plate latency was 20 s.

**Complete Freund's adjuvant model.** CFA (Sigma Aldrich) was diluted in a 1:1 ratio in saline and 30 µl were injected in the left hind paw of the mouse[87].

**von Frey assay.** To monitor mechanical allodynia, we used von Frey filaments with ascending forces expressed in grams[88] (Stoelting). Filaments were applied five times in a row against the mid-plantar area of the left hind paw, with all mice receiving a filament application before returning for the next application to the first mouse. Hind paw withdrawal or licking was marked as a positive allodynia response. The second force in which we observed a positive response in three out of five repetitive stimuli was defined as the allodynia threshold. Mechanical allodynia was measured 1 h after drug or saline administration.

**Hargreaves assay for thermal hyperalgesia.** Male mice were placed in Plexiglas boxes on top of a glass surface (IITC Life Sciences). The latency of withdrawal of the left injured paw was measured after a high intensity heat beam was applied to the mid-plantar area[89] (IITC Life Sciences). Three measurements were obtained, and the average was defined as the thermal nociceptive threshold. We used intensity level of 40 and a cut-off time of 20 s. Hind paw withdrawal or licking was marked as a positive allodynia response. Hargreaves assay was performed 1 h after drug or saline administration.

## AAV injection

Mice were anaesthetized with isoflurane (5% for induction and 1.5% for maintenance) for the duration of the injection and placed in a stereotaxic apparatus for head fixation. A small hole was drilled into the skull above the targeted coordinates for virus injection and fibre implant. After virus infusion, the syringe was left in place for 5 min and then slowly removed. Fibre optic studs (400 μm core diameter and 0.5 NA) (RWD Life Science) were placed in the NAc shell and fixed with dental cement (Dentalon, Kulzer). Three weeks after the surgery the fibre photometry experiments were performed.

**Dopamine dynamics.** Mice were injected unilaterally in the shell of the nucleus accumbens based on the Paxinos atlas stereotaxic coordinates (anterior–posterior (AP): 1.1, medial–lateral (ML): 0.7, from bregma dorsal–ventral (DV): −4.2 from dura mater) with 500 nl of dLight1.3b [ssAAV-1/2-hSyn1-chI-dLight1.3b-WPRE-bGHp(A)] at $7.6 \times 10^{12}$ vg ml$^{-1}$ using a 2 μl Hamilton syringe (Hamilton Neuros) driven by a syringe pump (Stoelting) at a flow rate of 50 nl min$^{-1}$.

**Axon-GCaMP6s dynamics.** TH-cre mice were unilaterally injected in the ventral tegmental area (coordinates: AP: −3.2 ML: −0.5 DV: −4.4) with 500 nl of AAV9-hSynapsin1-FLEx-axon-GCaMP6s at $5 \times 10^{12}$ vg ml$^{-1}$ using a 2 μl Hamilton syringe. Optic fibre was implanted in the NAc (coordinates: AP: 1.3 ML: −0.9 DV: −4.5).

## Fibre photometry

**dLight 1.3b.** After allowing three weeks for surgery recovery and biosensor expression, fibre photometry experiments were performed in awake male mice as previously described[55]. Extracellular dopamine was recorded using a Fiber Photometry Console (Doric) and Neuroscience Studio V6 (Doric) and signal was measured as changes in fluorescence emission after excitation with a light source at 470 nm. Isosbestic signal, excited with 405 nm, was recorded for artefact correction and signal decay. During dopamine recordings, animals explored a 20 × 15 cm cage, resembling their home cage, for 10 min followed by saline, FNZ (3 or 100 μg kg$^{-1}$), DFNZ (0.3, 1 or 3 mg kg$^{-1}$), morphine (40 mg kg$^{-1}$) or fentanyl (0.3 mg kg$^{-1}$) (all drugs were prepared in 0.9% NaCl at 10 ml kg$^{-1}$ of body weight) subcutaneous administration after which animals were placed back in the cage and recorded for another 30 min. Mice received drugs in randomized order in a counterbalanced design. Animal behaviour was recorded using a webcam (Logitech) in synchrony with the photometry signal. Photometry data were analysed using custom MATLAB scripts. Data were downsampled (10×) and low pass (10 Hz) filtered. Using a polynomial fit, the isosbestic signal was rescaled to the peak-free biosensor signal obtained via symmetric least-squares filtering of the baseline period. Then, the biosensor signals were corrected using the formula d$F/F = (F − F_0/F_0)$, where $F$ is the fluorescence of the signal at a given time point and $F_0$ is the corresponding rescaled isosbestic signal. To identify transients, a prominence over baseline threshold was set for every animal on a baseline recording and it remained constant across all experimental conditions for that animal. Then the signal was analysed to isolate and quantify the properties (amplitude, duration, frequency) of the spontaneous fluorescence transients. To measure slow changes in dopamine concentration, data were further low pass filtered using a Butterworth filter at 0.1 Hz and the traces from all animals were aligned and averaged and area under the curve was quantified in 5-min intervals. Following photometry experiments, animals were sacrificed, their brains dissected and imaged using a Zeiss Apotome 3 microscope to verify virus expression and correct fibre placement. Animals without virus expression or incorrect fibre location were excluded.

**Axon-GCaMP6s.** Signals were recording using a TDT RZ10 system (Tucker-Davis Technologies) controlled by Synapse Software v.95-44132P. Excitation was delivered at 465 nm and 405 nm isosbestic reference signal was recorded for artefact and decay correction. Male mice were allowed to freely explore a 45 × 45 cm open field for 10 min (baseline). After this the mice received intraperitoneal injections of FNZ (0.03 or 0.1 mg kg$^{-1}$), DFNZ (0.3, 1 or 3 mg kg$^{-1}$), fentanyl (0.3 mg kg$^{-1}$) or morphine (40 mg kg$^{-1}$). All drugs were prepared in 0.9% NaCl and administered at 10 ml kg$^{-1}$. Following injection, mice were returned to the chamber and recorded for 60 additional minutes. Photometry data were analysed using custom MATLAB scripts. d$F/F$ was calculated by scaling the baseline of the 405 nm isosbestic signal to the baseline of the 465 nm calcium signal using a polynomial fit and then using the formula d$F/F = (F − F_0/F_0)$, where $F$ is the fluorescence signal and $F_0$ is the fitted isosbestic reference. Calcium transients were identified using an event prominence threshold determined from baseline recordings for each animal and held constant across all drug conditions. Event amplitude, duration, and frequency were quantified. For slow signal changes, $\Delta F/F$ traces were aligned to the time of drug administration and the area under the curve was measured over 5 min time windows.

## Microdialysis

Procedures were as described previously[46,47]. Prior to surgery, male rats were injected with meloxicam (1 mg kg$^{-1}$, subcutaneously), then anaesthetized with a mixture of ketamine/xylazine (80 mg kg$^{-1}$ intraperitoneally and 10 mg kg$^{-1}$ intraperitoneally, respectively) and implanted unilaterally into the VTA (coordinates from bregma with a 10° angle in the coronal plane; anterior: −5.7 mm; lateral: −2.4 mm; vertical: −9 mm) with a regular microdialysis probe or with a specially designed microdialysis probe that allows the direct infusion of large peptides within the sampling area[46,47]. After surgery, the rats were allowed to recover in freely rotating hemispherical plastic bowls equipped with overhead fluid swivels. Twenty hours after probe implantation, experiments were performed on freely moving rats in the same hemispherical cages in which they recovered overnight from surgery. An artificial cerebrospinal fluid (ACSF) solution containing 144 mM NaCl, 4.8 mM KCl, 1.7 mM CaCl$_2$, and 1.2 mM MgCl$_2$ in ultrapure water was pumped through the probe at a constant rate of 1.25 μl min$^{-1}$. After washout period of 90 min, dialysate samples were collected at 20-min intervals. For peptide infusion, transmembrane peptides with the amino acid sequence of TM5 and TM7 of the MOR were dissolved in 0.1% DMSO in ACSF to a final concentration of 60 μM. Both peptides were injected with a 10-μl syringe (Hamilton) driven by an infusion pump and coupled with silica tubing (73-μm inner diameter; Polymicro Technologies) to the microdialysis probe/infusion cannula (dead volume, 40 nl), which was primed with ACSF and plugged during implantation. Each peptide was delivered at a rate of 15 nl min$^{-1}$ starting 20 min before an intraperitoneal dose of DFNZ (1 mg kg$^{-1}$). Two other groups received either FNZ (0.1 mg kg$^{-1}$) or DFNZ (1 mg kg$^{-1}$) without transmembrane peptide infusion. At the end of the experiment, rats were given an overdose of pentobarbital-based euthanasia solution, the brains were extracted and fixed in formaldehyde, and probe placement was histologically verified. Dopamine content was measured by HPLC coupled with a coulometric detector (5200a Coulochem III; ESA).

## Self-administration

**Surgery.** Procedures were as described previously[90]. Male and female rats were anaesthetized with isoflurane. Catheters were made from Silastic tubing attached to a modified 22-gauge cannula (Plastics One, C313G-5up) and cemented to polypropylene mesh (Elko Filtering, 05–1000/45 or Industrial Netting, XN3019-47.5). The catheter was inserted into the jugular vein, and the mesh was fixed to the mid-scapular region of the rat. Rats were injected subcutaneously with ketoprofen (2.5 mg kg$^{-1}$, subcutaneously, Covetrus) at the beginning of surgery or with carprofen (2.5 mg kg$^{-1}$, Norbrook) after surgery and on the following day. Rats recovered for 6 days before training. Catheters were flushed daily with gentamicin (4.25 mg ml$^{-1}$, Fresenius Kabi, 1002)

dissolved in sterile saline. If we suspected catheter failure, we tested patency by intravenous infusion of a short-acting barbiturate anaesthetic Brevital (methohexital sodium, 10 mg ml$^{-1}$ in buffered saline, 0.1–0.2 ml injection volume) or Diprivan (propofol, NIDA pharmacy, 10 mg ml$^{-1}$, 0.1–0.2 ml injection volume, intravenously). Rats were not food- nor water-restricted throughout this experiment.

**Apparatus.** We used Med Associates chambers with MED-PC v.4.2 software with two levers located ~8 cm above the grid floor. Lever presses on the active, retractable lever activated the infusion pump, whereas lever presses on the inactive, retractable lever had no consequences. Each session began with the illumination of a house light that remained on for the entire session. The active and inactive levers were inserted into the chamber 10 s after the house light was illuminated. During the self-administration sessions, a fixed ratio 1 (FR1), FR3 or progressive ratio schedules were used. Each infusion was paired with a 20-s white-light cue. A 20-s timeout followed infusions before subsequent responses resulted in another infusion.

**Effect of opioids on heroin and food.** Rats were not food- nor water-restricted throughout these experiments. The first goal was to examine how different subcutaneous doses of FNZ (0–17 µg kg$^{-1}$), DFNZ (0–1 mg kg$^{-1}$) or fentanyl (0–0.1 mg kg$^{-1}$) affect the intake of heroin in rats with chronic history of heroin IVSA. The second goal was to assess how FNZ, DFNZ and fentanyl at these same doses affect the self-administration of food pellets (TestDiet, 1811155, 12.7% fat, 66.7% carbohydrate, and 20.6% protein). Rats were implanted with jugular vein catheters as described previously[90] and trained to self-administer heroin or food. Saline, FNZ, DFNZ or fentanyl were injected immediately prior to self-administration testing and behavioural responses were recorded for up to 180 min.

**Opioid self-administration training.** Male and female rats were trained to self-administer heroin (100 µg kg$^{-1}$), FNZ (1 µg kg$^{-1}$), or DFNZ (30 µg kg$^{-1}$) intravenously for 3 h per day. Drug was infused over 3.5 s. Responses on the active lever during the timeout period and the inactive lever were recorded but had no consequences. Rats were considered trained if they pressed the active lever 80% of all presses with <20% variation in infusions per session over consecutive daily sessions.

**Dose response.** Rats were placed on a multiple-dose schedule (3 h per day for 6–9 days) with their FR3 or progressive ratio schedules. All parameters remained the same as FR3 training, except for drug dose. The number of infusions, and lever responses were recorded for each session.

**Extinction and drug-induced reinstatement.** Rats were retrained on heroin (100 µg kg$^{-1}$ per infusion), FNZ (1 µg kg$^{-1}$ per infusion) or DFNZ (30 µg kg$^{-1}$ per infusion) at FR3 for one session, followed by 8 days of saline self-administration with all cues present. Rats were then injected intravenously with heroin (100 µg kg$^{-1}$), FNZ (1 µg kg$^{-1}$) or DFNZ (30 µg kg$^{-1}$) immediately prior to a saline self-administration session.

## Electrochemical oxygen measurements

Male rats were implanted with an oxygen sensor in NAc and equipped with an intravenous jugular vein catheter, allowing for direct assessment of opioid-induced brain hypoxia with high selectivity and temporal precision[91,92]. Using this procedure, we previously showed that intravenous fentanyl had over tenfold greater potency to promote brain hypoxia compared with intraperitoneal fentanyl[30]. Experiments began 4–5 days after the surgeries and continued over several daily sessions (3–5). FNZ (0.01, 0.03 mg kg$^{-1}$) (n = 3 rats, 14 independent experiments), DFNZ (0.1, 0.3 mg kg$^{-1}$) (n = 2 rats, 14 independent experiments) and fentanyl (0.03 mg kg$^{-1}$) (n = 3 rats, 8 independent experiments) were delivered to rats via a slow, stress-free iv injection via the catheter

extension. In a second cohort of rats, tariquidar (0.1, 1 mg kg$^{-1}$) (n = 2 rats, 13 independent experiments) was injected 10 min prior to DFNZ (0.1 mg kg$^{-1}$) (n = 2 rats, 7 independent experiments). In a third cohort, rats were injected subcutaneously with saline (n = 3 rats, 9 independent experiments) or 1 mg kg$^{-1}$ DFNZ (n = 3 rats, 9 independent experiments). Data were calculated as changes in oxygen levels relative to pre-injection baseline (100%).

## Tolerance and mechanical hypersensitivity using von Frey assay

The purpose of this experiment was to assess the degree to which FNZ, DFNZ and fentanyl produce changes in mechanical sensitivity characteristic of tolerance and allodynia or hyperalgesia. Procedures were adopted from a prior study and were as previously described[33] with modifications. Naive male and female rats were handled and habituated to chambers for 15–30 min per day and received saline subcutaneous injections in the week prior to testing. Rats received daily subcutaneous escalating drug doses for 4 weeks (5 days per week). Rats were tested each week after drug administration (saline, FNZ (0.003–0.03 mg kg$^{-1}$), DFNZ (0.1–1 mg kg$^{-1}$) or fentanyl (0.01–0.03 mg kg$^{-1}$)). von Frey responses were tested 30 min post injection (tolerance) and then again 4–6 h post injection (mechanical hypersensitivity). For both, responses were measured 6 times per time point, alternating between hind paws with at least 5 min intervals between acquisitions. Rats were habituated to the testing room for at least 30 min and acclimatized to the testing apparatus for at least 15 min before testing.

## Naloxone-precipitated withdrawal

**Induction of opioid dependency.** Procedures were adapted from previous studies[32,93,94]. Morphine was administered subcutaneously to male and female rats once daily at 10:00 for 7 days via an escalating dose schedule: 20 mg kg$^{-1}$ (day 1), 40 mg kg$^{-1}$ (day 2/3), 60 mg kg$^{-1}$ (day 4/5) and 80 mg kg$^{-1}$ (days 6/7). Owing to their faster pharmacokinetics, FNZ (30 µg kg$^{-1}$) and DFNZ (1 mg kg$^{-1}$) were administered subcutaneously to male and female rats at 10:00 and 16:00 for 7 days.

**Naloxone-precipitated withdrawal.** On day 8, rats received the morning opioid administration. Four hours later they were injected intraperitoneally with 1 mg kg$^{-1}$ naloxone and placed into a $40 \times 40 \times 30$ cm plexiglass box to record somatic signs of withdrawal syndrome for 10 min. Rats were weighed twice: immediately prior to naloxone injection and 1 h later. Videos were scored by two blind experimenters as described previously[33,95,96]. Graded signs included jump attempts (1 point for 1–4 attempts, 2 points for 5–9 attempts, 3 points for >10 attempts), paw tremor (2 points for 1–2, 4 points for >3), wet dog shakes (1 point for 1–2, 2 points for >3) and fecal deposits (1 point per deposit). Checked signs included abdominal spasms (2 points), abnormal posture (3 points), diarrhea (2 points), irritability/vocalization (3 points), genital grooming (3 points), profuse salivation (7 points), ptosis (2 points), swallowing movements (2 points) and teeth chattering (2 points). Weight loss following naloxone administration was also considered with 1 point per gram lost. These scores were averaged across the two experimenters and summed to produce an overall withdrawal score.

## NanoBiT assays

Human MOR and Gα$_o$ cDNA were cloned in the pIREShyg3 plasmid vector within the Afe I and Xba I restriction enzyme sites (Clontech Laboratories) containing the sequences for the small subunit (SmBiT) and long subunit (LgBiT) of nanoluciferase[97] respectively (pIRES-HA-PS-MORSmBiT, pIRES-HA- GαoLgBiT). MOR included the mGlu5 receptor signal peptide in 5′ of the multiple cloning site to allow for plasma membrane trafficking. All constructs were haemagglutinin tagged for detection and verified by DNA sequencing.

HEK293T cells (sourced from ATCC, tested for mycoplasma contamination) (CRL-321, RRID: CVCL_0063) and grown in Dulbecco's modified Eagle medium (DMEM) supplemented with 10% (v/v) fetal bovine serum,

100 U ml$^{-1}$ penicillin, 100 μg ml$^{-1}$ streptomycin, and 2 mM L-glutamine. Cells were cultured in CytoOne (USA Scientific) treated culture plates in a Forma Series II Water Jacket incubator (Thermo Fisher Scientific) at 37 °C, 5% $CO_2$, 90% humidity. For $G\alpha_o$ engagement assays, HEK293T cells were plated in a 24-well plate and transiently transfected with 0.25 μg pIRES-HA-PS-MORSmBiT and pIRES-HA- GαoLgBiT together with 1.0 μg pIRES-HA-PS-hGalR or pcDNA3.1 using PEI transfection reagent[98]. Cell medium was replaced with fresh DMEM after 4 h and incubated overnight. The next day, cells were trypsinized, replated to 96-well poly-D-lysine precoated plates, and allowed to incubate overnight. The following day, medium was replaced with 0.1% glucose/2 mM $NaHSO_3$/PBS with various concentrations of FNZ, DFNZ or (S)-methadone and incubated for 15 min at 37 °C. Coelenterazine H (MedLumine) was subsequently added in a final concentration of 5 μM and plates were incubated at 37 °C for 5 min. Luminescence was measured at 485 nm in a Mithras LB 940 plate reader (Berthold Technologies). Results are expressed as percentage of the maximal recorded luminescence of cells transfected with MOR alone (RLU % ctrl).

## Statistics

All data were analysed using GraphPad Prism 10, unless otherwise noted, using one-way, two-way, mixed-effects or non-parametric ANOVA, two-sided independent $t$-tests, two-sided paired-sample $t$-tests, taking repeated measures into account where appropriate. Post hoc tests were performed two-tailed and using the appropriate multiple comparison correction.

## Reporting summary

Further information on research design is available in the Nature Portfolio Reporting Summary linked to this article.

## Data availability

The data supporting the findings of this study are available within the paper and its supplementary information files. The cryo-EM density map has been deposited in the Electron Microscopy Data Bank under accession codes EMD-70069 (global refinement), EMD-70070 (local refinement, receptor) and EMD-70071 (composite map) and the coordinates have been deposited in the Protein Data Bank under accession number 9O36. The mass spectrometry proteomics data have been deposited to the ProteomeXchange Consortium via the PRIDE (https://www.ebi.ac.uk/pride/) partner repository[99] with the dataset identifier PXD062863. Reviewers can access these data using the project accession PXD062863 and reviewer token 62GfWQNbF6w3. Alternatively, these data may also be accessed by logging into the PRIDE website using the reviewer username reviewer_pxd062863@ebi.ac.uk and password mmgadjWUcRls. Should any data that are not included in the paper be needed, they are available from the corresponding author upon reasonable request. Source data are provided with this paper.

## Code availability

Analysis code is available at GitHub (https://github.com/wdunne3/Calcium-Fiber-Photometry-Analysis/ and https://github.com/BonaventuraLab/fiber-photometry).

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

**Acknowledgements** This work was supported by the National Institute on Drug Abuse (NIDA) Intramural Research Program (ZIA-DA000069 (M.M.), ZIA-DA000606 (L.S.)), NIDA grant R01DA056354 (R.H.), the Spanish Ministerio de Ciencia e Innovación/Agencia Estatal de Investigación MICIU/AEI/10.13039/501100011033 and 'ESF Investing in Your Future' grants RYC-2019-027371-I (J.B.) and PID2023-147013OB-I00 (J.B.), the Ministerio de Sanidad/Plan Nacional Sobre Drogas grant 20211070 (J.B.), and the St. Jude Children's Research Hospital Collaborative Research Consortium on G-protein-coupled receptors (GPCR) (G.S.). We are grateful for support from the NIDA Medication Development Program (G. Tanda, and A. Newman), the NIDA Translational and Analytical Core (S. Jackson, and L. Kryszak), the NIDA Drug Design and Synthesis Section (S. Hubbard) and the NIDA Addiction Treatment Discovery Program (ATDP) (D. White). We thank M. Raley for providing illustrations. This research was supported in part by the Intramural Research Program of the NIH. The contributions of the NIH authors are considered works of the US government. The findings and conclusions presented in this paper are those of the authors and do not necessarily reflect the views of the NIH or the US Department of Health and Human Services.

**Author contributions** Conceptualization: K.C.R. and M.M. Methodology: J.L.G., E.N.V., Z.J.F., A.S., M.J.R., M.D.S., R.C.B., I.M.G., T.X., O.S., A.E.T., J.M.B., K.E.C., H.B., A.E., Z.M.G.-P., S.C., M.R.N., F.L., G.C.G., M.R., A.A.M., D.R.B., G.E., W.D., C.Q., I.S., C.B.L, R.R., D.P.H., R.F.D., L.S., R.H., S.F., E.K., J.B., Y.S., V.Z., M.H.B., G.S., K.C.R. and M.M. Investigation: J.L.G., E.N.V., Z.J.F., A.S., M.J.R., M.D.S., R.C.B., I.M.G., T.X., O.S., A.E.T., J.M.B., K.E.C., H.B., A.E., Z.M.G.-P., S.C., M.R.N., F.L., A.A., G.C.G., M.R., L.C., A.A.M., D.R.B., G.E., W.D., C.Q., I.S., C.B.L., R.R., D.P.H., G.S., E.K. and M.H.B. Funding acquisition: L.S., R.H., S.F., E.K., J.B., Y.S., V.Z., M.H.B., G.S., K.C.R. and M.M. Project administration: L.S., R.H., S.F., E.K., J.B., Y.S., V.Z., M.H.B., G.S., K.C.R. and M.M.

Supervision: L.S., R.H., S.F., R.R., R.F.D., E.K., J.B., Y.S., V.Z., M.H.B., G.S., K.C.R. and M.M. Writing, original draft: M.M. Writing, review and editing: all co-authors.

**Competing interests** J.L.G., A.S., M.H.B., K.C.R. and M.M. are listed as co-inventors on a patent application (WO2024196438A1) for the use of novel fluorinated etonitazene analogues in anaesthesia, pain and substance use disorders. J.L.G., A.S., M.H.B., K.C.R. and M.M. have assigned any potential patent rights to the US government but will share a percentage of any royalties that may be received by the government. G.S. is a co-founder of and consultant for Deep Apple Therapeutics.

**Additional information**
**Correspondence and requests for materials** should be addressed to Georgios Skiniotis, Kenner C. Rice or Michael Michaelides.

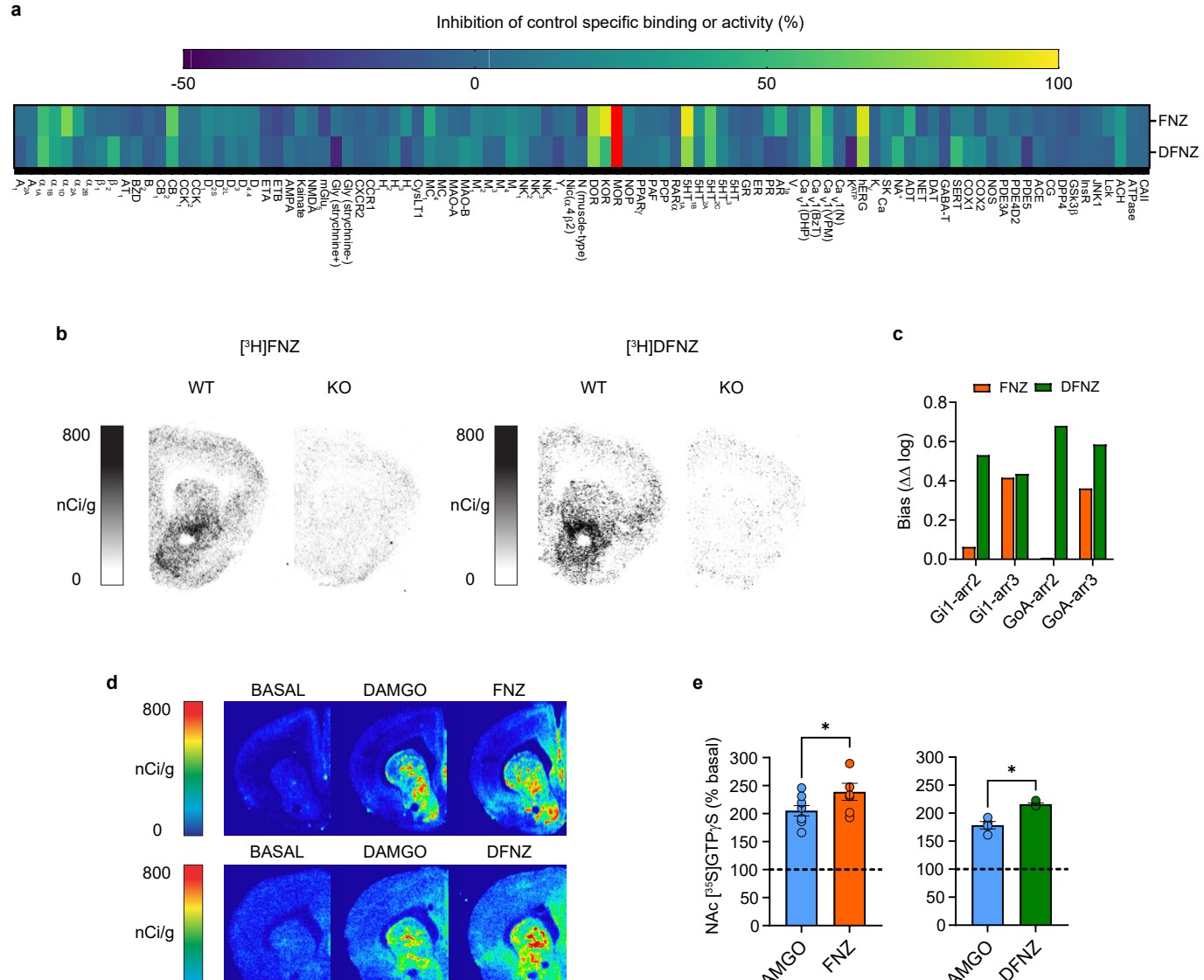

**Extended Data Fig. 1 | Selectivity and G protein activation.** (**a**) Competitive binding screens of FNZ and DFNZ at 10 μM across a panel of various receptors, enzymes and transporters. (**b**) [³H]FNZ and [³H]DFNZ autoradiography in C57Bl/6 J wild type (WT) (n = 1 mouse) and *Oprm1* knockout (KO) mice (n = 2 mice). (**c**) Bias calculations for functional BRET assays shown in Fig. 1. (**d**, **e**) [³⁵S] GTPγS autoradiography of DAMGO-, FNZ- and DFNZ-induced G protein

activation in the rat nucleus accumbens (NAc) showing that FNZ (paired t-test; two-sided, p = 0.02) (n = 4 rats over 8 independent experiments) and DFNZ (paired t-test; two-sided, p = 0.01) (n = 4 rats over 4 independent experiments) induce significantly greater [³⁵S]GTPγS binding than DAMGO. Data shown as mean ± SEM.

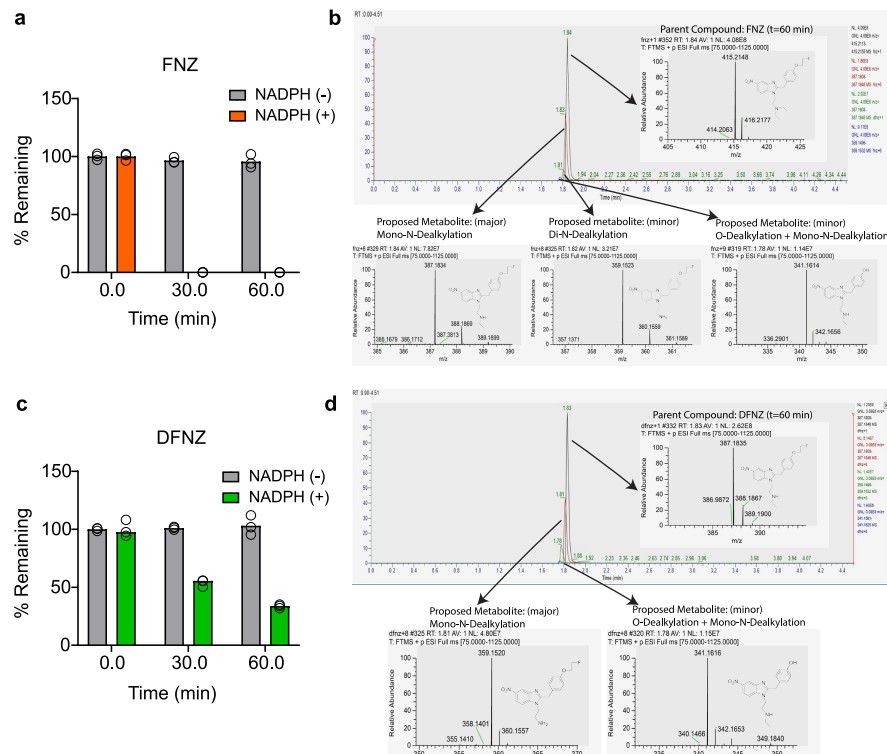

**Extended Data Fig. 2 | Stability and metabolite identification.** Stability of FNZ (**a**) and DFNZ (**c**) in Phase I metabolism in mouse liver microsomes fortified with NADPH (n = 3 independent experiments/drug) compared to negative control without NADPH (n = 3 independent experiments/drug). FNZ showed instability to Phase I metabolism in mouse liver microsomes and was undetected at both 30- and 60-min post incubation. DFNZ (major metabolite of FNZ) showed modest stability, with ~50% and 30% remaining at 30 and 60 min, respectively. Both compounds showed stability without NADPH, suggesting both compounds are metabolized through CYP enzymes. Metabolite identification (MET-ID) studies of FNZ (**b**) and DFNZ (**d**) in mouse liver microsomes following 60 min incubations. For FNZ, m/z 415.2140 peak was only observed at 0 min (retention time = 1.84 min) corresponding to the presence of FNZ. After 60 min, the major metabolite DFNZ (m/z = 387.1827, retention time = 1.83) was detected, accounting for over 50% of FNZ metabolism. For DFNZ (d), m/z 387.1827 peak was observed at 60 min (~30% remaining, retention time = 1.83 min). The Mono-N-Dealkylation metabolite, m/z = 359.1514, was observed as a major metabolite after 60 min incubation. Data shown as mean ± SEM.

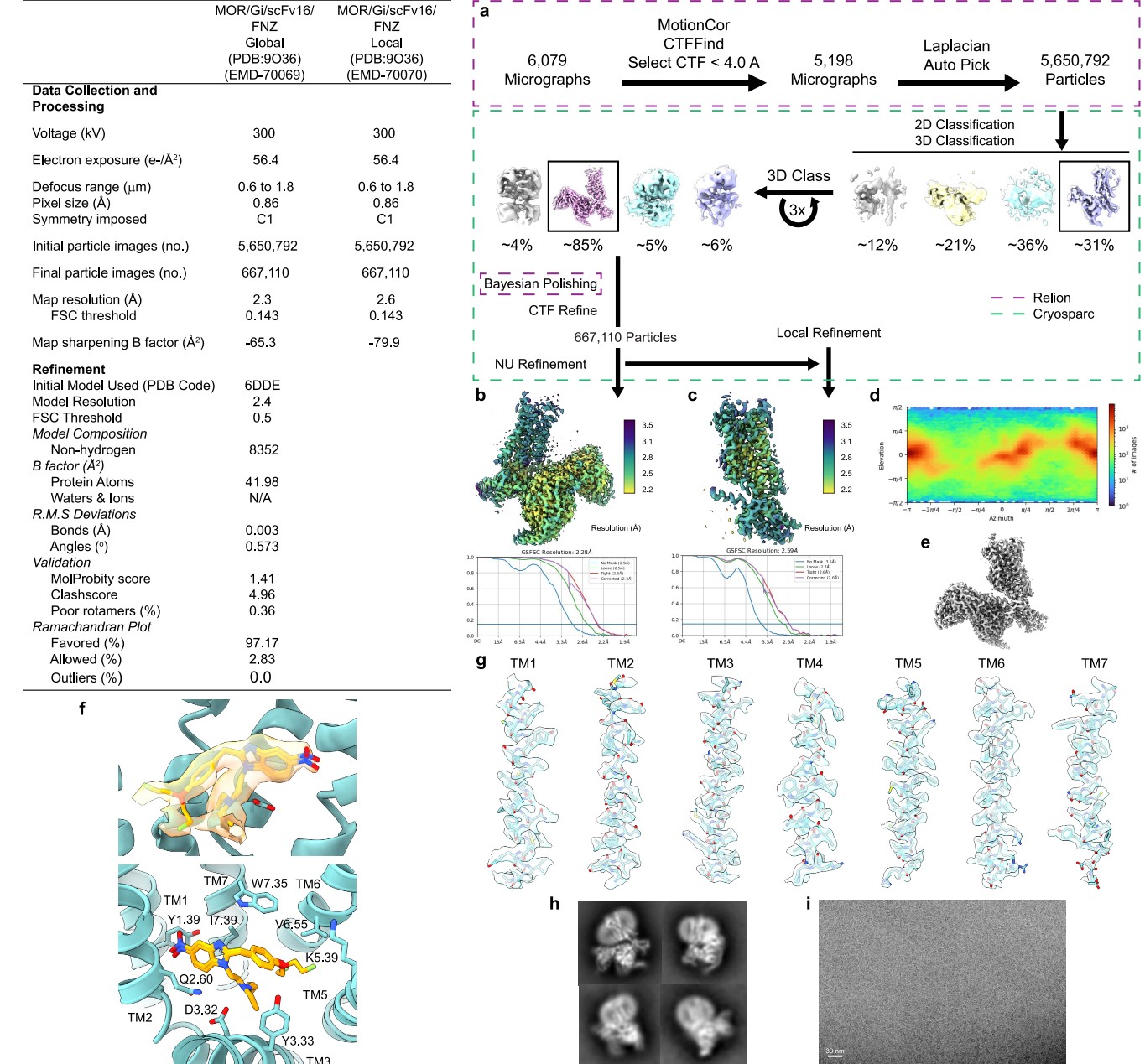

| | MOR/Gi/scFv16/ FNZ Global (PDB:9O36) (EMD-70069) | MOR/Gi/scFv16/ FNZ Local (PDB:9O36) (EMD-70070) |
|---|---|---|
| **Data Collection and Processing** | | |
| Voltage (kV) | 300 | 300 |
| Electron exposure (e-/Å²) | 56.4 | 56.4 |
| Defocus range (μm) | 0.6 to 1.8 | 0.6 to 1.8 |
| Pixel size (Å) | 0.86 | 0.86 |
| Symmetry imposed | C1 | C1 |
| Initial particle images (no.) | 5,650,792 | 5,650,792 |
| Final particle images (no.) | 667,110 | 667,110 |
| Map resolution (Å) | 2.3 | 2.6 |
| FSC threshold | 0.143 | 0.143 |
| Map sharpening B factor (Å²) | -65.3 | -79.9 |
| **Refinement** | | |
| Initial Model Used (PDB Code) | 6DDE | |
| Model Resolution | 2.4 | |
| FSC Threshold | 0.5 | |
| *Model Composition* | | |
| Non-hydrogen | 8352 | |
| *B factor (Å²)* | | |
| Protein Atoms | 41.98 | |
| Waters & Ions | N/A | |
| *R.M.S Deviations* | | |
| Bonds (Å) | 0.003 | |
| Angles (°) | 0.573 | |
| *Validation* | | |
| MolProbity score | 1.41 | |
| Clashscore | 4.96 | |
| Poor rotamers (%) | 0.36 | |
| *Ramachandran Plot* | | |
| Favored (%) | 97.17 | |
| Allowed (%) | 2.83 | |
| Outliers (%) | 0.0 | |

**Extended Data Fig. 3 | Cryo-EM data collection and processing. (a)** Cryo-EM data processing workflow. **(b)** Local resolution of global refinement with FSC curve below. **(c)** Local resolution of local refinement with FSC curve below. **(d)** Euler angle distribution. **(e)** Composite map. **(f)** Structure of both FNZ poses bound to MOR **(g)** Map-model agreement for the seven transmembrane helices of MOR. **(h)** 2D Class averages of MOR-Gi complex. **(i)** Example micrograph of MOR-Gi complex. 6,079 micrographs were collected in total from a single cryo-EM grid.

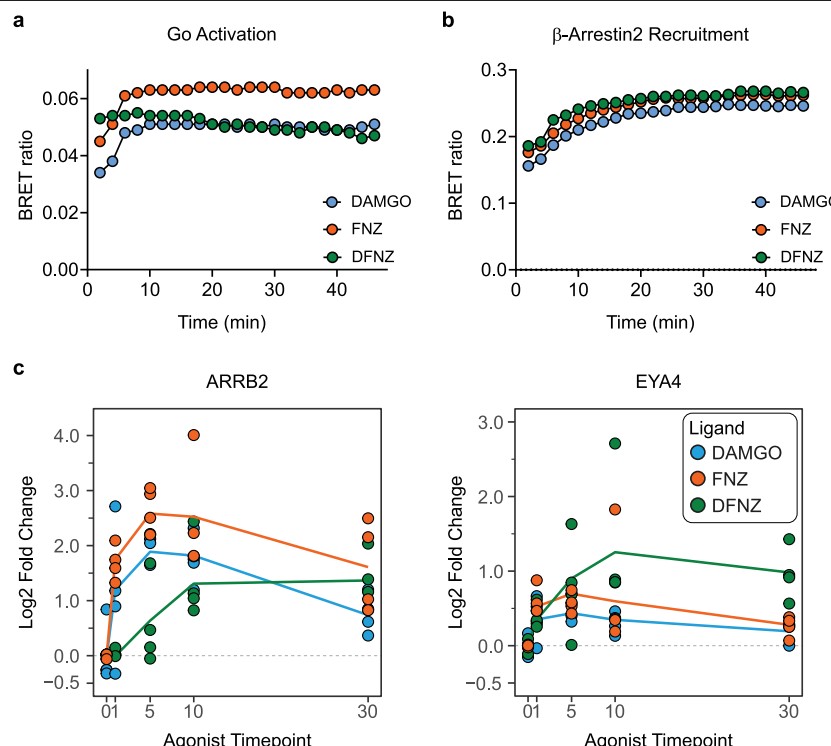

**Extended Data Fig. 4 | NET-BRET values and proximal labeling fold change.** (**a**, **b**) net-BRET values for $G_o$ activation and (b) for β-arrestin2 recruitment for the kinetic BRET experiments shown in Fig. 2. (**c**) Line charts showing log2 fold change in proximal labeling over the time course of receptor activation for ARRB2, a regulator of MOR endocytosis, and EYA4, a modulator of $G_{\alpha i}$ signaling downstream of MOR activation.

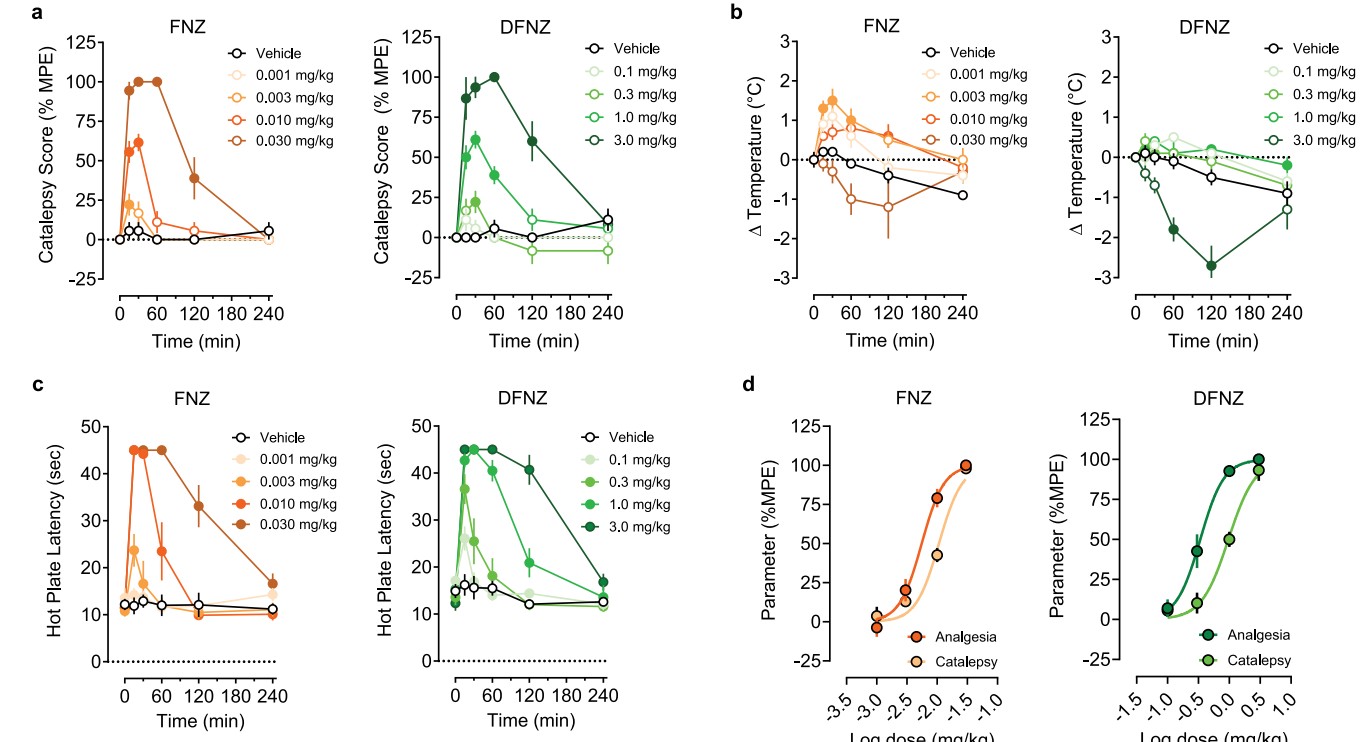

**Extended Data Fig. 5 | Catalepsy and body temperature.** (**a**) Catalepsy and (**b**) body temperature changes in rats (n = 6 rats/drug) in response to FNZ and DFNZ after subcutaneous injection (2-way ANOVA; dose x time interaction; p < 0.001; Dunnett's post-hoc tests; p < 0.05; closed circles denote statistical significance from vehicle). (**c**) Raw hot plate latency values for the data shown in Fig. 3b and c. (**d**) Analgesic potency is significantly different from cataleptic potency for both FNZ (F(1,43) = 21.83, p < 0.001) and DFNZ (F(1,43) = 41.96, p < 0.001). Data shown as mean ± SEM.

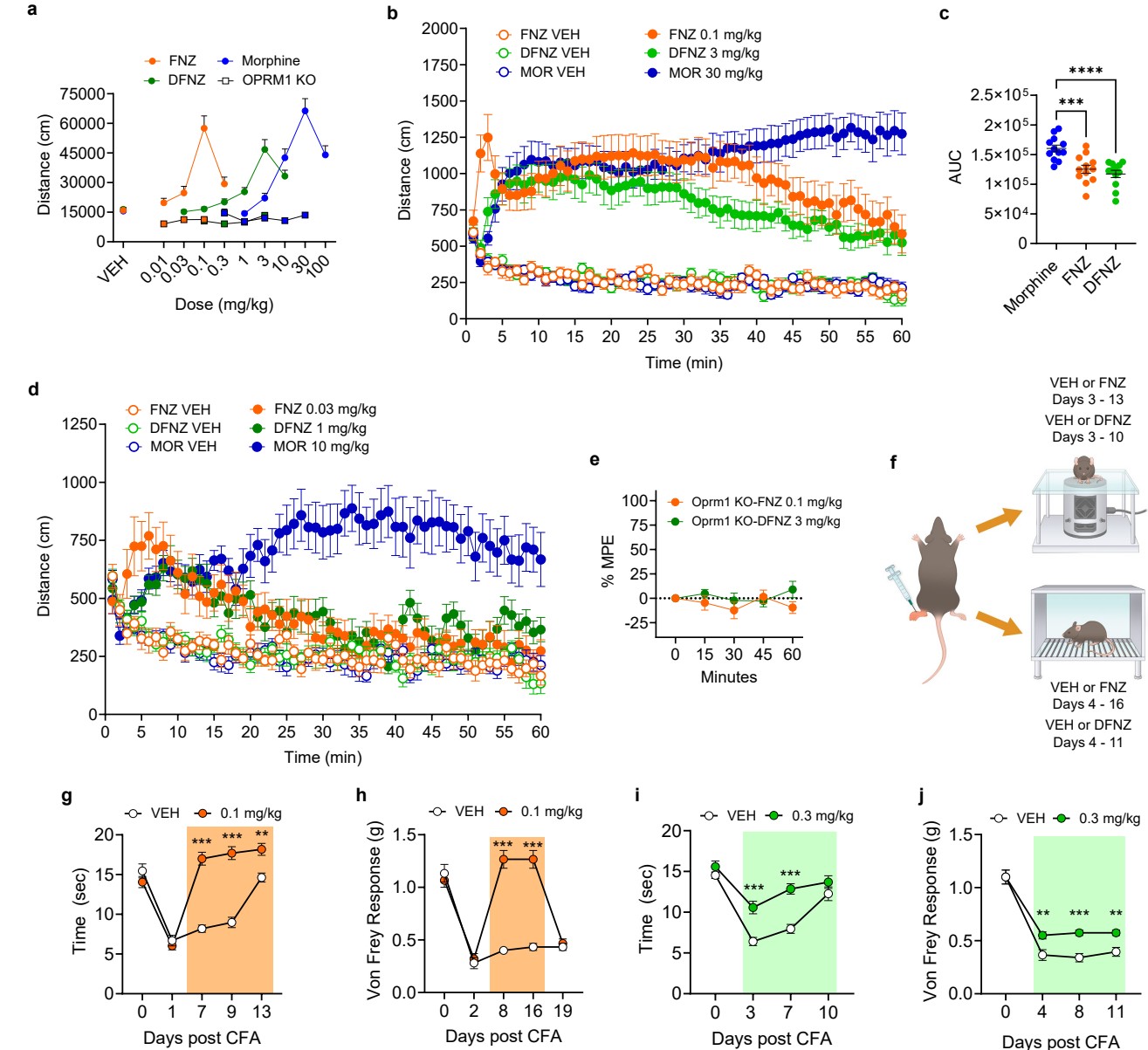

**Extended Data Fig. 6 | Locomotion and analgesia. (a)** Dose-response of drug-induced hyperlocomotion in wild type (n = 12 mice/drug) and *Oprm1* knockout (KO) mice (n = 8 mice/drug) (VEH=vehicle, MOR= morphine). **(b)** Hyperlocomotion data from wild type mice. At maximal drug doses, MOR hyperlocomotion increased by min ~10, was sustained for at least 30 min, and then further escalated. FNZ hyperlocomotion rapidly peaked at min ~3, quickly declined by min ~5, and was sustained until min ~40. DFNZ hyperlocomotion peaked at min ~5 and was sustained until min ~30. **(c)** Area under the curve (AUC) of top 3 doses shown in (a) (ANOVA; F(2, 33) = 13.54; p < 0.001; Dunnett post-hoc; p < 0.001). **(d)** Additional hyperlocomotion data from wild type mice. **(e)** Hotplate responses to FNZ (0.1 mg/kg) (n = 7 mice) and DFNZ (3 mg/kg) (n = 7 mice) in *Oprm1* KO mice. **(f)** Schematic showing mouse CFA model. **(g, h)** CFA induced a significant decrease in paw withdrawal latency (Hargreaves test) and threshold (von Frey test) post injection starting on day 1

while 0.1 mg/kg FNZ treatment (n = 6 mice) reversed these responses to pre-CFA levels 1 h after drug compared to VEH (n = 6 mice) (Hargreaves: 2-way RM-ANOVA; treatment x time; F(4, 40) = 22.95; p < 0.001; Tukey post-hoc; p < 0.01, von Frey: 2-way RM-ANOVA; treatment x time; F(4, 40) = 30.48; p < 0.001; Tukey post-hoc; p < 0.001). **(i, j)** Repeated daily injections of 0.3 mg/kg DFNZ in mice (n = 8 mice) produced significant and sustained antinociception compared to VEH (n = 8 mice) (assessed at 1 h post-injection) in both Hargreaves (2-way RM-ANOVA; treatment x time; F(3, 42) = 4.99; p = 0.004; Tukey post-hoc; p < 0.001) and von Frey tests 1 h after drug administration (2-way RM-ANOVA; treatment effect; F(1, 14) = 32.14; p < 0.001; Tukey post-hoc; p < 0.01). Neither FNZ nor DFNZ treatment triggered tolerance despite repeated daily administration. All post hoc tests were two-sided. Data shown as mean ± SEM. ****p < 0.0001, ***p < 0.001, **p < 0.01.

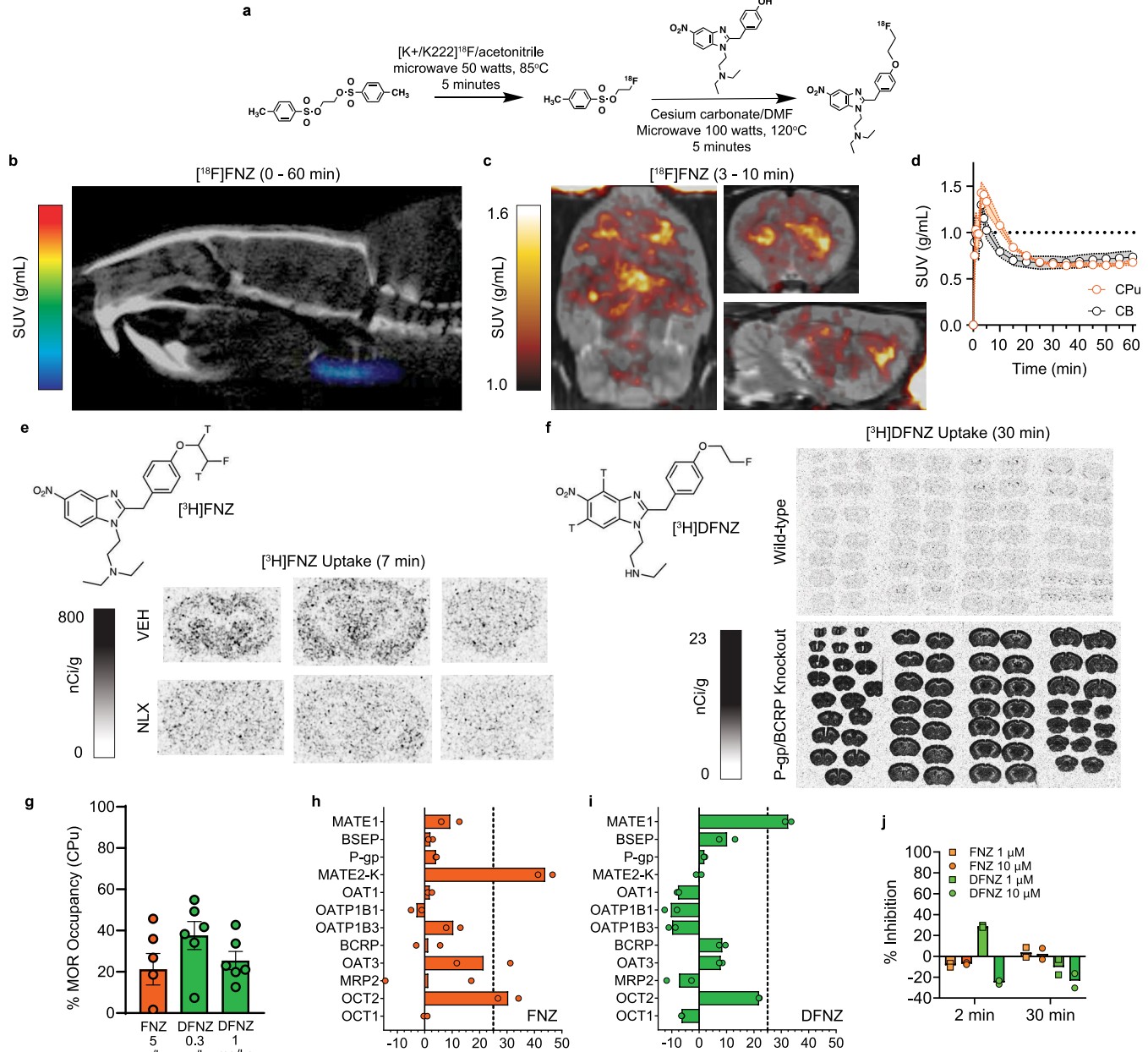

**Extended Data Fig. 7 | Pharmacokinetics, occupancy, and efflux transporter interactions.** (**a**) Synthesis scheme of [18F]FNZ. (**b**) Representative PET/CT image showing lack of [18F]FNZ brain uptake (Standard uptake value (SUV)) over 60 min after intravenous injection. (**c**) PET SUV image coregistered to MRI image showing accumulation of [18F]FNZ in the brain from 3 to 10 min after its injection (average SUV data from n = 3 rats). (**d**) SUV time activity curves in caudate putamen (CPu) and cerebellum (CB) (n = 3 rats). (**e**) [3H]FNZ (10 μg/kg) injected intravenously rapidly occupies MOR in rats (n = 2) pretreated with an intraperitoneal vehicle (VEH) injection but not in rats pretreated with 1 mg/kg naloxone (NLX) (n = 2). (**f**) [3H]DFNZ (100 μg/kg) injected subcutaneously localizes to ventricles in wild-type (n = 2 mice) and in the brain in P-gp/Bcrp knockout mice (n = 2 mice). (**g**) MOR occupancy of FNZ and DFNZ in caudate putamen (CPu) in rats (n = 6 rats/drug). (**h-i**) Transporter inhibition profiling of FNZ and DFNZ at 100 nM. (**j**) Percent inhibition by FNZ and DFNZ in the Mate1 substrate assessment assay (n = 2 independent experiments). Data shown as mean ± SEM.

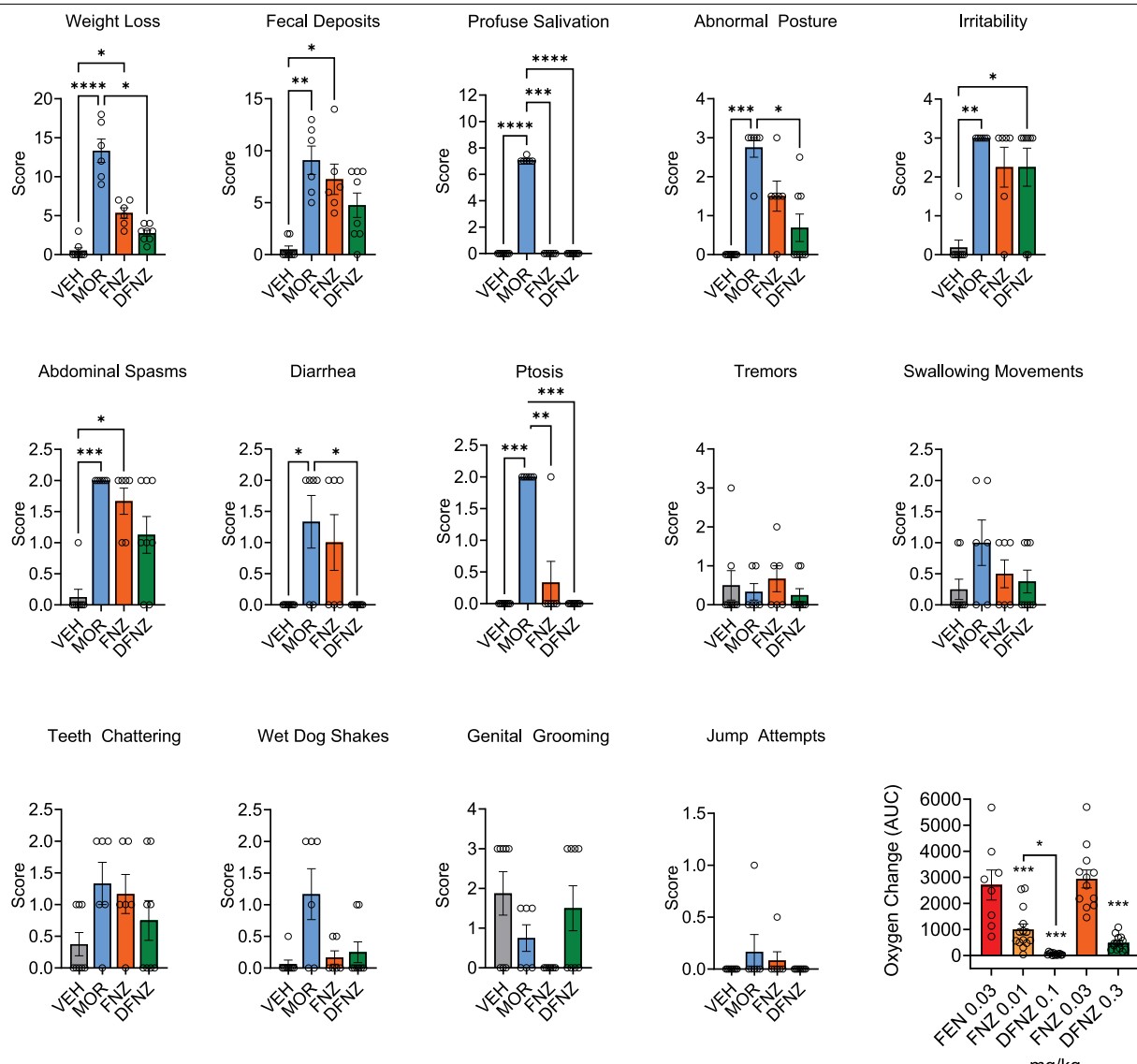

**Extended Data Fig. 8 | Specific precipitated withdrawal behaviors and brain oxygen measures.** Withdrawal behaviors were assessed in rats exposed to vehicle (VEH, n = 6 rats), morphine (MOR, n = 6 rats), FNZ (n = 6 rats), or DFNZ (n = 8 rats) after injection of 1 mg/kg naloxone. All data analyzed using Kruskal-Wallis ANOVA with Dunn's post hoc test. Weight Loss: H(3) = 23.42; p < 0.001; Fecal Deposits: H(3) = 16.19; p = 0.001; Profuse Salivation: H(3) = 26.78; p < 0.001; Abnormal Posture: H(3) = 18.57; p < 0.001; Irritability: H(3) = 15.87; p = 0.001; Abdominal Spasms: H(3) = 17.87; p < 0.001; Diarrhea: H(3) = 12.43; p = 0.006; Ptosis: H(3) = 22.71; p < 0.001. Area under the curve (AUC) of brain oxygen changes from data shown in Fig. 5a: FNZ (14 independent experiments), DFNZ (n = 14 independent experiments), or fentanyl (FEN) (n = 8 independent experiments) (ANOVA; F(4, 56) = 25.45; p < 0.001; Holm-Sidak post-hoc tests; p < 0.05). Data shown as mean ± SEM. ****p < 0.0001, ***p < 0.001, **p < 0.01, *p < 0.05.

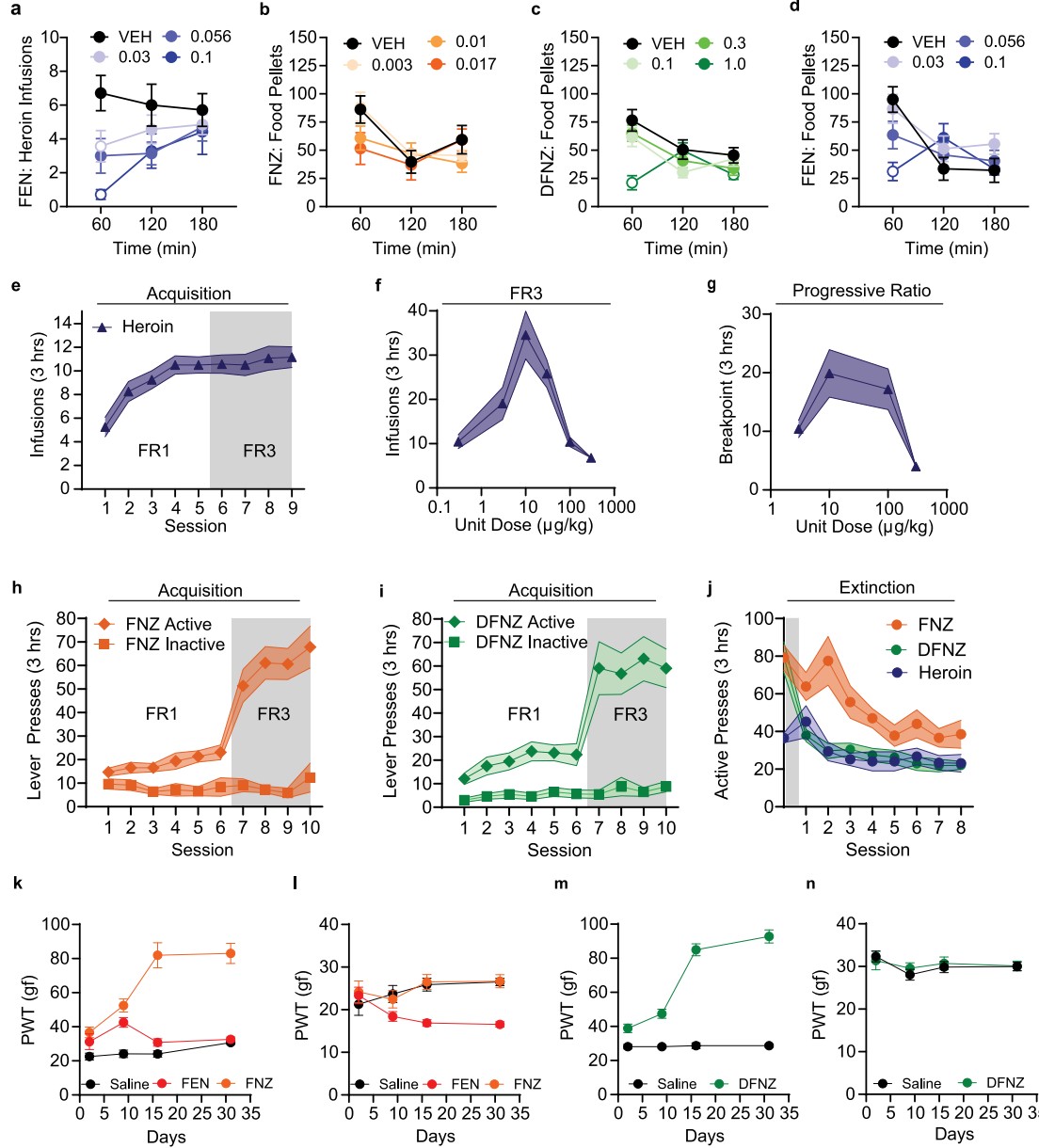

**Extended Data Fig. 9 | Reinforcing effects and tolerance.** (**a**) Effect of vehicle (VEH) or fentanyl (FEN) on heroin self-administration (n = 7 rats). (**b**) Effect of FNZ (n = 6 rats), (**c**) DFNZ (n = 14 rats), and (**d**) FEN (n = 6 rats) on food pellet responding. All data analyzed via 2-way repeated measures or mixed effects models with significant dose main effects or dose x time interaction effects (p < 0.05) and Dunnett's post-hoc (p < 0.05). All open circles denote statistical significance from VEH at each corresponding time point. (**e**) Acquisition (n = 12 rats) (FR1: sessions 1–6 and FR3 (sessions 7–10)), (**f**) FR3 (n = 11 rats) and (**g**) progressive ratio dose response curves for intravenous heroin

self-administration in rats (n = 10 rats). (**h**) Intravenous self-administration training (active and inactive lever presses) of FNZ (n = 11 rats, 1 μg/kg/infusion) and (**i**) DFNZ (n = 8 rats, 30 μg/kg/infusion) in rats. (**j**) Active lever presses for FNZ (n = 11 rats), DFNZ (n = 8 rats) and heroin (n = 10 rats) during normal acquisition (shaded area, session 0) and extinction sessions (session 1–8). Raw grams force (gf) paw withdrawal threshold (PWT) data shown in Fig. 5f and g at 30 min (**k**, **m**) and -5 h (**l**, **n**) after injection in rats exposed to repeated drug injections. Data shown as mean ± SEM.

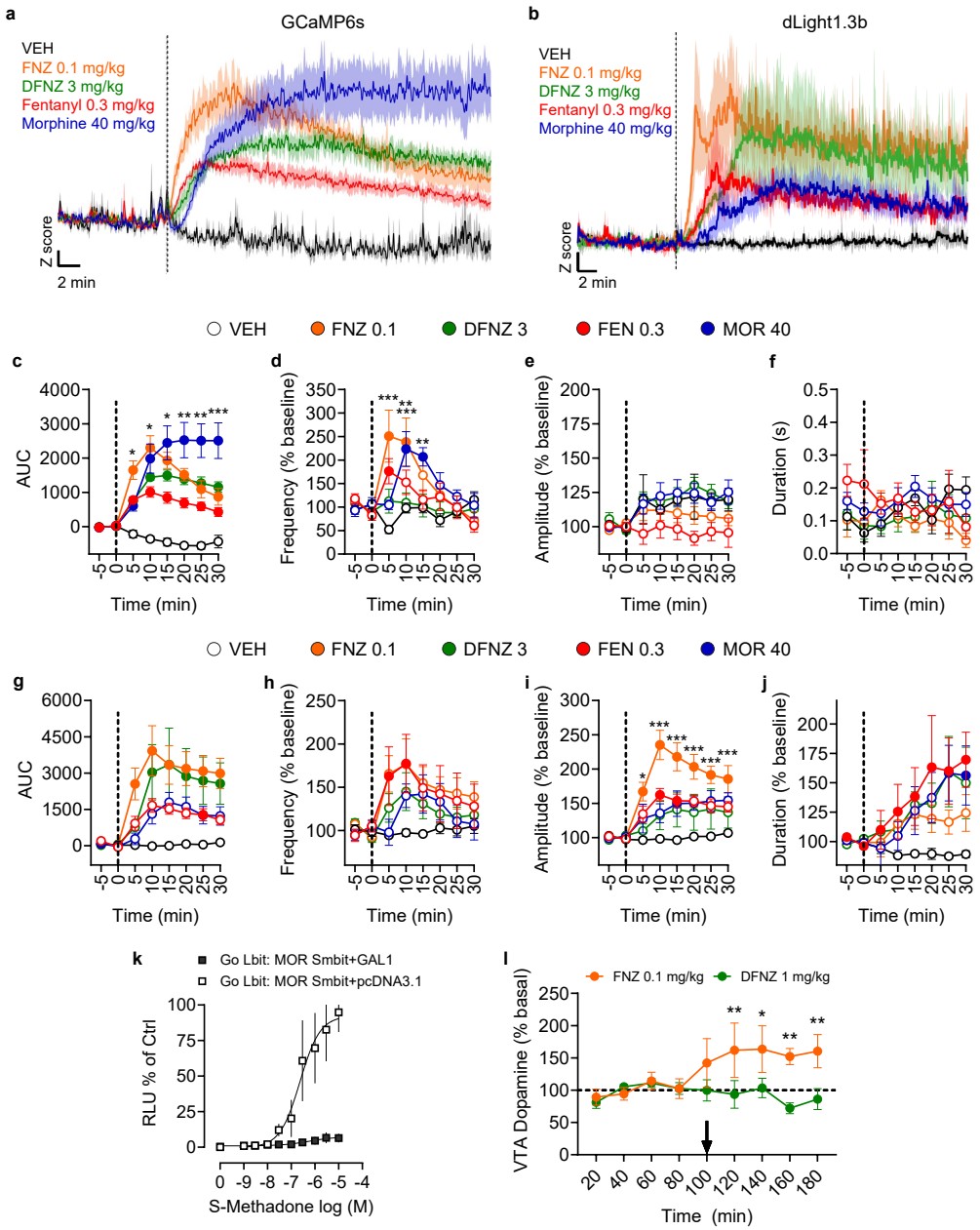

**Extended Data Fig. 10 | Photometry and MOR-Gal1R experiments.**
(**a**) Averaged raw axon-GCaMP6s (n = 5 mice/drug) and (**b**) dLight traces (n = 4–6 mice/drug) following vehicle (VEH), fentanyl (FEN), morphine (MOR), FNZ, or DFNZ. (**c-f**) Averaged axon-GCaMP6s traces showing AUC (n = 5 mice/drug) (Mixed-effects ANOVA; treatment x time; $F_{(28, 110)}$ = 13.90; p < 0.001; Holm-Sidak post-hoc; p < 0.05), spike frequency (n = 5 mice/drug) (Mixed-effects ANOVA; treatment x time; $F_{(28, 110)}$ = 3.54; p < 0.001; Tukey post-hoc; p < 0.05), spike amplitude, and spike duration. (**g-j**) Averaged traces of dLight1.3b activity (VEH: 5 mice, FEN: 4 mice, MOR: 5 mice, FNZ: 6 mice, DFNZ: 4 mice) showing AUC (2-way RM-ANOVA; treatment x time; $F_{(28, 129)}$ = 3.69; p < 0.001; Tukey post-hoc; p < 0.05), peak frequency (2-way RM-ANOVA; treatment x time;

$F_{(28, 133)}$ = 1.83; p = 0.01; Tukey post-hoc; p < 0.05), peak amplitude (2-way RM-ANOVA; treatment x time; $F_{(28, 133)}$ = 4.65; p < 0.001; Tukey post-hoc; p < 0.05), and peak duration (2-way RM-ANOVA; treatment x time; $F_{(28, 133)}$ = 2.19; p < 0.001; Tukey post-hoc; p < 0.05). Dotted line indicates injection of saline or drugs. Closed circles denote statistical significance from saline. (**k**) (S)-methadone has a loss of MOR $G_o$ efficacy in cells co-transfected with MOR and GAL1 receptor. (**l**) FNZ (n = 5 rats) significantly increased somatodendritic dopamine in the VTA but DFNZ (n = 5 rats) did not (2-way RM-ANOVA; treatment effect; $F_{(1, 8)}$ = 5.67; p = 0.04; Tukey post-hoc; p < 0.05). All post hoc tests were corrected for multiple comparisons. Data shown as mean ± SEM. ***p<0.001, **p < 0.01, *p < 0.05.

# Reporting Summary

## Statistics

For all statistical analyses, confirm that the following items are present in the figure legend, table legend, main text, or Methods section.

| n/a | Confirmed | |
|---|---|---|
| ☐ | ☒ | The exact sample size ($n$) for each experimental group/condition, given as a discrete number and unit of measurement |
| ☐ | ☒ | A statement on whether measurements were taken from distinct samples or whether the same sample was measured repeatedly |
| ☐ | ☒ | The statistical test(s) used AND whether they are one- or two-sided<br>*Only common tests should be described solely by name; describe more complex techniques in the Methods section.* |
| ☐ | ☒ | A description of all covariates tested |
| ☐ | ☒ | A description of any assumptions or corrections, such as tests of normality and adjustment for multiple comparisons |
| ☐ | ☒ | A full description of the statistical parameters including central tendency (e.g. means) or other basic estimates (e.g. regression coefficient) AND variation (e.g. standard deviation) or associated estimates of uncertainty (e.g. confidence intervals) |
| ☐ | ☒ | For null hypothesis testing, the test statistic (e.g. $F$, $t$, $r$) with confidence intervals, effect sizes, degrees of freedom and $P$ value noted<br>*Give P values as exact values whenever suitable.* |
| ☒ | ☐ | For Bayesian analysis, information on the choice of priors and Markov chain Monte Carlo settings |
| ☒ | ☐ | For hierarchical and complex designs, identification of the appropriate level for tests and full reporting of outcomes |
| ☒ | ☐ | Estimates of effect sizes (e.g. Cohen's $d$, Pearson's $r$), indicating how they were calculated |

*Our web collection on statistics for biologists contains articles on many of the points above.*

## Software and code

Policy information about availability of computer code

| | |
|---|---|
| Data collection | MED-PC v. 4.2 (Med Associates), Synapse Software v. 95-44132P (Tucker Davis) , Neuroscience Studio v6 (Doric), Nucline NanoScan 3.04.025 (Mediso), Multigauge v3 (Fujifilm), Spectronaut v19.7 (Biognosys), Microbeta2 workstation software (Revvity), Serial EM (ver. 3.9), EPU (v. 2.10). |
| Data analysis | Microsoft Excel 2024 and 2025, GraphPad Prism 10, MatlabR2016 & MatlabR2023b, Statistical Parametric Mapping (SPM12), PMOD v4.1, Multigauge v3 (Fujifilm), MSStats v.4.10.1,  Xcalibur v.4.4.16.14 (ThermoFisher), Neuroscience Studio v6 (Doric). Code for photometry analysis is available at Github (https://github.com/wdunne3/Calcium-Fiber-Photometry-Analysis/ and https://github.com/BonaventuraLab/fiber-photometry). RELION-4.0, cryoSPARC (v4.0), CTFFIND4, and MotionCor2 were used to process cryoEM data. Molecular model was carried out using Coot (version 0.9.8.1 EL) and Phenix (version 1.20.1-4487). GemSpot pipeline utility of Maestro (v.13.8) (Schrodinger) was used to dock ligands. |

For manuscripts utilizing custom algorithms or software that are central to the research but not yet described in published literature, software must be made available to editors and reviewers. We strongly encourage code deposition in a community repository (e.g. GitHub). See the Nature Portfolio guidelines for submitting code & software for further information.

# Data

Policy information about availability of data

All manuscripts must include a data availability statement. This statement should provide the following information, where applicable:

- Accession codes, unique identifiers, or web links for publicly available datasets
- A description of any restrictions on data availability
- For clinical datasets or third party data, please ensure that the statement adheres to our policy

The authors declare that the data supporting the findings of this study are available within the paper and its Supplementary Information files. The cryoEM density map has been deposited in the Electron Microscopy Data Bank under accession code EMD-70069 (global refinement), EMD-70070 (local refinement, receptor), EMD-70071 (composite map) and the coordinates have been deposited in the Protein Data Bank under accession number 9O36. The mass spectrometry proteomics data have been deposited to the ProteomeXchange Consortium via the PRIDE (https://www.ebi.ac.uk/pride/) partner repository71 with the dataset identifier PXD062863. Reviewers can access these data using the project accession PXD062863 and reviewer token "62GfWQNbF6w3". Alternatively, these data may also be accessed by logging into the PRIDE website using the reviewer username "reviewer_pxd062863@ebi.ac.uk" and password "mmgadjWUcRls". The PDB:6DDE dataset is available at https://www.rcsb.org/structure/6DDE. Source data are provided in the manuscript. Should any other data be needed they are available from the corresponding author upon reasonable request.

# Research involving human participants, their data, or biological material

Policy information about studies with human participants or human data. See also policy information about sex, gender (identity/presentation), and sexual orientation and race, ethnicity and racism.

| Reporting on sex and gender | N/A |
|---|---|
| Reporting on race, ethnicity, or other socially relevant groupings | N/A |
| Population characteristics | N/A |
| Recruitment | N/A |
| Ethics oversight | N/A |

Note that full information on the approval of the study protocol must also be provided in the manuscript.

# Field-specific reporting

Please select the one below that is the best fit for your research. If you are not sure, read the appropriate sections before making your selection.

☒ Life sciences    ☐ Behavioural & social sciences    ☐ Ecological, evolutionary & environmental sciences

For a reference copy of the document with all sections, see nature.com/documents/nr-reporting-summary-flat.pdf

# Life sciences study design

All studies must disclose on these points even when the disclosure is negative.

| Sample size | Sample sizes were estimated based on experience from relevant past work in our laboratories and prior literature (e.g. J Neurophysiol. 2026 Jan 1;135(1):130-141. doi: 10.1152/jn.00504.2025, Nature. 2025 Oct;646(8085):746-753. doi: 10.1038/s41586-025-09427-8, Neuropsychopharmacology. 2017 Jun;42(7):1548-1556. doi: 10.1038/npp.2017.4, and Mol Psychiatry. 2024 Mar;29(3):624-632. doi: 10.1038/s41380-023-02353-z.). |
|---|---|
| Data exclusions | Animals were excluded if they lost catheter patency or due to lack of photometry signal. Rats for self-administration were excluded for sessions where the infusion line disconnected from the rat's catheter port. Photometry data were excluded for sessions where the patchcord became disconnected from the implanted fiber. |
| Replication | Detailed methods are provided to aid in replication by others. Behavioral (analgesia, locomotor, hypoxia, tolerance, self-administration) and photometry experiments were replicated in at least 2 cohorts or species and by more than one experimenter. Pharmacokinetic experiments were replicated in at least two species. PET studies were replicated across individual animals. In vitro assays (binding and functional) were replicated in at least two different laboratories and were performed with at least two independent experiments. All other in vitro assays were replicated with at least two independent replicates. All replication attempts were successful. |
| Randomization | Animals were randomly assigned to experimental groups and treatment conditions. Samples for in vitro assays were randomized with respect to treatment conditions. |
| Blinding | Experimenters were not blinded to group allocation during data collection for most in vivo experiments because data were collected in an |

| Blinding | automated manner without manual scoring. Data were analyzed blind if applicable (e.g., PET, photometry) because of automated analysis but experimenters were always aware of the conditions. Withdrawal experiments were performed blinded as noted in the manuscript. |

# Reporting for specific materials, systems and methods

We require information from authors about some types of materials, experimental systems and methods used in many studies. Here, indicate whether each material, system or method listed is relevant to your study. If you are not sure if a list item applies to your research, read the appropriate section before selecting a response.

## Materials & experimental systems

| n/a | Involved in the study |
|-----|-----------------------|
| ☒ ☐ | Antibodies |
| ☐ ☒ | Eukaryotic cell lines |
| ☒ ☐ | Palaeontology and archaeology |
| ☐ ☒ | Animals and other organisms |
| ☒ ☐ | Clinical data |
| ☒ ☐ | Dual use research of concern |
| ☒ ☐ | Plants |

## Methods

| n/a | Involved in the study |
|-----|-----------------------|
| ☒ ☐ | ChIP-seq |
| ☒ ☐ | Flow cytometry |
| ☒ ☐ | MRI-based neuroimaging |

## Eukaryotic cell lines

Policy information about cell lines and Sex and Gender in Research

| Cell line source(s) | ATCC CRL-321, HEK-293, HEK293T |
| Authentication | Cell lines were not authenticated after receipt from ATCC |
| Mycoplasma contamination | Cell lines were tested for mycoplasma contamination as noted in the manuscript. If contaminated cultures were detected, they were not used for experiments. |
| Commonly misidentified lines (See ICLAC register) | HEK cells can be misidentified as HeLa cells but HeLa cells were not used in these experiments. |

## Animals and other research organisms

Policy information about studies involving animals; ARRIVE guidelines recommended for reporting animal research, and Sex and Gender in Research

| Laboratory animals | Sprague-Dawley rats (Charles River) (6-8 weeks old), C57BL/6J mice (Charles River or Jackson Laboratory) (8 weeks old), TH-cre mice (B6.Cg-7630403G23RikTg(Th-cre)1Tmd/J, NIDA) (10-12 weeks old), Oprm1 knockout mice (B6.129S2-Oprm1tm1Kff/J, Jackson Laboratory) (6-8 weeks old), P-gp/Bcrp knockout mice (FVB.129P2-Abcb1atm1BorAbcb1btm1BorAbcg2tm1Ahs, Taconic Biosciences) (6-8 weeks old). |
| Wild animals | The study did not involve wild animals |
| Reporting on sex | We used male and female rats and mice for all experiments unless specified in the methods. |
| Field-collected samples | The study did not involve animals collected from the field |
| Ethics oversight | Experiments and procedures complied with ethical regulations for animal testing and research, followed the NIH guidelines and were approved by the relevant institutional animal care and use committees (NIDA, Boston University, University of Barcelona). |

Note that full information on the approval of the study protocol must also be provided in the manuscript.

# Plants

Seed stocks

*Report on the source of all seed stocks or other plant material used. If applicable, state the seed stock centre and catalogue number. If plant specimens were collected from the field, describe the collection location, date and sampling procedures.*

Novel plant genotypes

*Describe the methods by which all novel plant genotypes were produced. This includes those generated by transgenic approaches, gene editing, chemical/radiation-based mutagenesis and hybridization. For transgenic lines, describe the transformation method, the number of independent lines analyzed and the generation upon which experiments were performed. For gene-edited lines, describe the editor used, the endogenous sequence targeted for editing, the targeting guide RNA sequence (if applicable) and how the editor was applied.*

Authentication

*Describe any authentication procedures for each seed stock used or novel genotype generated. Describe any experiments used to assess the effect of a mutation and, where applicable, how potential secondary effects (e.g. second site T-DNA insertions, mosiacism, off-target gene editing) were examined.*

