## [Peer Review file · Nature]

A μ opioid receptor superagonist analgesic with minimal adverse effects

Corresponding Author: Dr Michael Michaelides

Version 0:

Reviewer comments:

Referee #1

(Remarks to the Author)

Gomez and describes a newly discovered agonist for the human μ -type opioid receptor and present a series of studies demonstrating its efficacy and minimal adverse effects in pain treatment. The authors obtained a cryo-EM structure of the MOR-Gi protein complex bound to this agonist and compared the binding pockets across different opioid receptor types to assess the agonist's selectivity.

This technical assessment is focused on the cryo-EM aspects of the work.

The cryo-EM maps were refined to high resolution, and the atomic model fits well within them, with both possible agonist conformations included. The supplementary figure related to the cryo-EM analysis provides detailed information on data processing and evaluations of the maps and model. Overall, the data are of high quality, and the conclusions are well supported.

Several points should be addressed:

1. The notation for amino acid residues is confusing and we could not understand it (e.g., D3.32, I7.39). Our expectation is that D3 refers to Asp3 (Aspartate 3) and I7 refers to Ile7 (Isoleucine 7) but this does not appear to be the case. Please clarify the labelling of amino acid residues in Fig. 1L
2. The cryo-EM figures are rendered beautifully, but the text size could be increased for clarity. In addition, if the FNZ molecule is modelled in two different conformations, perhaps Figure 1K should show both conformations, and include a corresponding callout in the text.
3. Although compelling map density is shown for FNZ, map density should be shown somewhere (e.g. in an extended data figure) for the amino acid side chains depicted in Fig. 1L to convince the reader that the orientations of the side chains in the model are reliable.
4. In Figure 1L, the D3.32 label is near where Q2.60 is, and be closer to the actual side chain.
5. Features discussed in the text should be clearly highlighted in the figures. For example:
 - a. Figure 1K should highlight the FNZ binding site (the main takeaway from the figure) with a circle or box in the composite map.
 - b. TM2 and 6, which are discussed in the text, should be labeled in Figure 1K or a new supplementary figure (with an appropriate figure callout).
 - c. Residues like I7.39 and Y1.39 are discussed in the text but not shown in Figure 1L, raising questions about their relevance or importance.
6. A representative micrograph and example 2D class average images should be shown (e.g. in Extended Data Fig. 6).

7. There is a discrepancy in the chain ID of FNZ in the model: it appears as chain D in the atomic model provided to us, but is labeled as chain R in the PDB report. This might have been corrected already, but please double check.

8. In the cryo-EM statistics table (currently part of Extended Data Fig. 6):

- under-focus values should be positive numbers with standard expressions for the CTF.
- B factors used for sharpening should be negative
- The cutoff for assessing the resolution of a noise-free model versus the map from the full dataset should be at FSC=0.5 not FSC=0.143 (FSC=0.143 is the cutoff used for half-map versus half-map).
- Please also include the PDB code in the Table, not just the EMD codes

Referee #2

(Remarks to the Author)

I co-reviewed this manuscript with one of the reviewers who provided the listed reports.

Referee #3

(Remarks to the Author)

Summary of key results

The manuscript entitled "A μ opioid receptor superagonist that produces analgesia with minimal adverse effects" reports on two fluorinated nitazenes: fluornitazene (FNZ) and its primary metabolite N-desethyl-FNZ (DFNZ). The authors studied pharmacodynamic, pharmacokinetic and behavioural response profiles of both of these compounds, comparing them to DAMGO or to opioid analgesics such as morphine or fentanyl. They conclude that DFNZ is a MOR agonist with supra maximal intrinsic efficacy (superagonist) which preferentially activates G protein signaling over α arr recruitment. They speculate that this unique bias profile in association with low brain accumulation and reduced signaling efficacy at the MOR-Gal1R heteromer may explain in vivo responses observed for DFNZ. The authors have interpreted DFNZ's in vivo response profile as an indication that this ligand behaves as an analgesic that does not produce respiratory depression, tolerance nor withdrawal. They also propose that DFNZ has weak reinforcing properties and fails to modify dopaminergic transmission in the NAcc of the reward circuit. Based on these results the authors challenge the notion that high efficacy MOR agonists are not suitable for development as safe analgesics.

Originality, significance

The subject of the study is very important and timely, given the current opioid overdose crisis. In this context, efforts to develop safer opioid analgesics warrant serious consideration. The techniques that were used to characterize pharmacodynamic, pharmacokinetic and behavioural response profiles are state of the art. The notion that superagonists of the MOR receptor could lead to development of therapeutically safe opioid analgesics is certainly thought provoking. On these bases the study merits consideration. However, my main concern is that the study falls short of demonstrating DFNZ's safety.

Main concerns

With safety in mind, it is imperative to show that, given at similarly effective analgesic doses as currently available painkillers the superagonist DFNZ has a better profile of respiratory effects and reduced reinforcement properties. In the present study the side effect profile of DFNZ was most frequently (if not systematically) established using doses whose analgesic effects were less than those of the parent compound (FNZ) and that of current therapeutic standards (morphine, fentanyl). I have detailed below the examples where I think low effective doses of DFNZ are at the bases of the claimed safety of this novel product and have suggested possible experiments to address this issue.

Another important concern involves pharmacodynamic measures that support the claim for biased signaling. There is an incongruence in EC50 values presented for GalphaoA in figures 1 and 2. Detailed comments on this point follow below.

Finally, no actual experimental verification was provided to substantiate that either bias, low brain accumulation or reduced efficacy at the MOR-Gal1R heteromer were responsible for improved side effects profile of DFNZ at analgesic doses. A mechanistic verification that any of these mechanisms is actually involved in supporting a safe respiratory profile for DFNZ would considerably strengthen the manuscript.

Detailed comments for the authors on data, methodology, statistics, robustness of conclusions and suggested improvements

PHARMACODYNAMIC PARAMETERS, SIGNALING BIAS AND TRAFFICKING PROFILES

1) Kinetic assays show that FNZ and DFNZ have similar MOR binding properties but K_i values for DAMGO displacement indicate >10-fold higher affinity for DFNZ than for FNZ. An explanation for these differences should be provided.

2) Figure 1c, 1e-1j: The legend indicates four replicates. Replicates usually refer to the number of repeats within one experiment. What is the number of independent experiments for the curves shown in these figures?

3) Figure 2a needs to be corrected to indicate the use of BRET1 and BRET2 biosensors as described in methodological

section. According to methods, the Galpha biosensor contains GFP2 and not Venus as indicated.

4) Methods should include a section describing how DDLog and its compound error were calculated for figure 2f. As it stands the reader does not know whether bias measures were calculated from logistic or operational parameters.

Do graphs in Figure 2b-2f correspond to representative examples? How many times were kinetic experiments repeated?

5) There is incongruence in EC50 data from curves in Figure 1h (and described in pg 3) and the pEC50 value shown in figure 2d.

i.e: according to the methodological section the curves and corresponding EC50 values in figure 1h were obtained 15 min after addition of the ligands. GalphaoA EC50 values obtained from these curves (described in pg 3) indicate that FNZ is more potent (6.6 fold) than DFNZ. In figure 2d the pEC50 values at the 15 min timepoint show that DFNZ is more potent than FNZ. This incongruence needs to be revised and explained since the authors conclude that higher DFNZ potency for the G protein response is what drives DFNZ's unique bias profile. This bias profile is suggested as a possible mechanism underlying beneficial side effects profile of DFNZ.

6) The authors use BRET2 (GFP2, Coelenterazine 400a) to monitor G protein responses and BRET1 (Venus, Coelenterazine h) to monitor beta-arrestin recruitment. Luminescence by Coelenterazine 400a decays much quicker than that of Coelenterazine h, implying that the time frame for visualization of the G protein response is shorter than that of visualization of beta-arr recruitment. Note that the difference in time frame of effective energy transfer is not corrected by the ratiometric nature of the BRET response. BRET figures (2b-2f) provided by the authors show normalized rather than net-BRET values so we do not know how the actual G protein and beta-arr2 responses of the standard DAMGO evolve overtime. net-BRET data at least for the standard should be provided in supplementary information. I would also appreciate the authors' insight on how the use of BRET1 and BRET2 biosensors may have influenced their conclusions in relation to biased signaling.

7) The notion that G protein over beta-arr bias could yield safer opioid analgesics was based on initial observations by the Bohn laboratory showing that barr2-KO mice had reduced respiratory depression and less constipation when transgenic mice were injected with opioids (doi: 10.1021/acs.jmedchem.8b01136). These observations could not be reproduced by three independent laboratories (doi: 10.1111/bph.15004) and since then the notion that G protein signaling preference is a good predictor of opioid safety has been questioned (e.g.: doi: 10.1016/j.tips.2020.09.009). The authors should provide rationalization of how avoiding beta-arr signaling may contribute to DFNZ's beneficial side effect profile and/or provide experimental evidence that bias is at the basis of DFNZ's safety. Note that a comprehensive study of nitazene biased signaling recently concluded that intrinsic efficacy of nitazenes in Galpha protein signaling is a common property underlying their high risk of overdose (doi: 10.1021/acschemneuro.3c00750). It is also worth noting that bupropion, another highly efficacious MOR agonist that was developed as a safer, G protein-biased, opioid analgesic (doi: 10.1021/acs.jmedchem.8b01136) was at the basis of numerous overdose deaths (doi: <https://doi.org/10.1093/jat/bkab082>). How does the superagonist DFNZ differ from G protein-biased nitazenes and bupropion to be considered safer?

8) The methodological section indicates that Emax was measured as ... "tangent to the plateau of the dose-response curve". The maximal response elicited by a drug is measured by the SPAN = maximal asymptote – minimal asymptote. The use of SPAN allows to consider variations in basal activity (for example a partial agonist in an activated system can produce the same maximal asymptote as a full agonist). Also note that Emax should not be equated with efficacy since in amplified systems full and partial agonists produce maximal system responses. Emax values should be corrected to conform to Emax definition (i.e. SPAN).

9) In addition to G protein/beta-arrestin signaling, the authors provide a trafficking profile by FNZ and DFNZ. However, there is no further mention how this profile may mechanistically determine the beneficial side effect profile that the authors associate with DFNZ.

Also, the authors seem to associate DFNZ's trafficking profile to preferential G protein activation (pg 5). It should be noted that ligands like morphine or PMZ21 that are less efficacious than DAMGO showed as similar profile as DFNZ with less endosomal trafficking, stronger EYA4 interaction (considered by the authors as a surrogate of G protein activation) and weaker beta-arr2 labeling (doi.org/10.1038/s41589-024-01588-3). How do the authors rationalize the fact that DFNZ's trafficking profile, which they define as superagonist, resembles that of partial agonists. How do the observed trafficking properties contribute to beneficial side effects of DFNZ as compared to morphine whose trafficking it resembles.

10) In the methodological section for APEX, the references provided do not match the reference list.

ANALGESIA

1) Analgesic responses in the hot plate test were assessed in rats over a period of 3 h with the higher doses tested producing an analgesic response over this full time period. However, dose response curves only considered the first 60 min of analgesia, which inevitably skews ED50 towards lower doses. Why was this done?

2) Authors compare ED50 values for analgesia, catalepsy and hypothermia without providing statistics (pg 5-6). Assessment of statistical significance of the distinct potencies for inducing desired (analgesia) and undesired effects (catalepsy, hypothermia) should be provided.

3) In the CFA mouse model, a single dose was tested. How were these doses chosen? When one looks at hot plate tests in

rats, the maximal analgesic response for DFNZ was attained at a dose ~100 times higher than that of FNZ. For CFA experiments DFNZ was given at a dose only 3 times higher than FNZ, and the DFNZ response was smaller than that of FNZ. This choice of doses does not allow to determine if DFNZ is as effective as FNZ to treat subacute inflammatory pain. If DFNZ is not fully effective in subacute/chronic pain models there is risk that dose escalation in search of analgesia will result in doses beyond safety levels. Can the authors show that DFNZ is as effective as the parent compound FNZ in models of subacute/chronic pain?

BRAIN PENETRATION

1) Authors reasoned... "should DFNZ have high brain penetrance, metabolized [18F]DFNZ would be visualized in the brain after [18F]FNZ injection. However, no specific binding was observed in the brain starting at ~10 min after [18F]FNZ injection, indicating DFNZ has impaired brain penetrance" (pg 7).

Relative to this statement, it would be appreciated if the authors clarified whether at the dose that the parent radiotracer [18F]FNZ was injected (0.2 micrograms/kg) its metabolism produces sufficient accumulation of [18F]DFNZ to be detected in the brain within the timeframe of brain measures taken. Is it possible to provide actual [18F]DFNZ data rather than speculate on this point?

2) In relation to figures 4c-d and the statement given by the authors in pg 7: "In contrast, DFNZ is a substrate only for MATE1 (Fig. 4d) which is highly expressed at the BBB23-25":

a) What does % inhibition mean in figure 4c and 4d? What does the dotted line represent in these figures?

b) The assertion that MATE1 is highly expressed in BBB is not supported by the references provided in the manuscript. In particular, in figure 3 of REF #24 it is shown that the MATE1 protein is not expressed in the brain though present in kidney, heart, stomach. REF #25 is a review that indicates that BBB expression for MATE transporters is quite low and data is quite variable. The cited article questions MATE1 function at the BBB indicating that it needs clarification. The most abundant solute carriers in the BBB are Pg-P and BRCP, with lower levels of OCTs. Loperamide, which was mentioned by the authors as an opioid with low brain accumulation, is extruded from the brain by the transporter with highest expression levels, namely Pg-P (doi: 10.2174/156802611795371288). The authors should clarify how they believe the stated selectivity for MATE1 determines effective extrusion and reduced brain accumulation of DFNZ given the very low/absent expression levels of this transporter in BBB.

3) The description of how brain/plasma concentrations support higher ratios for DFNZ than for FNZ is quite clear. However, at the doses tested (0.03 mg/kg FNZ and 0.3 mg/kg DFNZ) analgesia by FNZ is maximal and that DFNZ's is submaximal (Figs 3b and 3c) Despite the submaximal analgesic response brain concentration of DFNZ is 13.5 fold higher than that of FNZ ($12.24/0.91 = 13.5$). Both ligands have similar binding affinity for MOR (as suggested by kinetic binding assays) so even at a less effective analgesic dose occupation of central MORs is proportionally >10 fold higher for DFNZ than FNZ. These figures do not seem warrant lower, safer, lower brain levels of DFNZ even at doses that produce submaximal analgesia by this compound.

4) In relation to statistical comparisons in extended figure 10 e: The authors used paired t test for comparison of caudate putamen to cerebellum in three independent experiments. It is conventional to use non-parametric tests (Mann Whitney) for $n < 5$. This should be corrected.

RESPIRATORY DEPRESSION, BRAIN HYPOXIA, WITHDRAWAL SCORES, ANALGESIC TOLERANCE AND HYPERSENSITIVITY

1) For oxygen saturation studies shown in figure 5a DFNZ was administered at a dose that corresponds to its analgesic ED50 in the hot plate test, FNZ was given at twice its analgesic ED50 and morphine was given at five times its analgesic ED50 (Note: an ED50 for morphine of 4 mg/kg was obtained from REF19, where one of the the co-authors of the current study used the same methodological approach as the one applied in the current manuscript to evaluate hot plate analgesia). Meaningful conclusions with respect to opioid safety require drugs to be at least compared at equianalgesic doses, to show that DFNZ is effectively safer than the other two drugs at doses that similarly alleviate pain. Without this constraint, the data presented in the figure shows that DFNZ, given at lower doses than the other two drugs, does not produce hypoxia. To ensure a trustworthy safety profile, it is preferable that drugs be tested at concentrations that produce maximal possible analgesic effect and not only at ED50.

2) Same comment as above for figure 5b: to ensure DFNZ safety, doses that produce maximal possible analgesic effect should be preferably tested. When FNZ was tested at a dose that reached the fully effective analgesic response (0.3 mg/kg), its hypoxia response was comparable to fentanyl. None of the DFNZ doses monitored in this test produce maximal effective analgesia.

3) Figure 5d: the treatment regimen that was used to induce morphine withdrawal included escalating doses that ranged from 5 to 20 times the analgesic ED50. Withdrawal for FNZ and for DFNZ was induced with a fixed dose regimen using doses that respectively represent 6-fold and actual analgesic ED50. This type of comparison does not provide adequate information to evaluate withdrawal relative to morphine. Comparable withdrawal induction schemes should be used.

4) Figures 5e-5g: It was not clear from the results and methodological sections whether von Frey filaments were applied on naïve animals or on animals that had been previously subject to any type of manipulation to induce pain hypersensitivity. Von Frey filaments are usually applied when there is a context of hypersensitivity such as allodynia. The rationale for similarly measuring nociceptive tolerance and hypersensitivity with von Frey filaments, particularly if this was done on naïve animals,

should be explained in the methodological section. experimental data should be presented as actual withdrawal threshold as was done in figures 4f and 4i for the analgesic responses. Actual threshold values would allow the reader to gain insight into the pressure that needs to be applied to evaluate nociceptive tolerance and hypersensitivity.

5) In figure 5h the authors show a downregulation of DAMGO binding sites by fentanyl but not by FNZ nor DFNZ. In the abstract the authors state: “DFNZ does not induce MOR desensitization after repeated exposure.” It is important to distinguish desensitization from downregulation. The manuscript does not provide experimental evidence indicating that DFNZ fails to induce functional desensitization of MOR.

6) In figures 5i-5j the use of a range of doses that ensure effective analgesia by FNZ (0.01-0.017 mg/kg) and DFNZ (0.3-1 mg/kg) indicated that both ligands can substitute for heroin in self administration trials. The authors concluded that this observation is consistent with reinforcing effects for both of the nitazene derivatives (pg 11). However, to dispel the notion that DFNZ may have significant reinforcing properties the drug was tested in a second series of trials where it was administered by i.v. infusions of 0.03 mg/kg/infusion. Knowing that analgesic ED50 for DNFZ in hot plate is 10 times this dose, how safe is it to conclude that this drug has weak reinforcing properties when used therapeutically?

HYPERLOCOMOTION AND DOPAMINERGIC NEUROTRANSMISSION

1) In figure 6a-c locomotor activity (over 30 min) was evaluated for DNFZ at 0.3 mg/kg (which corresponds to analgesic hot plate ED50), FNZ was evaluated at 0.03 mg/kg (6 fold analgesic hot plate ED50), fentanyl at 0.1 mg/kg (5 fold analgesic hot plate ED50) and morphine at 20 mg/kg (5 fold analgesic hot plate ED50). At these doses DFNZ was the only drug that failed to induce locomotion

However, in extended data figure 9, it can be seen that within the first 30 min post injection (similar time frame as shown in figure 6c) 3 mg/kg of DFNZ (corresponding to 10-fold the ED50 for hot plate analgesia) produced similar locomotor activity as 30 mg of morphine (corresponding to 7.5-fold the the ED50 for hot plate analgesia).

These two sets of data clearly show that unique 30 min locomotion profile described for DFNZ in figure 6 is dose dependent, and disappears at higher doses tested in extended data figure 9.

2) Same low doses as the ones used to evaluate locomotion in figure 6a-c were used in figure 6d-e to conclude: “Overall, fentanyl, morphine, and FNZ mimicked the effect of classical MOR agonists on dopamine transmission in NAcc. In contrast, DFNZ was ineffective, providing additional evidence for the unique behavioral and physiological effects of this new drug”. Effects of doses similar to those used in extended data figure 9 should be explored.

Referee #4

(Remarks to the Author)

This elegant study reports the development and characterization of two novel mu opioid receptor (MOR) agonists, FNZ and DFNZ, derived from the nitazene scaffold. Conducted by leading experts in the field, this work provides promising opioid candidates, that produce strong analgesic effects with minimal side effects, to the scientific and biomedical communities.

This study focusing on mu opioid receptor agonists is highly relevant, given their central role in pain management and their significant societal impact due to the risk of abuse. By demonstrating potent analgesia with reduced adverse effects of FNZ metabolite, DFNZ, represents important advances in opioid pharmacology and hold potential for safer therapeutic applications.

In brief, the authors demonstrate that FNZ and its metabolite, DFNZ, exhibit high affinity and plausible selectivity for the mu opioid receptor (MOR) in cell assays. They further characterized the FNZ binding site within the MOR-Gi protein complex using cryo-EM at a 2.3 Å resolution. The study also explored the spatiotemporal signaling profiles of FNZ and DFNZ, revealing that DFNZ displays a distinct signaling bias—favoring Go over β -arrestin 2—as well as slower receptor internalization and a lower endosomal trafficking.

As anticipated, both FNZ and DFNZ produced robust analgesic effects in rodent models of acute pain (hot plate test) and chronic pain (complete Freund’s adjuvant model assessed via Hargreaves and von Frey tests). Crucially, DFNZ showed reduced brain penetration in both mice and rats compared to FNZ, suggesting a lower potential for central side effects. Consistent with this, DFNZ was associated with significantly attenuated adverse effects, including the absence of respiratory depression, no tolerance or hypersensitivity, reduced physical withdrawal, and rapid extinction with no reinstatement of drug-seeking behavior. Mechanistically, the authors show that unlike classical opioids such as morphine and fentanyl, DFNZ does not increase dopamine release in the nucleus accumbens and does not alter MOR density in this brain region. Additionally, they report that DFNZ exhibits reduced efficacy at MOR-Galanin 1 receptor heteromers in vitro.

Taken together, DFNZ appears to be a MOR agonist that combines strong analgesic efficacy with a reduced impact on receptor trafficking and brain penetration, resulting in fewer side effects—including no respiratory depression, no tolerance, and a reduced risk of opioid use disorder. In conclusion the present study supports the strong therapeutic potential of DFNZ. Overall, the manuscript is clearly written, with a nice experimental design, and the data are well presented. However, we believe that certain methodological details warrant further discussion, and one or two additional key experiments may be required to fully support the study’s conclusions.

Comments:

-The authors, characterized nicely the structure of FNZ with MOR-Gi complex at a 2.3 Å resolution (Figure 1k). Is there a reason, why the authors did not characterize the structure of DFNZ with the MOR-Gi complex? as DFNZ seems the best promising compound, it may be important to determine whether the binding site is the same of FNZ.

-The authors nicely demonstrated the selectivity for MOR for FNZ and DFNZ competitive binding screen using rat brain membranes (Figure 1b). To confirm the selectivity of FNZ and DFNZ compounds, it will be important to check it in wild-type and MOR knockout animals. We believe that it will be important to realize 3H FNZ and 3H DFNZ autoradiography in wild-type and MOR knockout animals.

-Similarly, authors reported a strong analgesic effect of FNZ and DFNZ (Figure 3). A control experiment using MOR knockout mice would be critical to confirm on-target effects in vivo.

-The authors used both rats and mice, which is interesting and might strengthen the conclusions (Figure 3). Furthermore, in rats, both males and females have been used. This aspect is crucial, given the sex differences observed in both pain perception and addictive-like behaviors. Yet, the author never mentioned if they observed any sex difference in the behavioral paradigms they used. This is very likely due to the low number of rats used. It might be important to increase the number of rats and analyze more precisely the potential sex differences. Then, for mice, only males have been used. For such an important topic it appears of crucial interest to include both sexes.

-While the authors evaluated the effect of FNZ and DFNZ on catalepsy and hyperlocomotion in rats and mice, respectively, it appears intriguing not to see a justification of their behavioral design when assessing pain and addictive-like behaviors. It is very surprising to see that for rats they used a test where an absence of movement could be interpreted as analgesic properties of the compound, while they show catalepsy induced by both compounds. Indeed, the hot plate results in rats show a strong increase in the latency to jump or lick the paw, interpreted as an analgesic effect of the compound, yet, this effect is observable at the same timing as what the authors described as catalepsy, for the same doses. In other words, it seems that this increased latency might be simply due to the FNZ and DFNZ-induced catalepsy and has nothing to do with analgesia. Could the authors precise the design and 1- show the raw data (in sec) for the hot plate and 2- give insight about the rats' ability to move after FNZ and DFNZ injections.

-Then, in regard to analgesic tests with mice, the authors described an hyperlocomotion. This effect is weaker than the one provoked by morphine but still it is present. Could the author precise their design and clearly rule out any potential effect on locomotion that could interfere with the data obtained while evaluating nociception.

One option could be to evaluate the analgesic effect of both compounds with a test that does not require movements (grimace scale, physiological variables modified after pain...). Another control option, may be to test the effects of DFNZ on hotplate in mice and on CFA/Von Frey in rats.

-One claim of this article is that these compounds have little if no addictive potential. Indeed, rats present a rapid extinction and no clear relapse has been observed. However, these animals still work for the compounds (FR1, FR3 and PR) and show a classic inverted-U dose response curve (Figure 5k-n). These results suggest rewarding properties. Since these FNZ and DFNZ are MOR agonists, it appears crucial to test these "rewarding" properties in animals suffering from chronic pain (For ex CFA) and see if these individuals might develop addictive-like behaviors.

-Finally, the absence of NAc dopamine release after DFNZ injection is surprising (Figure 6) as DFNZ was shown to recruit Gi, and it is well known that the increased dopamine released from opioid is caused by the inhibition of GABAergic interneurons of the VTA. It could be interesting to monitor the calcium activity of dopaminergic neurons in the VTA to evaluate whether this specific compound does not require dopamine signaling at all or if instead it specifically prevents the dopamine release within the NAc. Another option may be to test the effect on NAc dopamine dynamics of a higher dose of DFNZ which produce a high level of DFNZ in the brain.

Minor comments:

-In figure 4 f. the two levels of green are not easy to distinguish.

-In Figure 4 h-l, it is not clear whether the dose of 0,3 and 1 mg/kg were tested for FNZ. It may be useful to change the graphical representation of these data.

Referee #5

(Remarks to the Author)

I co-reviewed this manuscript with one of the reviewers who provided the listed reports.

Version 1:

Reviewer comments:

Referee #3

(Remarks to the Author)

I have examined the revised version of the manuscript submitted by Gomez et al. where the authors have replied to the different concerns raised upon their first submission. I thank the authors for their clear responses and the experiments that were added to substantiate their point of view. Many points have been resolved but issues remain concerning dosing in relation to potential for respiratory depression, assessment of brain penetration at therapeutic doses as well as the accuracy of the originality statement in relation to the mechanism supporting the extrusion of DFNZ from the brain. Below are my comments in relation to the changes done, addressing these points in particular.

PHARMACODYNAMIC PARAMETERS, SIGNALING BIAS AND TRAFFICKING PROFILES

The authors have provided satisfactory answers clarifying the use of different BRET techniques for G protein data gathered in figures 1 and 2 and have carried out new displacement experiments to correct previously unexplained differences in MOR affinity. Also, I fully appreciate and agree with the decision of de-emphasizing any claim that G protein over barr2 bias may underlie DFNZ's safety profile.

In this sense I would like to point out that although the authors indicate that preferential G-protein signaling has not been linked to greater chance of overdose, we also know that G-protein mediated signaling is directly implicated in respiratory depression, which is the main cause of death by opioid overdose (nitazenes included). Respiratory depression by opioids involves reduced excitability of respiratory brainstem nuclei, including MOR-mediated hyperpolarization of neurons in the preBöttinger complex and MOR-mediated reduction of glutamate release onto these neurons. These effects involve G - mediated regulation of membrane channels while inhibition of cAMP by Gi/o proteins reduces excitability mediated by sodium-channels. Thus, upon its penetration into the brain, DFNZ activation of MOR receptors on the respiratory nuclei will preferentially engage the pathways that mediate respiratory depression.

That being said, the data are acceptable to show differences between FNZ and DFNZ and these differences are further supported by APEX data.

BRAIN PENETRATION

In the previous submission the authors had obtained in vitro data that ruled out a role of P-gp/Bcrp in DFNZ efflux from the brain and had proposed a role for MATE1 in this effect. Following re-assessment of the literature the authors carried out alternative in vitro assays to evaluate the effect of DFNZ on these targets. They now find that DFNZ is indeed a substrate for P-gp/Bcrp, but not MATE1 (n=2). They also used transporter knockout mice to show that at an in vivo dose of 100 micrograms/kg of [3H]DFNZ (s.c.; n=2) was extruded from the brain parenchyma by the P-gp/Bcrp transporters. The authors further show that rats that received 100 micrograms (i.v.; n=8) of DFNZ the inhibition of P-gp by tariquidar precipitates ~15% decrease in O2 saturation, thus linking P-gp-mediated extrusion of DFNZ to the safety of low doses of the drug.

It was not clearly explained why the in vitro assays used in the two different occasions produced opposing results. However, the in vivo data support the latest in vitro results directly implicating P-gp/Bcrp in the exclusion of low doses of DFNZ from the brain.

These new findings indicate that DFNZ shares the exclusion mechanism of the prototypical P-gp substrate loperamide, which is also a MOR super-agonist (e.g.: PMID: 41193810).

Given this precedent, the authors may want to reformulate the statement "a MOR super-agonist with limited brain penetrance, such as DFNZ has not been previously reported" and provide information on how DFNZ differs from loperamide, a previously described super-agonist with limited brain penetrance.

BRAIN EXCLUSION AND DFNZ SAFETY

MICE

The authors showed that a subcutaneous dose of 100 micrograms/kg [3H]DFNZ given to mice does not induce accumulation of the drug in brain parenchyma, remaining within the ventricles. They have also carried out extra experiments showing that higher DFNZ doses (1 mg/kg, s.c.) than the ones previously tested (0.3 mg/kg, s.c.), produces fully effective analgesia in the mouse model of inflammatory pain (CFA).

There was no exploration of how this fully effective therapeutic dose affects brain accumulation of DFNZ. This information is crucial not only in trying to differentiate this novel ligand from loperamide but in establishing the doses that warrant therapeutic actions with limited brain penetration. Whether it is via use of [3H]DFNZ or displacement of [3H]DAMGO the authors have the means to do it.

Although there is no direct data on how therapeutic doses of DFNZ accumulate in the brain, the functional data provided by the authors indicates that a fully effective analgesic dose of 1mg/kg of DFNZ can produce maximal central effects in the reward system, implying sufficient penetration to produce similar effects as fentanyl also given at an analgesic dose.

In particular, I am referring to the fact that 1 mg/kg, s.c. of DFNZ produces Ca²⁺ mobilization in NAcc dopamine terminals reaching similar maximal peak as a dose of 3 mg/kg of the drug (Figure 6b), which in turn surpasses the effects caused by 0.3 mg/kg of fentanyl (Supplementary Figure 10 a). Therefore, at the fully effective analgesic dose of 1 mg/kg brain penetration is sufficient to support significant effects of DFZN in the reward circuit, and in this case the effects are greater than those of brain penetrating fentanyl. How can we be sure that central MOR on the respiratory nuclei are spared from activation if those in the reward system are? The authors did not consider assessing the scenario where brain penetration of high doses of this lipophilic nitazene overrides the extrusion mechanism that is operant at lower doses.

I would have preferred the use of a greater dose range of DFNZ to directly evaluate brain penetration. In absence of such data it is difficult to accept the conclusion that "CNS penetration of DFNZ is limited", particularly at therapeutically relevant doses.

RATS

New data obtained in rats indicate that 300 micrograms/kg of DFNZ given subcutaneously roughly produce 40% occupation of DAMGO labelled sites in NAcc, indicating partial occupation of central MORs at this submaximal analgesic dose. At a fully effective analgesic dose of 1 mg/kg, occupation was not evaluated. I can only reiterate the need for exploration of a full

dose range to evaluate occupancy at fully effective analgesic doses..

HYPOXIA

Pulse oximetry measures reported in the previous version of the manuscript had evaluated oxygen saturation following subcutaneous administration of submaximal analgesic doses of DFNZ. The authors have now abandoned this approach on the basis of its high variability in freely moving animals. This precludes the evaluation of whether fully effective subcutaneous analgesic doses of DFNZ, which we know can penetrate the brain to induce significant effects in the reward system, do actually spare the activation of central MORs in respiratory nuclei.

Nonetheless the authors use an electrochemical approach to evaluate oxygen saturation. These estimates are more precise than pulse oximetry but require i.v. administration, preventing the direct assessment of the respiratory effects of the subcutaneous DFNZ doses that were used to evidence its analgesic effects.

The i.v. doses of DFNZ that were tested for hypoxic effects are 100 and 300 micrograms/kg. The authors argue that 300 micrograms/kg of DFNZ given i.v. should be considered equivalent to 1 mg/kg of DFZ given s.c., and we should take these results as evidence of DFNZ safety.

Because we are speaking about safety of a super-agonist that is biased towards a signaling path that directly engages the mechanisms of respiratory depression, and since functional data indicate significant brain penetration at therapeutic doses, the speculation on dose equivalence is not re-assuring. Exploration of how a larger range of DFNZ doses induces hypoxia would have been preferred. If DFNZ is meaningfully excluded from the brain despite dose escalation, its safety should be revealed by a ceiling effect on hypoxia. This or any similar type of evidence was not provided.

DOPAMINERGIC NEUROTRANSMISSION

The authors now provide evidence that similar to the highly efficacious agonist methadone but unlike the weaker morphine, DFNZ (1 mg/kg) activates the MOR-Gal1R heterodimer to reduce DA release in the VTA.

(Remarks on code availability)

N/A

Referee #4

(Remarks to the Author)

Authors did a great job, adding several experiments and thoroughly addressing most of our concerns. However, we still have three comments:

- Regarding nociception, authors claimed to have tested the effect of DFNZ in WT and KO animals in the hot plate test in extended data 6. We found the data for locomotion in WT and KO animals but not the data for the hot plate test. Could the authors clarify or add the experiment to study the effect of DFNZ on analgesia in WT and KO MOR animals? It would be crucial to demonstrate that the DFNZ analgesic effect is mediated via MOR.

- We appreciate the authors' explanation regarding the challenges in establishing pain-induced increases in opioid self-administration, especially in such labs with leading experts in the field. However, our concern is not solely about a potential escalation of intake under pain conditions but rather about whether FNZ and DFNZ possess different reinforcing properties in animals with chronic pain. Although the authors argue that pain is unlikely to alter self-administration, the current data still indicate that the compounds maintain operant responding (FR1, FR3, PR) and produce a classic inverted-U dose-response function, which is consistent with rewarding effects.

To more directly address the claim that these compounds have little or no addictive potential—particularly given their MOR agonism—we suggest performing a conditioned place preference assay across a range of doses in both control and CFA-treated mice. We would also see whether the U-inverted curve is still observable and, more importantly, if it is shifted in animals with pain conditions. It would also be important to determine that this rewarding effect is absent when MOR is deleted. CPP in WT and MOR KO animals is less labor-intensive than self-administration procedures and would provide complementary evidence regarding the reinforcing or motivational properties of FNZ and DFNZ in the context of chronic pain. Such data would help strengthen the conclusions about the compounds' abuse liability.

- In panel 5d, the statistical reporting is unclear. It is not evident what the asterisks refer to—specifically, which groups are being compared when 5 stars are written, and the legend does not describe the significance.

(Remarks on code availability)

We detected nothing wrong in the code.

Referee #5

(Remarks to the Author)

I co-reviewed this manuscript with one of the reviewers who provided the listed reports.

(Remarks on code availability)

Version 2:

Reviewer comments:

Referee #3

(Remarks to the Author)

I have examined the revised version of the manuscript submitted by Gomez et al. this last January.

The authors have carried out new experiments eloquently showing that fully effective analgesic subcutaneous doses of DFNZ have limited brain penetration and do not produce hypoxia.

I have no outstanding issues regarding the study.

I would also like to congratulate the authors on their discovery.

(Remarks on code availability)

Referee #4

(Remarks to the Author)

The authors have satisfactorily addressed our previous concerns and have done an excellent job overall. We have only a few remaining minor comments:

1. Pain assay in Oprm1 KO animals:

For clarity, we suggest that the authors specify in Extended Data Figure 6, panel i the following treatment groups: Oprm1 KO–FNZ (0.1 mg/kg) and Oprm1 KO–DFNZ (3 mg/kg) instead of FNZ 0.1 mg/kg and DFNZ 3 mg/kg.

2. Assessment of misuse liability in pain conditions:

We thank the authors for adding a discussion addressing the risk of misuse in individuals suffering from pain. Regarding the suggested conditioned place preference (CPP) procedure, we proposed an alternative, simpler approach, given that the authors indicated intravenous self-administration under pain conditions is technically challenging at the first revision.

Our original objective was to evaluate self-administration, extinction, and drug-induced reinstatement in a chronic pain model; however, the authors declined this approach due to the extensive workload it would entail. We nevertheless believe this question remains important, as these compounds are not intended for recreational use but rather as analgesics. Thus, although FNZ and DFNZ do not appear to confer abuse liability in pain-naïve animals, they are likely to be used, and potentially misused, by individuals experiencing pain. Assessing abuse potential in this clinically relevant context would therefore strengthen the translational impact of the study.

3. Statistical annotation in Figure 2e:

In Figure 2e, we suggest replacing the asterisk (*) with a different symbol (e.g., # or &) to indicate differences between FNZ and DFNZ relative to morphine (bottom row), in order to avoid any potential confusion to have a same sign for two conditions.

(Remarks on code availability)

No issues detected.

Referee #5

(Remarks to the Author)

I co-reviewed this manuscript with one of the reviewers who provided the listed reports.

(Remarks on code availability)

11-23-2025

This letter is in response to the request for revision of our manuscript (2025-04-09979), entitled "A μ opioid receptor superagonist that produces analgesia with minimal adverse effects". We thank the reviewers for their enthusiasm, positive evaluation, and constructive comments, which have considerably strengthened our manuscript.

The revised manuscript includes additional characterization of FNZ and DFNZ pharmacology, brain distribution, hyperlocomotion, analgesic efficacy, respiratory depression, and dopamine neurotransmission. We now provide mechanistic insights into DFNZ's impaired brain penetrance, weak respiratory effects, and weak reinforcement profile.

Below we address the reviewers' specific comments in blue font. We also highlight new text in the revised manuscript using tracked changes.

The additional experiments we performed are listed below:

1. [3 H]FNZ and [3 H]DFNZ autoradiography in *Oprm1* knockout mice (**Extended Data Fig. 2**).
2. *In vivo* studies (hyperlocomotion, hot plate) of FNZ and DFNZ in *Oprm1* knockout mice (**Extended Data Fig. 6**).
3. Additional testing of FNZ and DFNZ in mouse CFA chronic pain experiments (**Fig. 3**).
4. MOR occupancy assessment by FNZ and DFNZ in rats (**Fig. 4** and **Extended Data Fig. 7**).
5. *Mate1*, P-gp, and Bcrp transporter substrate assays (**Fig. 4** and **Extended Data Fig. 7**).
6. [3 H]DFNZ brain uptake in wild-type and P-gp/Bcrp knockout mice (**Fig. 4** and **Extended Data Fig. 7**).
7. Brain hypoxia experiments using DFNZ with tariquidar (P-gp inhibitor) pretreatment (**Fig. 5**).
8. Effects of FNZ, DFNZ, morphine, and fentanyl on NAc dopamine neuron activity using GCaMP6s fiber photometry (**Fig. 6** and **Extended Data Fig. 10**).
9. Additional testing of FNZ, DFNZ, morphine and fentanyl using dLight1.3b photometry (**Fig. 6** and **Extended Data Fig. 10**).
10. Dopamine microdialysis studies in rats using MOR-Gal1R targeting peptides (**Fig. 6** and **Extended Data Fig. 10**).

Sincerely,

Michael Michaelides, Georgios Skiniotis, and Kenner Rice (on behalf of all coauthors)

Referees #1 & 2 (Remarks to the Author):

Gomez and describes a newly discovered agonist for the human mu-type opioid receptor and present a series of studies demonstrating its efficacy and minimal adverse effects in pain treatment. The authors obtained a cryo-EM structure of the MOR–Gi protein complex bound to this agonist and compared the binding pockets across different opioid receptor types to assess the agonist's selectivity.

This technical assessment is focused on the cryo-EM aspects of the work.

The cryo-EM maps were refined to high resolution, and the atomic model fits well within them, with both possible agonist conformations included. The supplementary figure related to the cryo-EM analysis provides detailed information on data processing and evaluations of the maps and model. Overall, the data are of high quality, and the conclusions are well supported.

We thank the Reviewers for their positive comments.

Several points should be addressed:

1. The notation for amino acid residues is confusing and we could not understand it (e.g., D3.32, I7.39). Our expectation is that D3 refers to Asp3 (Aspartate 3) and I7 refers to Ile7 (Isoleucine 7) but this does not appear to be the case. Please clarify the labelling of amino acid residues in Fig. 1L

The notation used follows the Ballesteros-Weinstein numbering system for GPCRs, where the first number denotes the TM helix, and the second number indicates the residue's position relative to the most conserved residue within that TM, defined as .50. For example, D3.32 refers to the Asp located 18 residues N-terminal to the most conserved in TM3, R3.50. We clarified this in the text (**page 4**) and in the **Fig. 1** legend and included the relevant citation.

2. The cryo-EM figures are rendered beautifully, but the text size could be increased for clarity. In addition, if the FNZ molecule is modelled in two different conformations, perhaps Figure 1K should show both conformations, and include a corresponding callout in the text.

We thank the reviewer for pointing this out. We increased the font size in addition to adding both conformations in the revised **Fig. 1** as well as a callout in the text.

3. Although compelling map density is shown for FNZ, map density should be shown somewhere (e.g. in an extended data figure) for the amino acid side chains depicted in Fig. 1L to convince the reader that the orientations of the side chains in the model are reliable.

We added map-model agreement figures for each receptor helix and the ligand to **Extended Data Fig. 4**.

4. In Figure 1L, the D3.32 label is near where Q2.60 is, and be closer to the actual side chain.

We thank the reviewer for pointing this out. We adjusted the position of the label.

5. Features discussed in the text should be clearly highlighted in the figures. For example:

a. Figure 1K should highlight the FNZ binding site (the main takeaway from the figure) with a circle or box in the composite map.

We added the inset box to revised **Fig. 1**.

b. TM2 and 6, which are discussed in the text, should be labeled in Figure 1K or a new supplementary figure (with an appropriate figure callout).

We added labels to the transmembrane helices in **Fig. 1L**.

c. Residues like I7.39 and Y1.39 are discussed in the text but not shown in Figure 1L, raising questions about their relevance or importance.

We now show these residues in **Extended Data Fig. 4**.

6. A representative micrograph and example 2D class average images should be shown (e.g. in Extended Data Fig. 6).

We added a representative micrograph and example 2D class averages to **Extended Data Fig. 4**.

7. There is a discrepancy in the chain ID of FNZ in the model: it appears as chain D in the atomic model provided to us, but is labeled as chain R in the PDB report. This might have been corrected already, but please double check.

We checked and the residue has been relabeled as chain R in the PDB deposition process; the structure should otherwise be identical to the one provided.

8. In the cryo-EM statistics table (currently part of Extended Data Fig. 6):

- under-focus values should be positive numbers with standard expressions for the CTF.
- B factors used for sharpening should be negative
- The cutoff for assessing the resolution of a noise-free model versus the map from the full dataset should be at FSC=0.5 not FSC=0.143 (FSC=0.143 is the cutoff used for half-map versus half-map).
- Please also include the PDB code in the Table, not just the EMD codes

These have all been corrected in **Extended Data Fig. 4**.

Referee #2 (Remarks to the Author):

I co-reviewed this manuscript with one of the reviewers who provided the listed reports.

We thank the Reviewer for reviewing our manuscript.

Referee #3 (Remarks to the Author):

Summary of key results

The manuscript entitled “A μ opioid receptor superagonist that produces analgesia with minimal adverse effects” reports on two fluorinated nitazenes: fluornitazene (FNZ) and its primary metabolite N-desethyl-FNZ (DFNZ). The authors studied pharmacodynamic, pharmacokinetic and behavioural response profiles of both of these compounds, comparing them to DAMGO or to opioid analgesics such as morphine or fentanyl. They conclude that DFNZ is a MOR agonist with supra maximal intrinsic efficacy (superagonist) which preferentially activates G protein signaling over β -arr recruitment. They speculate that this unique bias profile in association with low brain accumulation and reduced signaling efficacy at the MOR-Gal1R heteromer may explain in vivo responses observed for DFNZ. The authors have interpreted DFNZ’s in vivo response profile as an indication that this ligand behaves as an analgesic that does not produce respiratory depression, tolerance nor withdrawal. They also propose that DFNZ has weak reinforcing properties and fails to modify dopaminergic transmission in the NAcc of the reward circuit. Based on these results the authors challenge the notion that high efficacy MOR agonists are not suitable for development as safe analgesics.

Originality, significance

The subject of the study is very important and timely, given the current opioid overdose crisis. In this context, efforts to develop safer opioid analgesics warrant serious consideration. The techniques that were used to characterize pharmacodynamic, pharmacokinetic and behavioural

response profiles are state of the art. The notion that superagonists of the MOR receptor could lead to development of therapeutically safe opioid analgesics is certainly thought provoking. On these bases the study merits consideration. However, my main concern is that the study falls short of demonstrating DFNZ's safety.

We thank the Reviewer for their positive comments, and we provide clarifications and additional data below that addresses the Reviewer's concern.

Main concerns

With safety in mind, it is imperative to show that, given at similarly effective analgesic doses as currently available painkillers the superagonist DFNZ has a better profile of respiratory effects and reduced reinforcement properties. In the present study the side effect profile of DFNZ was most frequently (if not systematically) established using doses whose analgesic effects were less than those of the parent compound (FNZ) and that of current therapeutic standards (morphine, fentanyl). I have detailed below the examples where I think low effective doses of DFNZ are at the bases of the claimed safety of this novel product and have suggested possible experiments to address this issue.

We appreciate the concern. We addressed the respective examples that the Reviewer noted in detail in the comments below.

Another important concern involves pharmacodynamic measures that support the claim for biased signaling. There is an incongruence in EC50 values presented for GalphaoA in figures 1 and 2. Detailed comments on this point follow below.

We have responded to this concern in detail in the respective comments below. We now show congruence between the functional assay data in **Figs. 1 and 2**.

Finally, no actual experimental verification was provided to substantiate that either bias, low brain accumulation or reduced efficacy at the MOR-Gal1R heteromer were responsible for improved side effects profile of DFNZ at analgesic doses. A mechanistic verification that any of these mechanisms is actually involved in supporting a safe respiratory profile for DFNZ would considerably strengthen the manuscript.

We now include in the revised manuscript experimental and mechanistic verification that:

- (i) P-gp- and Bcrp-mediated transport of DFNZ mediates its weak brain penetrance (**Fig. 4 and Extended Data Fig. 7**). For this, we used established *in vitro* assays to show that DFNZ is a transportable substrate for P-gp and Bcrp. We then performed *in vivo* autoradiography studies of [³H]DFNZ brain uptake in wild-type and P-gp/Bcrp knockout mice. These studies show that DFNZ has impaired brain penetrance in wild-type mice but high brain penetrance and widespread brain accumulation in knockout mice.
- (ii) DFNZ's interaction with P-gp regulates its safe respiratory profile (**Fig. 5**). For this, we show that pretreatment with tariquidar, a P-gp inhibitor, produced robust and sustained brain hypoxia in rats that were co-injected with an analgesic dose of DFNZ that, in the absence of tariquidar, did not produce brain hypoxia.
- (iii) DFNZ's reduced efficacy in the VTA is dependent on MOR-Gal1R heteromerization (**Fig. 6 and Extended Data Fig. 10**). For this, we used the same methodology as in our previous studies to demonstrate that VTA MOR-Gal1R heteromers mediate the dopaminergic effects of opioids (PMIDs: 28007761; 30913037; 38145984). Specifically, we performed *in vivo* microdialysis studies of extracellular dopamine in rats injected with DFNZ and concurrently infused with specific heteromer disrupting TM peptides in the VTA. We previously demonstrated that only domains TM5 and TM6 of both receptors are involved in the MOR-Gal1R heteromeric interface (PMID: 35750299). We now show that an analgesic dose of DFNZ (1 mg/kg, i.p.) in rats produced somatodendritic dopamine release only after local VTA infusion of the TM5-targeting peptide but not the TM7 (control) peptide.

Detailed comments for the authors on data, methodology, statistics, robustness of conclusions and suggested improvements

PHARMACODYNAMIC PARAMETERS, SIGNALING BIAS AND TRAFFICKING PROFILES

1) Kinetic assays show that FNZ and DFNZ have similar MOR binding properties but K_i values for DAMGO displacement indicate >10-fold higher affinity for DFNZ than for FNZ. An explanation for these differences should be provided.

Because we synthesized FNZ and DFNZ at different times, the [3 H]DAMGO binding shown in the original **Fig. 1** was also performed at different times while using different membrane preparations and batches of [3 H]DAMGO. We now repeated these experiments by testing FNZ and DFNZ at the same time and using the same membrane preparation and batch of [3 H]DAMGO. The results from these new competition binding experiments agree with the results from the kinetic binding experiments and are shown in revised **Fig. 1**.

2) Figure 1c, 1e-1j: The legend indicates four replicates. Replicates usually refer to the number of repeats within one experiment. What is the number of independent experiments for the curves shown in these figures?

We believe that the Reviewer is referring to the legend of original **Fig. 2**, which mentioned that the kinetic functional assays were performed using at least 4 replicates. For the curves in **Fig. 1**, we performed at least 2-3 independent experiments per curve. We updated the Methods and respective figure legends accordingly.

3) Figure 2a needs to be corrected to indicate the use of BRET1 and BRET2 biosensors as described in methodological section. According to methods, the Galpha biosensor contains GFP2 and not Venus as indicated.

We apologize for the confusion. In the first submission, we inadvertently did not include the methods corresponding to the kinetic BRET experiments shown in **Fig. 2**. This is now corrected. To clarify, the assays in **Fig. 1** and **Fig. 2** were performed using different BRET approaches. Specifically, the BRET assays in **Fig. 1** used BRET2 for G-protein activation and BRET1 for beta-arrestin recruitment whereas the kinetic BRET assays in **Fig. 2** used BRET1 and Venus for both G protein activation and beta-arrestin recruitment.

4) Methods should include a section describing how DDLog and its compound error were calculated for figure 2f. As it stands the reader does not know whether bias measures were calculated from logistic or operational parameters.

These details are now included in the revised Methods section (**page 51**).

Do graphs in Figure 2b-2f correspond to representative examples? How many times were kinetic experiments repeated?

The graphs in **Fig. 2b-f** correspond to the averaged results. The experiments were repeated at least 10 times (i.e., 10 independent replications) for Go and 9 times (i.e., 9 independent replications) for beta-arrestin. We have added this information to the revised Methods section and the **Fig. 2** legend.

5) There is incongruence in EC50 data from curves in Figure 1h (and described in pg 3) and the pEC50 value shown in figure 2d.

i.e: according to the methodological section the curves and corresponding EC50 values in figure 1h were obtained 15 min after addition of the ligands. GalphaoA EC50 values obtained from these curves (described in pg 3) indicate that FNZ is more potent (6.6 fold) than DFNZ. In figure 2d the pEC50 values at the 15 min timepoint show that DFNZ is more potent than FNZ. This incongruence needs to be revised and explained since the authors conclude that higher DFNZ potency for the G protein response is what drives DFNZ's unique bias profile. This bias profile is suggested as a possible mechanism underlying beneficial side effects profile of DFNZ.

As mentioned in Comment #3 above, the BRET assays in Fig. 1 and Fig. 2 were performed using different labelled constructs and BRET approaches. BRET2 differs from BRET1 in the donor-acceptor spectral pair and substrate, resulting in differences in intensity and spectral separation. Accordingly, the EC₅₀ differences between the BRET data in Fig. 1 and Fig. 2 are due to the underlying methodological differences.

(a) Bias calculations from BRET assays shown in Figure 1. (b) Bias calculation for GoA-arr2 from the kinetic BRET assays shown in Figure 2.

Nevertheless, the greater relative G-protein activation of DFNZ compared to that of FNZ is supported by the results from both assays. Please see above the bias calculations for the BRET assays in Fig. 1 (panel a) and the same calculations for the kinetic BRET assays shown in Fig. 2 (panel b), demonstrating congruence in GoA-arr2. We now include the bias calculation data for the Fig. 1 BRET experiments in revised Extended Data Fig. 2.

6) The authors use BRET2 (GFP2, Coelenterazine 400a) to monitor G protein responses and BRET1 (Venus, Coelenterazine h) to monitor beta-arrestin recruitment. Luminescence by Coelenterazine 400a decays much quicker than that of Coelenterazine h, implying that the time frame for visualization of the G protein response is shorter than that of visualization of beta-arr recruitment. Note that the difference in time frame of effective energy transfer is not corrected by the ratiometric nature of the BRET response. BRET figures (2b-2f) provided by the authors show normalized rather than net-BRET values so we do not know how the actual G protein and beta-arr2 responses of the standard DAMGO evolve overtime. net-BRET data at least for the standard should be provided in supplementary information. I would also appreciate the authors' insight on how the use of BRET1 and BRET2 biosensors may have influenced their conclusions in relation to biased signaling.

We now provide the requested net-BRET data in revised Extended Data Fig. 5. The Reviewer is correct that in Fig. 1, coelenterazine 400a is used to monitor G protein processes whereas coelenterazine h is used for the beta-arrestin assays. However, in Fig. 2, the kinetic BRET experiments use BRET1 with Coelenterazine h for both assays, allowing for the Go and beta-arrestin2 experiments to be temporally comparable in terms of the reagent decay rate. While the overall net-BRET signal slightly decreases over time in the beta-arrestin2 assay, the overall trend in DFNZ shows a higher initial signal compared to DAMGO and a more pronounced decline over time, consistent with the normalized data.

7) The notion that G protein over beta-arr bias could yield safer opioid analgesics was based on initial observations by the Bohn laboratory showing that barr2-KO mice had reduced respiratory depression and less constipation when transgenic mice were injected with opioids (doi: 10.1021/acs.jmedchem.8b01136). These observations could not be reproduced by three independent laboratories (doi: 10.1111/bph.15004) and since then the notion that G protein signaling preference is a good predictor of opioid safety has been questioned (e.g.: doi: 10.1016/j.tips.2020.09.009).

The authors should provide rationalization of how avoiding beta-arr signaling may contribute to DFNZ's beneficial side effect profile and/or provide experimental evidence that bias is at the basis of DFNZ's safety. Note that a comprehensive study of nitazene biased signaling recently concluded that intrinsic efficacy of nitazenes in Galpha protein signaling is a common property underlying their high risk of overdose (doi: 10.1021/acscchemneuro.3c00750). It is also worth noting that brorphine, another highly efficacious MOR agonist that was developed as a safer, G

protein-biased, opioid analgesic (doi: 10.1021/acs.jmedchem.8b01136) was at the basis of numerous overdose deaths (doi: <https://doi.org/10.1093/jat/bkab082>). How does the superagonist DFNZ differ from G protein-biased nitazenes and buprenorphine to be considered safer?

These are all great comments, and we apologize for any confusion. We did not explicitly state, nor do our data suggest that DFNZ avoids beta-arrestin signaling. Our intent was to describe the unique pharmacological differences between FNZ and DFNZ. One of these differences involves their relative actions at G protein activation vs. beta-arrestin recruitment (e.g., DFNZ more potently activates G_o over beta-arrestin2 compared to FNZ). Given that we do not provide mechanistic evidence linking this preference to DFNZ's safety profile, we have de-emphasized any such claims. We also explicitly mention in the revised main text that DFNZ is highly efficacious at beta-arrestin recruitment (**pages 5 and 16**).

Regarding the question of how DFNZ differs from other G protein-biased nitazenes, the mentioned study concerning the risk of G-protein signaling (Tsai et al., PMID: 38345920) states that high G-protein signaling does not exclude the risk of overdose; however, that study does not explicitly link increased G-protein signaling to increased chance of overdose. Importantly, although the tested nitazenes in Tsai et al. had high G-protein signaling, all were considered arrestin-biased or balanced agonists. That said, we cannot predict how DFNZ would differ from other G-protein biased nitazenes that might exist because nitazenes show wide structure activity relationships and different functional profiles. Regarding buprenorphine specifically, a recent study (PMID: 32734307) could not reproduce the bias profile that was originally reported by the Bohn lab (PMID: 30199635). Also, using the same functional assays and analysis protocols described in Tsai et al., we found that both buprenorphine and SR17018 have balanced profiles (Tsai et al., unpublished data).

8) The methodological section indicates that E_{max} was measured as ... "tangent to the plateau of the dose-response curve". The maximal response elicited by a drug is measured by the SPAN = maximal asymptote – minimal asymptote. The use of SPAN allows to consider variations in basal activity (for example a partial agonist in an activated system can produce the same maximal asymptote as a full agonist). Also note that E_{max} should not be equated with efficacy since in amplified systems full and partial agonists produce maximal system responses. E_{max} values should be corrected to conform to E_{max} definition (i.e. SPAN).

We corrected the E_{max} values and added the SPAN calculations to the Methods section (**page 54**). Specifically, each reading was normalized by subtracting the min asymptote of the DAMGO positive control and dividing by the span of the DAMGO positive control (max asymptote-min asymptote).

9) In addition to G protein/beta-arrestin signaling, the authors provide a trafficking profile by FNZ and DFNZ. However, there is no further mention how this profile may mechanistically determine the beneficial side effect profile that the authors associate with DFNZ.

Also, the authors seem to associate DFNZ's trafficking profile to preferential G protein activation (pg 5). It should be noted that ligands like morphine or PMZ21 that are less efficacious than DAMGO showed as similar profile as DFNZ with less endosomal trafficking, stronger EYA4 interaction (considered by the authors as a surrogate of G protein activation) and weaker beta-arrestin2 labeling (doi.org/10.1038/s41589-024-01588-3). How do the authors rationalize the fact that DFNZ's trafficking profile, which they define as superagonist, resembles that of partial agonists. How do the observed trafficking properties contribute to beneficial side effects of DFNZ as compared to morphine whose trafficking it resembles.

Indeed, DFNZ's MOR trafficking profile is different from that of FNZ and DAMGO and instead resembles that of morphine and PZM21, which activate MOR with lower efficacy than DFNZ, FNZ, and DAMGO. One potential explanation for this is DFNZ's relative functional bias for G-protein over beta-arrestin recruitment, a characteristic shared by low-efficacy agonists and not by FNZ or DAMGO. However, DFNZ differs from morphine in other properties such as

pharmacokinetics, metabolism, efficacy at MOR-Gal1R heteromers (PMID: 30913037), and dopamine neurotransmission and therefore it is unclear to what extent DFNZ's trafficking properties contribute to its beneficial side effects as compared to morphine. As such, it is unlikely that any single property, such as trafficking, for example, can solely explain DFNZ's safety profile compared to FNZ, morphine, or other MOR agonists. Instead, our cumulative data suggests that DFNZ's safety profile may result from its unique overall pharmacological profile, which includes its distinct activation and trafficking profiles, its impaired brain penetrance, its distinct pharmacodynamic effects at MOR-Gal1R and unique effects on dopamine neurotransmission. We now discuss this on **pages 5 and 16-18**.

10) In the methodological section for APEX, the references provided do not match the reference list.

This has been corrected. Thank you.

ANALGESIA

1) Analgesic responses in the hot plate test were assessed in rats over a period of 3 h with the higher doses tested producing an analgesic response over this full time period. However, dose response curves only considered the first 60 min of analgesia, which inevitably skews ED₅₀ towards lower doses. Why was this done?

We chose to use the first 1-h time interval for dose-response calculations because most drug effects are waning after 1 h. We did this previously to examine other opioids (PMID: 37276825). Using the entire time-course (i.e., 4 h) will effectively "dilute" all drug effects, whereby no treatment group will show maximal effects, leading to an inaccurate ED₅₀ calculation.

2) Authors compare ED₅₀ values for analgesia, catalepsy and hypothermia without providing statistics (pg 5-6). Assessment of statistical significance of the distinct potencies for inducing desired (analgesia) and undesired effects (catalepsy, hypothermia) should be provided.

We appreciate the reviewer's concern in this matter. It is important to note that ED₅₀ potency values can be obtained for analgesia and catalepsy, but not for body temperature changes. The effects of FNZ and DFNZ on temperature are biphasic, with hypothermia only occurring at the highest dose tested for both drugs. Nevertheless, we used the "compare" feature in the GraphPad Prism non-linear regression menu to directly assess potential differences between LogEC₅₀ values for analgesia vs. catalepsy. In short, the analgesic potency is significantly different from cataleptic potency for both FNZ ($F[1,43]=21.83$, $p<0.001$) and DFNZ ($F[1,43]=41.96$, $p<0.001$). We now report this in the main text (**page 6**) and revised **Extended Data Fig. 6**.

3) In the CFA mouse model, a single dose was tested. How were these doses chosen? When one looks at hot plate tests in rats, the maximal analgesic response for DFNZ was attained at a dose ~100 times higher than that of FNZ. For CFA experiments DFNZ was given at a dose only 3 times higher than FNZ, and the DFNZ response was smaller than that of FNZ. This choice of doses does not allow to determine if DFNZ is as effective as FNZ to treat subacute inflammatory pain. If DFNZ is not fully effective in subacute/chronic pain models there is risk that dose escalation in search of analgesia will result in doses beyond safety levels. Can the authors show that DFNZ is as effective as the parent compound FNZ in models of subacute/chronic pain?

We now show in revised **Fig. 3** that DFNZ produces maximal analgesia and is as effective as FNZ in the CFA pain model. Specifically, we repeated the CFA experiments in mice using a lower FNZ dose (0.03 mg/kg) and a higher DFNZ dose (1 mg/kg). We now show that 0.3 mg/kg DFNZ produced equivalent analgesia in Hargreaves as 0.03 mg/kg FNZ whereas 1 mg/kg DFNZ produced maximal analgesia that was significantly greater than analgesia produced by 0.03 mg/kg FNZ. As before, responses were assessed at 1 h after FNZ and DFNZ injections indicating that the analgesic effects were sustained even though the drugs promoted weak and transient hyperlocomotion that lasted for less than 30 min (0.03 FNZ and 1 mg/kg DFNZ) or no

hyperlocomotion at all (0.3 mg/kg DFNZ) (shown in **Extended Data Fig. 6**). Notably, no tolerance was observed despite the repeated dosing of either drug. We now report the original 0.3 mg/kg DFNZ and 0.1 mg/kg FNZ CFA data in revised **Extended Data Fig. 6**.

BRAIN PENETRATION

1) Authors reasoned... “should DFNZ have high brain penetrance, metabolized [¹⁸F]DFNZ would be visualized in the brain after [¹⁸F]FNZ injection. However, no specific binding was observed in the brain starting at ~10 min after [¹⁸F]FNZ injection, indicating DFNZ has impaired brain penetrance” (pg 7).

Relative to this statement, it would be appreciated if the authors clarified whether at the dose that the parent radiotracer [¹⁸F]FNZ was injected (0.2 micrograms/kg) its metabolism produces sufficient accumulation of [¹⁸F]DFNZ to be detected in the brain within the timeframe of brain measures taken. Is it possible to provide actual [¹⁸F]DFNZ data rather than speculate on this point?

If an ¹⁸F-labeled compound has brain penetrant radiometabolites, the high sensitivity of ¹⁸F should, in theory, allow one to visualize them using PET, as discussed in detail in this review (PMID: 19616318). In the case of [¹⁸F]FNZ, there is no specific binding in the brain after ~15 min. Therefore, it is highly likely that [¹⁸F]FNZ does not have active brain-penetrant radiometabolites. Nevertheless, we agree with the Reviewer that providing actual data is preferred over speculation. Synthesizing the precursor for labeling DFNZ with ¹⁸F is of much greater difficulty than both the synthesis of DFNZ or the synthesis of the precursor of [¹⁸F]FNZ synthesis. The precursor of [¹⁸F]DFNZ would require re-protection of the secondary amine (the *N*-desethyl group), so the phenol group could be selectively reacted with the activated [¹⁸F]fluoroethanol. Given the difficulty of this synthesis, and our *a priori* knowledge that it would not serve as a PET radiotracer, we decided not to pursue the synthesis of [¹⁸F]DFNZ.

However, to address the Reviewer’s question, we performed [³H]DFNZ uptake experiments using autoradiography. We injected wild-type mice with 100 µg/kg of [³H]DFNZ and euthanized them 30 min later. We then extracted the brains, sectioned them and imaged the sections using phosphorimaging. Consistent with our hypothesis, we found that these mice showed no detectable uptake of [³H]DFNZ in the brain. Instead, we only detected [³H]DFNZ accumulation in ventricles (**Fig. 4** and **Extended Data Fig. 7**).

In relevance to the Reviewer’s other comments on brain penetration and efflux transporter interactions, we also performed efflux transporter substrate assays as well as the [³H]DFNZ uptake experiment in P-gp/Bcrp knockout mice. We found that DFNZ was a substrate for P-gp and Bcrp whereas FNZ was not (shown in revised **Fig. 4**). Furthermore, in contrast to wild-type mice, P-gp/Bcrp knockout mice showed high accumulation of [³H]DFNZ in the brain (revised **Fig. 4** and **Extended Data Fig. 7**) confirming DFNZ’s weak brain penetration and attributing it to P-gp/Bcrp-mediated efflux.

2) In relation to figures 4c-d and the statement given by the authors in pg 7: “In contrast, DFNZ is a substrate only for MATE1 (Fig. 4d) which is highly expressed at the BBB23-25”:

a) What does % inhibition mean in figure 4c and 4d? What does the dotted line represent in these figures?

This is a high throughput screening panel for measuring transporter inhibition performed by Eurofins (<https://www.eurofinsdiscovery.com/catalog/transporter-inhibition-panel-us/g346>). The effects of FNZ and DFNZ on inhibiting the activity of each transporter were calculated as % inhibition of control transporter activity. For these assays, results showing an inhibition >25% at a given transporter are considered to represent significant effects of the test compounds and evidence that they interact with the given transporter. We added this information to the Methods (**page 61**).

b) The assertion that MATE1 is highly expressed in BBB is not supported by the references provided in the manuscript. In particular, in figure 3 of REF #24 it is shown that the MATE1

protein is not expressed in the brain though present in kidney, heart, stomach. REF #25 is a review that indicates that BBB expression for MATE transporters is quite low and data is quite variable. The cited article questions MATE1 function at the BBB indicating that it needs clarification. The most abundant solute carriers in the BBB are Pg-P and BRCP, with lower levels of OCTs. Loperamide, which was mentioned by the authors as an opioid with low brain accumulation, is extruded from the brain by the transporter with highest expression levels, namely Pg-P (doi: 10.2174/156802611795371288). The authors should clarify how they believe the stated selectivity for MATE1 determines effective extrusion and reduced brain accumulation of DFNZ given the very low/absent expression levels of this transporter in BBB.

We thank the Reviewer for this comment which prompted us to perform more direct testing of FNZ and DFNZ as transportable substrates of Mate1, P-gp, and Bcrp.

To clarify, the high throughput assay panel we used in our original submission does not always differentiate transportable substrates from non-transportable substrates (i.e., inhibitors). Therefore, we performed additional experiments using the more specific monolayer Caco-2 efflux assay (PMID: 11602674) to test whether FNZ and DFNZ were transportable substrates for P-gp and Bcrp (<https://www.eurofindiscovery.com/catalog/caco-2-efflux-transporter-substrate-panel-us/G357>). We also tested Mate1 using a specific substrate permeability assay (<https://www.eurofindiscovery.com/catalog/5092>).

FNZ was not a transportable substrate for P-gp or Bcrp. However, DFNZ was a transportable substrate for both P-gp and Bcrp. Neither FNZ nor DFNZ were transportable substrates for Mate1. We now show the P-gp and Bcrp substrate assay results in revised **Fig. 4** and the Mate1 substrate assay results (along with the prior transporter inhibition assays) in revised **Extended Data Fig. 7**.

3) The description of how brain/plasma concentrations support higher ratios for DFNZ than for FNZ is quite clear. However, at the doses tested (0.03 mg/kg FNZ and 0.3 mg/kg DFNZ) analgesia by FNZ is maximal and that DFNZ's is submaximal (Figs 3b and 3c) Despite the submaximal analgesic response brain concentration of DFNZ is 13.5 fold higher than that of FNZ ($12.24/0.91 = 13.5$). Both ligands have similar binding affinity for MOR (as suggested by kinetic binding assays) so even at a less effective analgesic dose occupation of central MORs is proportionally >10 fold higher for DFNZ than FNZ. These figures do not seem warrant lower, safer, lower brain levels of DFNZ even at doses that produce submaximal analgesia by this compound.

The analgesia effects and doses the Reviewer mentions are from rat experiments whereas the brain concentrations the reviewer refers to were determined in mice. Therefore, we cannot directly compare them to each other.

Brain concentrations of FNZ and DFNZ, as determined by the analytical approaches (e.g. LC-MS), we reported in **Fig. 4** are indirect measures of target engagement and may not necessarily reflect FNZ and DFNZ MOR occupancy. This is because the drugs can differ in how they are distributed across different brain compartments (e.g., brain vasculature, CSF, brain parenchyma). For example, our uptake experiments showing that systemically delivered [³H]DFNZ preferentially accumulates in ventricles (i.e., CSF/choroid plexus) whereas [³H]FNZ accumulates in the brain parenchyma.

To address the Reviewer's concern, we performed *ex vivo* receptor occupancy using [³H]DAMGO in rats injected with equianalgesic ED₅₀ doses of FNZ and DFNZ. FNZ and DFNZ produced comparable MOR occupancy with no significant difference observed between the two drugs even though DFNZ has similar affinity to FNZ and was (i) injected at a 60-fold higher dose than FNZ and (ii) detected at >16 times greater concentration than FNZ in brain tissue homogenates. Note that we assessed occupancy at 15 min after injection, and due to its rapid brain entry and metabolism, FNZ would be expected to reach even higher MOR occupancy levels prior to this timepoint. Therefore, we likely underestimate MOR occupancy by FNZ at this timepoint.

Taken together with the [³H]FNZ and [³H]DFNZ results, the MOR occupancy results show that FNZ occupies brain MORs more effectively at lower doses, explaining its greater *in vivo* potency. These occupancy results are now discussed in the manuscript and shown in revised Fig. 4 and revised Extended Data Fig. 7.

4) In relation to statistical comparisons in extended figure 10 e: The authors used paired t test for comparison of caudate putamen to cerebellum in three independent experiments. It is conventional to use non-parametric tests (Mann Whitney) for $n < 5$. This should be corrected.

We performed a non-parametric test (Mann Whitney), but it was not significant due to the low sample size. This figure is not critical to the interpretation of the results, and we removed it from the manuscript.

RESPIRATORY DEPRESSION, BRAIN HYPOXIA, WITHDRAWAL SCORES, ANALGESIC TOLERANCE AND HYPERSENSITIVITY

1) For oxygen saturation studies shown in figure 5a DFNZ was administered at a dose that corresponds to its analgesic ED₅₀ in the hot plate test, FNZ was given at twice its analgesic ED₅₀ and morphine was given at five times its analgesic ED₅₀ (Note: an ED₅₀ for morphine of 4 mg/kg was obtained from REF19, where one of the co-authors of the current study used the same methodological approach as the one applied in the current manuscript to evaluate hot plate analgesia).

Meaningful conclusions with respect to opioid safety require drugs to be at least compared at equianalgesic doses, to show that DFNZ is effectively safer than the other two drugs at doses that similarly alleviate pain. Without this constraint, the data presented in the figure shows that DFNZ, given at lower doses than the other two drugs, does not produce hypoxia. To ensure a trustworthy safety profile, it is preferable that drugs be tested at concentrations that produce maximal possible analgesic effect and not only at ED₅₀.

We agree with the Reviewer that drugs should be compared at maximal equianalgesic doses. However, whereas pulse oximetry is used widely in humans, its adaptation to freely moving rats is difficult and leads to high variability (PMID: 30735692). Given the indirect nature of pulse oximetry for assessing hypoxia and its variability in freely moving conditions compared to our more precise and sensitive oxygen sensor and amperometry technique, we decided to remove the pulse oximetry data to ensure a trustworthy safety profile of FNZ and DFNZ.

2) Same comment as above for figure 5b: to ensure DFNZ safety, doses that produce maximal possible analgesic effect should be preferably tested. When FNZ was tested at a dose that reached the fully effective analgesic response (0.3 mg/kg), its hypoxia response was comparable to fentanyl. None of the DFNZ doses monitored in this test produce maximal effective analgesia.

In these experiments, the rats are freely moving, and drugs are injected intravenously to ensure that we obtain the most precise measurements of drug-induced brain hypoxia by avoiding confounding the physiological oxygen measures by experimenter handling. Using this procedure, we recently showed that intravenously delivered fentanyl elicited faster and stronger brain hypoxia responses compared to intraperitoneal fentanyl, with over 10-fold greater potency to induce brain hypoxia (PMID: 40086623). Therefore, the 0.3 mg/kg intravenous dose of DFNZ is not equivalent to the 0.3 mg/kg analgesic ED₅₀ dose which we determined after subcutaneous injections. At a minimum, the 0.3 mg/kg intravenous dose of DFNZ should be comparable to its maximally analgesic subcutaneously delivered dose (e.g., 1 mg/kg).

Regarding FNZ, 0.03 mg/kg intravenous FNZ and 0.03 mg/kg intravenous fentanyl produced equivalent hypoxia. However, FNZ has 4-fold greater analgesic potency than fentanyl—we previously showed that the subcutaneous analgesic ED₅₀ of fentanyl is 0.02 mg/kg (PMID: 37276825) whereas we show here that the subcutaneous ED₅₀ of FNZ is 0.005 mg/kg. As such, the more objective comparison between intravenous fentanyl and FNZ would be to compare

0.01 mg/kg intravenous FNZ to 0.03 mg/kg intravenous fentanyl. At these doses, FNZ produced significantly lower hypoxia than fentanyl.

3) Figure 5d: the treatment regimen that was used to induce morphine withdrawal included escalating doses that ranged from 5 to 20 times the analgesic ED₅₀. Withdrawal for FNZ and for DFNZ was induced with a fixed dose regimen using doses that respectively represent 6-fold and actual analgesic ED₅₀. This type of comparison does not provide adequate information to evaluate withdrawal relative to morphine. Comparable withdrawal induction schemes should be used.

Because of its propensity to cause analgesic tolerance, escalating the dosing of morphine is critical for both maintaining analgesia and for producing opioid withdrawal (PMID: 16289028 and 9631413). Indeed, a fixed dose of morphine given daily for just 2-3 days increases its analgesic ED₅₀ value by over 3-fold (PMIDs: 26235542 and 18046309). In contrast, as we show in revised **Fig. 3** and **Fig. 5**, repeated FNZ or DFNZ (for up to a month) did not produce analgesic tolerance and therefore, unlike morphine, escalating their dosing is not necessary for maintaining analgesia. For these reasons, it would not be meaningful to use an escalated dosing regimen to evaluate the propensity of FNZ and DFNZ to produce opioid withdrawal. Nevertheless, to make the withdrawal induction schemes for FNZ, DFNZ, and morphine more comparable, we injected FNZ and DFNZ twice daily and used high doses (more than 3-fold higher than their analgesic ED₅₀). In contrast, we injected morphine only once per day.

4) Figures 5e-5g: It was not clear from the results and methodological sections whether von Frey filaments were applied on naïve animals or on animals that had been previously subject to any type of manipulation to induce pain hypersensitivity. Von Frey filaments are usually applied when there is a context of hypersensitivity such as allodynia. The rationale for similarly measuring nociceptive tolerance and hypersensitivity with von Frey filaments, particularly if this was done on naïve animals, should be explained in the methodological section. experimental data should be presented as actual withdrawal threshold as was done in figures 4f and 4i for the analgesic responses. Actual threshold values would allow the reader to gain insight into the pressure that needs to be applied to evaluate nociceptive tolerance and hypersensitivity.

We apologize for the lack of clarity. Chronic opioid use can lead to both tolerance as well as hyperalgesia, the latter particularly during opioid withdrawal, and both tolerance and hyperalgesia can promote opioid misuse and relapse. We previously used this procedure in rats to model chronic opioid exposure-induced hyperalgesia (outside the context of a pain model) (PMID: 33997152). We now discuss the rationale for this experiment in the Methods section (**page 71**) and cite our prior study in the main text (**page 10**). We also specify that the rats were naïve prior to repeated drug exposure.

As requested, we now show the data as actual threshold values in revised **Extended Data Fig. 9**. FNZ and fentanyl were tested at the same time along with a saline-injected control group. Due to its later discovery, DFNZ was tested at a different time using another saline control group. We preferred to show the normalized data in revised **Fig. 5** because the saline control groups slightly differ from each other. For this reason, we normalized each drug group to its respective saline control so that we could plot them together on the same graph.

5) In figure 5h the authors show a downregulation of DAMGO binding sites by fentanyl but not by FNZ nor DFNZ. In the abstract the authors state: “DFNZ does not induce MOR desensitization after repeated exposure.” It is important to distinguish desensitization from downregulation. The manuscript does not provide experimental evidence indicating that DFNZ fails to induce functional desensitization of MOR.

We agree and therefore replaced “desensitization” with “downregulation”.

6) In figures 5i-5j the use of a range of doses that ensure effective analgesia by FNZ (0.01-0.017 mg/kg) and DFNZ (0.3-1 mg/kg) indicated that both ligands can substitute for heroin in self administration trials. The authors concluded that this observation is consistent with reinforcing effects for both of the nitazene derivatives (pg 11). However, to dispel the notion that

DFNZ may have significant reinforcing properties the drug was tested in a second series of trials where it was administered by i.v. infusions of 0.03 mg/kg/infusion. Knowing that analgesic ED₅₀ for DNFZ in hot plate is 10 times this dose, how safe is it to conclude that this drug has weak reinforcing properties when used therapeutically?

The rats learned to intravenously self-administer DFNZ at a unit dose of 0.03 mg/kg/infusion indicating that this was a reinforcing unit dose. At this unit dose, the rats committed ~20 infusions over a 3-hour period which amounts to a cumulative exposure of ~0.6 mg/kg in 3 hours. During the dose-response phase, the rats self-administered even more of the drug. For example, at the 0.1 mg/kg/infusion unit dose they self-administered ~10 infusions, receiving 1 mg/kg in 3 hrs. Given the intravenous delivery, these self-administered DFNZ doses are, at a minimum, comparable to the subcutaneous DFNZ doses that produced maximal analgesia and that decreased heroin self-administration. Moreover, the self-administering rats received this large intravenous exposure of DFNZ daily for several weeks. Despite this continuous and large exposure to DFNZ, and in contrast to heroin, the rats extinguished their self-administration of DFNZ immediately and did not reinstate DFNZ self-administration after extinction. We are not aware of any other opioid analgesic that is characterized by immediate extinction of self-administration. Furthermore, as shown in our extensive photometry experiments in revised **Fig. 6**, we found that DFNZ has limited effects on dopamine neurotransmission and a decreased capacity to promote fast (phasic) dopamine responses compared to other opioids even at high, maximally analgesic doses, much greater than self-administered doses. Combined, these data suggest that repeated analgesic doses of DFNZ would be expected to produce weak reinforcement if used therapeutically.

HYPERLOCOMOTION AND DOPAMINERGIC NEUROTRANSMISSION

1) In figure 6a-c locomotor activity (over 30 min) was evaluated for DNFZ at 0.3 mg/kg (which corresponds to analgesic hot plate ED₅₀), FNZ was evaluated at 0.03 mg/kg (6 fold analgesic hot plate ED₅₀), fentanyl at 0.1mg/kg (5 fold analgesic hot plate ED₅₀) and morphine at 20 mg/kg (5 fold analgesic hot plate ED₅₀). At these doses DFNZ was the only drug that failed to induce locomotion

However, in extended data figure 9, it can be seen that within the first 30 min post injection (similar time frame as shown in figure 6c) 3 mg/kg of DFNZ (corresponding to 10-fold the ED₅₀ for hot plate analgesia) produced similar locomotor activity as 30 mg of morphine (corresponding to 7.5-fold the the ED₅₀ for hot plate analgesia).

These two sets of data clearly show that unique 30 min locomotion profile described for DFNZ in figure 6 is dose dependent, and disappears at higher doses tested in extended data figure 9.

We would like to point out that the Reviewer compares the mouse locomotor doses to rat analgesic doses. A more objective comparison would be to compare mouse locomotor doses to mouse analgesic doses. We previously showed that 15 mg/kg morphine and 0.1 mg/kg fentanyl produced sub-maximal analgesia in mice (PMIDs: 21490202; 19914603; 29440403). Therefore, in mice, 20 mg/kg morphine and 0.1 mg/kg fentanyl are not representative of 5-fold the analgesic ED₅₀.

The Reviewer is correct, and as shown in revised **Extended Data Fig. 6** (original **Extended Data Fig. 9**), DFNZ induced hyperlocomotion in mice at very high doses (e.g. 3 mg/kg). For this reason, we repeated the dLight photometry experiments using both analgesic and peak locomotor doses of FNZ (0.03, 0.1 mg/kg) and DFNZ (0.03, 1, and 3 mg/kg). We also used peak locomotor doses of morphine (40 mg/kg) and fentanyl (0.3 mg/kg), as shown in revised **Extended Data Fig. 6** and as previously reported (PMID: 35489781). However, we decided to not include the photometry locomotor data because mice were tethered to the fiber patch cord which may have confounded their locomotion, and we already report extensive locomotor characterization in untethered mice (revised **Extended Data Fig. 6**; original **Extended Data Fig. 9**).

2) Same low doses as the ones used to evaluate locomotion in figure 6a-c were used in figure 6d-e to conclude: “Overall, fentanyl, morphine, and FNZ mimicked the effect of classical MOR agonists on dopamine transmission in NAcc. In contrast, DFNZ was ineffective, providing additional evidence for the unique behavioral and physiological effects of this new drug”.

Effects of doses similar to those used in extended data figure 9 should be explored.

As mentioned in our response to the above comment, we now include dLight photometry data at several additional doses of FNZ and DFNZ, including doses that produced peak hyperlocomotion as shown in the original **Extended Data Fig. 9**. We also include doses of morphine and fentanyl that produced peak hyperlocomotion (revised **Extended Data Fig. 6** and (PMID: 35489781)).

Referee #4 (Remarks to the Author):

This elegant study reports the development and characterization of two novel mu opioid receptor (MOR) agonists, FNZ and DFNZ, derived from the nitazene scaffold. Conducted by leading experts in the field, this work provides promising opioid candidates, that produce strong analgesic effects with minimal side effects, to the scientific and biomedical communities.

This study focusing on mu opioid receptor agonists is highly relevant, given their central role in pain management and their significant societal impact due to the risk of abuse. By demonstrating potent analgesia with reduced adverse effects of FNZ metabolite, DFNZ, represents important advances in opioid pharmacology and hold potential for safer therapeutic applications.

In brief, the authors demonstrate that FNZ and its metabolite, DFNZ, exhibit high affinity and plausible selectivity for the mu opioid receptor (MOR) in cell assays. They further characterized the FNZ binding site within the MOR–Gi protein complex using cryo-EM at a 2.3 Å resolution. The study also explored the spatiotemporal signaling profiles of FNZ and DFNZ, revealing that DFNZ displays a distinct signaling bias—favoring Go over β -arrestin 2—as well as slower receptor internalization and a lower endosomal trafficking.

As anticipated, both FNZ and DFNZ produced robust analgesic effects in rodent models of acute pain (hot plate test) and chronic pain (complete Freund’s adjuvant model assessed via Hargreaves and von Frey tests). Crucially, DFNZ showed reduced brain penetration in both mice and rats compared to FNZ, suggesting a lower potential for central side effects. Consistent with this, DFNZ was associated with significantly attenuated adverse effects, including the absence of respiratory depression, no tolerance or hypersensitivity, reduced physical withdrawal, and rapid extinction with no reinstatement of drug-seeking behavior. Mechanistically, the authors show that unlike classical opioids such as morphine and fentanyl, DFNZ does not increase dopamine release in the nucleus accumbens and does not alter MOR density in this brain region. Additionally, they report that DFNZ exhibits reduced efficacy at MOR–Galanin 1 receptor heteromers in vitro.

Taken together, DFNZ appears to be a MOR agonist that combines strong analgesic efficacy with a reduced impact on receptor trafficking and brain penetration, resulting in fewer side effects—including no respiratory depression, no tolerance, and a reduced risk of opioid use disorder. In conclusion the present study supports the strong therapeutic potential of DFNZ.

Overall, the manuscript is clearly written, with a nice experimental design, and the data are well presented. However, we believe that certain methodological details warrant further discussion, and one or two additional key experiments may be required to fully support the study’s conclusions.

We thank the Reviewer for their enthusiasm and positive comments.

Comments:

-The authors, characterized nicely the structure of FNZ with MOR-Gi complex at a 2.3 Å resolution (Figure 1k). Is there a reason, why the authors did not characterize the structure of DFNZ with the MOR-Gi complex? as DFNZ seems the best promising compound, it may be important to determine whether the binding site is the same of FNZ.

We did not have DFNZ synthesized at the time the cryo-EM experiments were performed. Given the structural similarity of FNZ and DFNZ, the fact that FNZ occupies the orthosteric site, and that each drug competitively inhibits each other's binding to MOR, it is certain that DFNZ has the same overall binding site.

-The authors nicely demonstrated the selectivity for MOR for FNZ and DFNZ competitive binding screen using rat brain membranes (Figure 1b). To confirm the selectivity of FNZ and DFNZ compounds, it will be important to check it in wild-type and MOR knockout animals. We believe that it will important to realize 3H FNZ and 3H DFNZ autoradiography in wild-type and MOR knockout animals.

We performed autoradiography using [³H]FNZ and [³H]DFNZ in *Oprm1* knockout mice. No specific binding of [³H]FNZ or [³H]DFNZ was observed in these mice. We include these results in **Extended Data Fig. 2** and discuss them in the revised manuscript (**page 3**).

-Similarly, authors reported a strong analgesic effect of FNZ and DFNZ (Figure 3). A control experiment using MOR knockout mice would be critical to confirm on-target effects *in vivo*.

We performed hyperlocomotion experiments using various doses of FNZ, DFNZ and morphine in *Oprm1* knockout mice. We also performed hot plate assessment using high (peak locomotor) doses of FNZ and DFNZ. Neither FNZ nor DFNZ produced any changes in locomotion or hot plate responses confirming their MOR selectivity *in vivo*. We include this data in revised **Extended Data Fig. 6** and discuss them in the revised manuscript (**pages 6-7**).

-The authors used both rats and mice, which is interesting and might strengthen the conclusions (Figure 3). Furthermore, in rats, both males and females have been used. This aspect is crucial, given the sex differences observed in both pain perception and addictive-like behaviors. Yet, the author never mentioned if they observed any sex difference in the behavioral paradigms they used. This is very likely due to the low number of rats used. It might be important to increase the number of rats and analyze more precisely the potential sex differences. Then, for mice, only males have been used. For such an important topic it appears of crucial interest to include both sexes.

We apologize for the lack of clarity. The mouse hyperlocomotion experiments were performed in both sexes using a sample size sufficient for observing meaningful sex-dependent effects (n=12 mice per drug, 6 male and 6 female). In addition, we included both males and females in the experiments with *Oprm1* knockout mice (n=4/sex). We did not observe any sex differences in these experiments.

Regarding rats, we used both males and females for most procedures, including pharmacokinetics assessments, MOR occupancy, naloxone-precipitated withdrawal, self-administration studies, tolerance, mechanical hypersensitivity, and MOR downregulation studies. Most of these experiments included 4-8 rats per sex per drug, which is typically enough of a sample size to reveal notable sex-dependent effects in these procedures. However, like in the mouse experiments, we did not observe any meaningful contribution of sex in these experiments.

Our plan is to comprehensively investigate the precise central vs. peripheral mechanisms driving the analgesic efficacy of DFNZ, along with any potential sex-dependent effects, in subsequent studies. We now discuss this in the revised manuscript (**page 18**).

-While the authors evaluated the effect of FNZ and DFNZ on catalepsy and hyperlocomotion in rats and mice, respectively, it appears intriguing not to see a justification of their behavioral design when assessing pain and addictive-like behaviors. It is very surprising to see that for rats they used a test where an absence of movement could be interpreted as analgesic properties of

the compound, while they show catalepsy induced by both compounds. Indeed, the hot plate results in rats show a strong increase in the latency to jump or lick the paw, interpreted as an analgesic effect of the compound, yet, this effect is observable at the same timing as what the authors described as catalepsy, for the same doses. In other words, it seems that this increased latency might be simply due to the FNZ and DFNZ-induced catalepsy and has nothing to do with analgesia. Could the authors precise the design and 1- show the raw data (in sec) for the hot plate and 2- give insight about the rats' ability to move after FNZ and DFNZ injections.

We previously used the specific rat hot plate assay as a measure of centrally mediated antinociception, along with catalepsy and body temperature effects, to characterize the pharmacology of different opioids (PMID: 37276825). It is unlikely that catalepsy was confounding analgesic effects of FNZ or DFNZ in rats because cataleptic effects were observed at doses that are above those inducing analgesia. The full time-course data for analgesia and catalepsy clearly illustrate this point. Stated another way, there are drug doses on the rising phase of the analgesic dose-response curve where no signs of catalepsy or other locomotor changes are present. As the Reviewer requested, we show the raw data (in sec) for the hot-plate in revised **Extended Data Fig. 6**.

-Then, in regard to analgesic tests with mice, the authors described an hyperlocomotion. This effect is weaker than the one provoked by morphine but still it is present. Could the author precise their design and clearly rule out any potential effect on locomotion that could interfere with the data obtained while evaluating nociception. One option could be to evaluate the analgesic effect of both compounds with a test that does not require movements (grimace scale, physiological variables modified after pain...). Another control option, may to test the effects of DFNZ on hotplate in mice and on CFA/Von Frey in rats.

We apologize for the lack of clarity in our text. The mouse analgesia assessments were initiated at 1 h after injection of FNZ or DFNZ, at time at which hyperlocomotion was absent, ruling out potential confounding effects. We mentioned this in the original text but have made it more explicit in the revision (**page 6**).

-One claim of this article is that these compounds have little if no addictive potential. Indeed, rats present a rapid extinction and no clear relapse has been observed. However, these animals still work for the compounds (FR1, FR3 and PR) and show a classic inverted-U dose response curve (Figure 5k-n). These results suggest rewarding properties. Since these FNZ and DFNZ are MOR agonists, it appears crucial to test these "rewarding" properties in animals suffering from chronic pain (For ex CFA) and see if these individuals might develop addictive-like behaviors.

We appreciate this suggestion, but the Shaham lab and the labs of Matt Banks and Steve Negus were unable to develop a rat model of pain (acute or chronic)-induced increases in opioid (fentanyl and heroin) self-administration, choice, and reinstatement (PMID: 33765177). With one exception, previous studies also failed to show pain-induced increases in opioid self-administration (PMIDs: 39366430, 39605379, 36948356, and 26338332). Thus, it is likely that pain exposure will have no effect on FNZ or DFNZ self-administration, choice, or reinstatement. With the Reviewer's and Editor's permission, we prefer not to perform such labor-intensive experiments as their results are unlikely to change the conclusions of our study.

-Finally, the absence of NAc dopamine release after DFNZ injection is surprising (Figure 6) as DFNZ was shown to recruit Gi, and it is well know that the increased of dopamine released from opioid is caused by the inhibition of Gabaergic interneurons of the VTA. It could be interesting to monitor the calcic activity of dopaminergic neurons in the VTA to evaluate whether this specific compound does not require dopamine signaling at all or if instead it specifically prevents the dopamine release within the NAc. Another option may to test the effect on NAc dopamine dynamics of a higher dose of DFNZ which produce a high level of DFNZ in the brain.

As per the Reviewer's suggestion, and as per our response to Reviewer 3, to further examine the capacity of DFNZ to modulate dopamine neurotransmission, we performed additional dLight

photometry experiments using higher doses of FNZ and DFNZ. As the Reviewer suggested, we also performed photometry experiments to measure the activity of dopamine neurons in the NAc by injecting a Cre-dependent AAV expressing axonal-GCaMP6s into the VTA of TH-cre mice and implanting a fiber probe into the NAc. Overall, we describe unique, converging effects of DFNZ on both dopamine neuron activity and dopamine responses compared to both FNZ, morphine, and fentanyl. As we report in the revised Discussion, we believe that these unique properties of DFNZ explain its capacity to produce hyperlocomotion and self-administration but not persistent drug seeking and reinstatement.

Minor comments:

-In figure 4 f. the two levels of green are not easy to distinguished.

Panels 4e and 4f have been revised to better distinguish the brain and plasma data.

-In Figure 4 h-l, it is not clear whether the dose of 0,3 and 1 mg/kg were tested for FNZ. It may be useful to change the graphical representation of these data.

We apologize for this confusion. We revised the legend of this figure to state that these high doses of FNZ were not tested.

Referee #5 (Remarks to the Author):

I co-reviewed this manuscript with one of the reviewers who provided the listed reports.

Thank you for reviewing our manuscript.

12-26-2025

This letter is in response to the request for revision of our manuscript (2025-04-09979A), entitled "A μ opioid receptor superagonist that produces analgesia with minimal adverse effects". We thank the reviewers for their enthusiasm, positive evaluation, and constructive comments, which have considerably strengthened our manuscript.

Below we address the reviewers' specific comments **in blue font** and highlight new text in the revised manuscript using tracked changes.

The specific experiments we performed are listed below:

1. **Assessment of MOR occupancy by DFNZ in rats using 1 mg/kg subcutaneous dose (Fig. 4 and Extended Data Fig. 7).**
2. **Assessment of brain hypoxia by DFNZ in rats using 1 mg/kg subcutaneous dose (Fig. 5).**

Sincerely,

Michael Michaelides, Georgios Skiniotis, and Kenner Rice (on behalf of all coauthors)

Referees' comments:

Referee #3 (Remarks to the Author):

I have examined the revised version of the manuscript submitted by Gomez et al. where the authors have replied to the different concerns raised upon their first submission. I thank the authors for their clear responses and the experiments that were added to substantiate their point of view. Many points have been resolved but issues remain concerning dosing in relation to potential for respiratory depression, assessment of brain penetration at therapeutic doses as well as the accuracy of the originality statement in relation to the mechanism supporting the extrusion of DFNZ from the brain. Below are my comments in relation to the changes done, addressing these points in particular.

We thank the Reviewer for their positive comments. We address their remaining issues directly in the responses below.

PHARMACODYNAMIC PARAMETERS, SIGNALING BIAS AND TRAFFICKING PROFILES

The authors have provided satisfactory answers clarifying the use of different BRET techniques for G protein data gathered in figures 1 and 2 and have carried out new displacement experiments to correct previously unexplained differences in MOR affinity. Also, I fully appreciate and agree with the decision of de-emphasizing any claim that G protein over barr2 bias may underlie DFNZ's safety profile.

In this sense I would like to point out that although the authors indicate that preferential G-protein signaling has not been linked to greater chance of overdose, we also know that G-protein mediated signaling is directly implicated in respiratory depression, which is the main cause of death by opioid overdose (nitazenes included). Respiratory depression by opioids involves reduced excitability of respiratory brainstem nuclei, including MOR-mediated hyperpolarization of neurons in the preBötzing complex and MOR-mediated reduction of glutamate release onto these neurons. These effects involve $G\beta\gamma$ -mediated regulation of membrane channels while inhibition of cAMP by $G\alpha i/o$ proteins reduces excitability mediated by sodium-channels. Thus, upon its penetration into the brain, DFNZ activation of MOR receptors on the respiratory nuclei will preferentially engage the pathways that mediate respiratory depression.

That being said, the data are acceptable to show differences between FNZ and DFNZ and these differences are further supported by APEX data.

We appreciate the Reviewer's comments.

BRAIN PENETRATION

In the previous submission the authors had obtained in vitro data that ruled out a role of P-gp/Bcrp in DFNZ efflux from the brain and had proposed a role for MATE1 in this effect. Following re-assessment of the literature the authors carried out alternative in vitro assays to evaluate the effect of DFNZ on these targets. They now find that DFNZ is indeed a substrate for P-gp/Bcrp, but not MATE1 (n=2). They also used transporter knockout mice to show that at an in vivo dose of 100 micrograms/kg of [3H]DFNZ (s.c.; n=2) was extruded from the brain parenchyma by the P-gp/Bcrp transporters. The authors further show that rats that received 100 micrograms (i.v; n=8) of DFNZ the inhibition of P-gp by tariquidar precipitates ~15% decrease in O₂ saturation, thus linking P-gp-mediated extrusion of DFNZ to the safety of low doses of the drug.

It was not clearly explained why the in vitro assays used in the two different occasions produced opposing results. However, the in vivo data support the latest in vitro results directly implicating P-gp/Bcrp in the exclusion of low doses of DFNZ from the brain.

We apologize for lack of clarity in our prior explanation. The assays we used measured different (but complementary) aspects of transporter-drug interaction and that is why they produced opposing results.

The transporter *inhibition* assays measure inhibition of transporter function by the test drug. The transporter *substrate* assays measure transport of the test drug by each transporter.

The inhibition assays showed that (I) DFNZ, but not FNZ, is an inhibitor of Mate1 and (II) neither FNZ nor DFNZ are inhibitors of P-gp or Bcrp.

The substrate assays showed that (I) DFNZ, but not FNZ, is transported by P-gp and Bcrp and (II) neither FNZ nor DFNZ are transported by Mate1.

Combined, these assays show that DFNZ is (I) transported by P-gp and Bcrp and (II) inhibits the function of Mate1. We revised the corresponding section to further increase its clarity (page 8).

These new findings indicate that DFNZ shares the exclusion mechanism of the prototypical P-gp substrate loperamide, which is also a MOR super-agonist (e.g.: PMID: 41193810). Given this precedent, the authors may want to reformulate the statement “a MOR super-agonist with limited brain penetrance, such as DFNZ has not been previously reported” and provide information on how DFNZ differs from loperamide, a previously described super-agonist with limited brain penetrance.

We thank the Reviewer for this information. Like DFNZ, loperamide is a P-gp substrate and according to recent work (PMID: 41193810) appears to have MOR superagonist properties.

Loperamide is subject to strong P-gp-mediated transport and has extremely poor brain penetrance, which renders it peripherally selective with no analgesic effects after systemic injection (PMIDs: 29499278, 10087042, 204876, 8647944). Unlike loperamide, DFNZ is subject to weaker P-gp-mediated transport, is a Bcrp substrate, and produces brain MOR occupancy, though its capacity to occupy brain MORs is limited. Therefore, DFNZ's capacity to occupy brain MORs, albeit it to a limited extent, differentiates it from loperamide, and combined with its MOR superagonism, explains its strong analgesic efficacy. We revised the specific sentence and corresponding text to highlight the differences in brain penetrance between loperamide and DFNZ (page 18).

BRAIN EXCLUSION AND DFNZ SAFETY MICE

The authors showed that a subcutaneous dose of 100 micrograms/kg [3H]DFNZ given to mice does not induce accumulation of the drug in brain parenchyma, remaining within the ventricles. They have also carried out extra experiments showing that higher DFNZ doses (1 mg/kg, s.c.) than the ones previously tested (0.3 mg/kg, s.c.), produces fully effective analgesia in the mouse model of inflammatory pain (CFA).

There was no exploration of how this fully effective therapeutic dose affects brain accumulation of DFNZ. This information is crucial not only in trying to differentiate this novel ligand from loperamide but in establishing the doses that warrant therapeutic actions with limited brain penetration. Whether it is via use of [3H]DFNZ or displacement of [3H]DAMGO the authors have the means to do it.

Although there is no direct data on how therapeutic doses of DFNZ accumulate in the brain, the functional data provided by the authors indicates that a fully effective analgesic dose of 1mg/kg of DFNZ can produce maximal central effects in the reward system, implying sufficient penetration to produce similar effects as fentanyl also given at an analgesic dose. In particular, I am referring to the fact that 1 mg/kg, s.c. of DFNZ produces Ca²⁺ mobilization in NAcc dopamine terminals reaching similar maximal peak as a dose of 3 mg/kg of the drug (Figure 6b), which in turn surpasses the effects caused by 0.3 mg/kg of fentanyl (Supplementary Figure 10 a). Therefore, at the fully effective analgesic dose of 1 mg/kg brain penetration is sufficient to support significant effects of DFZN in the reward circuit, and in this case the effects are greater than those of brain penetrating fentanyl. How can we be sure that central MOR on the respiratory nuclei are spared from activation if those in the reward system are? The authors did not consider assessing the scenario where brain penetration of high doses of this lipophilic

nitazene overrides the extrusion mechanism that is operant at lower doses.

I would have preferred the use of a greater dose range of DFNZ to directly evaluate brain penetration. In absence of such data it is difficult to accept the conclusion that “CNS penetration of DFNZ is limited”, particularly at therapeutically relevant doses.

Tritiated compounds are synthesized with high specific activity (the ratio of radioactivity to the amount of substance) which allows them to be used at extremely low (i.e., tracer level) molecular concentrations, sufficient for radiometric detection without introducing a significant chemical mass that would alter the biological system being studied. We cannot perform the [3H]DFNZ uptake assay using a dose of 1 mg/kg because the specific activity and concentration of [3H]DFNZ allows injecting a maximum dose of 0.1 mg/kg. Therefore, to assess the level of MOR occupancy at a higher DFNZ dose, we performed DFNZ occupancy experiments by injecting rats subcutaneously with saline or 1 mg/kg DFNZ, collecting the brains at 15 min after injection and performing [3H]DAMGO autoradiography. We found that 1 mg/kg DFNZ produced similar level of MOR occupancy as 0.3 mg/kg DFNZ and 5 µg/kg FNZ, confirming that DFNZ produces limited brain MOR occupancy, even at a therapeutic dose of 1 mg/kg. The limited brain MOR occupancy of DFNZ is likely due to its impaired brain penetrance as shown in the [3H]DFNZ uptake experiments. This leads us to conclude that the higher brain concentration of DFNZ with increasing dose as detected using LC/MS in brain homogenates represents its accumulation in ventricles (as shown in the [3H]DFNZ uptake experiments). The new DFNZ occupancy experiments are now presented in revised **Figure 4** and **Extended Data Figure 7**.

RATS

New data obtained in rats indicate that 300 micrograms/kg of DFNZ given subcutaneously roughly produce 40% occupation of DAMGO labelled sites in NAcc, indicating partial occupation of central MORs at this submaximal analgesic dose. At a fully effective analgesic dose of 1 mg/kg, occupation was not evaluated. I can only reiterate the need for exploration of a full dose range to evaluate occupancy at fully effective analgesic doses.

We agree and address this in the above comment.

HYPOXIA

Pulse oximetry measures reported in the previous version of the manuscript had evaluated oxygen saturation following subcutaneous administration of submaximal analgesic doses of DFNZ. The authors have now abandoned this approach on the basis of its high variability in freely moving animals. This precludes the evaluation of whether fully effective subcutaneous analgesic doses of DFNZ, which we know can penetrate the brain to induce significant effects in the reward system, do actually spare the activation of central MORs in respiratory nuclei. Nonetheless the authors use an electrochemical approach to evaluate oxygen saturation. These estimates are more precise than pulse oximetry but require i.v. administration, preventing the direct assessment of the respiratory effects of the subcutaneous DFNZ doses that were used to evidence its analgesic effects.

The i.v. doses of DFNZ that were tested for hypoxic effects are 100 and 300 micrograms/kg. The authors argue that 300 micrograms/kg of DFNZ given i.v. should be considered equivalent to 1 mg/kg of DFZ given s.c., and we should take these results as evidence of DFNZ safety. Because we are speaking about safety of a super-agonist that is biased towards a signaling path that directly engages the mechanisms of respiratory depression, and since functional data indicate significant brain penetration at therapeutic doses, the speculation on dose equivalence is not re-assuring. Exploration of how a larger range of DFNZ doses induces hypoxia would have been preferred. If DFNZ is meaningfully excluded from the brain despite dose escalation, its safety should be revealed by a ceiling effect on hypoxia. This or any similar type of evidence was not provided.

To reassure the Reviewer, we injected rats subcutaneously with saline or a therapeutic dose of DFNZ (1 mg/kg) and monitored oxygen levels in the NAc. DFNZ induced a moderate, tonically sustained increase in oxygen for at least 60 min, suggesting a lack of activation of central

MORs in respiratory nuclei at this therapeutic dose. We previously showed that low to moderate doses of morphine produce increases in NAc oxygen (hyperoxia), whereas higher doses produce hypoxia (PMID: 30183460). The hyperoxic effect of DFNZ at this therapeutic dose is consistent with hyperoxic effects of morphine at low to moderate doses. We include this new data and discuss it in the revised manuscript (Fig. 5).

DOPAMINERGIC NEUROTRANSMISSION

The authors now provide evidence that similar to the highly efficacious agonist methadone but unlike the weaker morphine, DFNZ (1 mg/kg) activates the MOR-Gal1R heterodimer to reduce DA release in the VTA.

We thank the Reviewer for his/her initial comments which led to this observation.

Referee #3 (Remarks on code availability):

N/A

Referee #4 (Remarks to the Author):

Authors did a great job, adding several experiments and thoroughly addressing most of our concerns. However, we still have three comments:

- Regarding nociception, authors claimed to have tested the effect of DFNZ in WT and KO animals in the hot plate test in extended data 6. We found the data for locomotion in WT and KO animals but not the data for the hot plate test. Could the authors clarify or add the experiment to study the effect of DFNZ on analgesia in WT and KO MOR animals? It would be crucial to demonstrate that the DFNZ analgesic effect is mediated via MOR.

We apologize for the confusion. We tested Oprm1 KO mice using very high, supra-analgesic doses of FNZ (0.1 mg/kg) and DFNZ (3 mg/kg). These results were included in Extended Data Figure 6 panel I. We did not claim to test WT mice and did not test WT mice in this assay because of the extensive testing performed in WT mice in other experiments such as the mouse CFA assays, locomotor assays, autoradiography experiments, and photometry experiments.

- We appreciate the authors' explanation regarding the challenges in establishing pain-induced increases in opioid self-administration, especially in such labs with leading experts in the field. However, our concern is not solely about a potential escalation of intake under pain conditions but rather about whether FNZ and DFNZ possess different reinforcing properties in animals with chronic pain. Although the authors argue that pain is unlikely to alter self-administration, the current data still indicate that the compounds maintain operant responding (FR1, FR3, PR) and produce a classic inverted-U dose-response function, which is consistent with rewarding effects.

To more directly address the claim that these compounds have little or no addictive potential—particularly given their MOR agonism—we suggest performing a conditioned place preference assay across a range of doses in both control and CFA-treated mice. We would also see whether the U-inverted curve is still observable and, more importantly, if it is shifted in animals with pain conditions. It would also be important to determine that this rewarding effect is absent when MOR is deleted. CPP in WT and MOR KO animals is less labor-intensive than self-administration procedures and would provide complementary evidence regarding the reinforcing or motivational properties of FNZ and DFNZ in the context of chronic pain. Such data would help strengthen the conclusions about the compounds' abuse liability.

We appreciate the suggestion for performing CPP studies to further inform the extent to which FNZ and DFNZ have addictive potential. However, we do not think that these experiments are necessary nor relevant to address this question because, regardless of their outcome, our

conclusion regarding the potential abuse liability of FNZ and DFNZ, which is based on the IVSA procedure, will remain the same. The IVSA procedure is the gold standard procedure for assessing the abuse liability of drugs. The CPP procedure is regarded as a model for assessing the initial acute rewarding effects of drugs. Thus, results from studies using the CPP procedure have little relevance to the abuse liability of novel compounds without supporting evidence from the IVSA procedure.

The main interpretation problems with the proposed CPP experiment are that (1) it does not address abuse liability and (2) relief of pain itself is inherently rewarding and induces CPP. That is, rodents will develop CPP to non-rewarding drugs that produce pain relief through a negative reinforcement mechanism (PMID:7838585). Therefore if FNZ and DFNZ show greater CPP in mice exposed to CFA compared to non-CFA mice, it is unclear whether the effect is due to the positive reinforcing effects of the drugs (which we have already established using IVSA) or whether the increase in CPP is due to the negative reinforcement produced by the analgesic effects of FNZ and DFNZ (an outcome that will confirm our other data on the effect of the two drugs in other pain-related models). Notably, both the study cited above and at least one more study (PMID:2853321) showed that CFA does not increase opioid CPP.

Finally, while less laborious than IVSA, the proposed CPP experiments would still require considerable time and resources. Assessing the effects of multiple doses of FNZ and DFNZ in both controls and CFA exposure and in WT and KO mice would require hundreds of mice and several months to complete.

To address the reviewer, we added a section to the Discussion acknowledging the lack of characterization of FNZ and DFNZ reinforcement in the context of pain as a limitation of our study.

- In panel 5d, the statistical reporting is unclear. It is not evident what the asterisks refer to—specifically, which groups are being compared when 5 stars are written, and the legend does not describe the significance.

We apologize for the lack of clarity. The asterisks have been clarified in the legend. There are no asterisks with 5 stars. There are two rows of asterisks for FNZ and DFNZ. The top row corresponds to the difference relative to VEH and the bottom relative to morphine.

Referee #4 (Remarks on code availability):

We detected nothing wrong in the code.

Thank you for reviewing the code.

Referee #5 (Remarks to the Author):

I co-reviewed this manuscript with one of the reviewers who provided the listed reports.

Thank you for reviewing our manuscript.

02-10-2026

This letter is in response to the request for revision of our manuscript (2025-04-09979B), now entitled "A μ opioid receptor superagonist analgesic with minimal adverse effects". Below we address the reviewers' specific comments in blue font.

Sincerely,

Michael Michaelides, Georgios Skiniotis, and Kenner Rice (on behalf of all coauthors)

Referees' comments:

Referee #3 (Remarks to the Author):

I have examined the revised version of the manuscript submitted by Gomez et al. this last January.

The authors have carried out new experiments eloquently showing that fully effective analgesic subcutaneous doses of DFNZ have limited brain penetration and do not produce hypoxia.

I have no outstanding issues regarding the study.

I would also like to congratulate the authors on their discovery.

We thank the Reviewer for their positive comments.

Referee #4 (Remarks to the Author):

The authors have satisfactorily addressed our previous concerns and have done an excellent job overall. We have only a few remaining minor comments:

1. Pain assay in Oprm1 KO animals:

For clarity, we suggest that the authors specify in Extended Data Figure 6, panel i the following treatment groups: Oprm1 KO–FNZ (0.1 mg/kg) and Oprm1 KO–DFNZ (3 mg/kg) instead of FNZ 0.1 mg/kg and DFNZ 3 mg/kg.

We revised this figure as per the Reviewer's suggestion.

2. Assessment of misuse liability in pain conditions:

We thank the authors for adding a discussion addressing the risk of misuse in individuals suffering from pain. Regarding the suggested conditioned place preference (CPP) procedure, we proposed an alternative, simpler approach, given that the authors indicated intravenous self-administration under pain conditions is technically challenging at the first revision.

Our original objective was to evaluate self-administration, extinction, and drug-induced reinstatement in a chronic pain model; however, the authors declined this approach due to the extensive workload it would entail. We nevertheless believe this question remains important, as these compounds are not intended for recreational use but rather as analgesics. Thus, although FNZ and DFNZ do not appear to confer abuse liability in pain-naïve animals, they are likely to be used, and potentially misused, by individuals experiencing pain. Assessing abuse potential in this clinically relevant context would therefore strengthen the translational impact of the study.

We appreciate the Reviewer's comment. We made additional revisions to the text to address this important point.

3. Statistical annotation in Figure 2e:

In Figure 2e, we suggest replacing the asterisk (*) with a different symbol (e.g., # or &) to indicate differences between FNZ and DFNZ relative to morphine (bottom row), in order to avoid any potential confusion to have a same sign for two conditions.

We believe the Reviewer is referring to Figure 5d as figure 2e does not have any asterisk or morphine comparison. We revised Figure 5d as per the Reviewer's suggestion to better illustrate the statistical differences and avoid potential confusion.

Referee #4 (Remarks on code availability):

No issues detected.

Referee #5 (Remarks to the Author):

I co-reviewed this manuscript with one of the reviewers who provided the listed reports.

Thank you for reviewing our manuscript.

02-11-2026

This letter is in response to the request for revision of our manuscript (2025-04-09979B), now entitled "A μ opioid receptor superagonist analgesic with minimal adverse effects". Below we address the editorial comments in blue font.

During these revisions we discovered that the plasmid used to derive the kinetic BRET results for β -arrestin2 recruitment shown in Figures 2c, 2e, and 2f included an unintended DNA sequence corresponding to the human nociception receptor (hNOP). This is unlikely to have influenced the previous results given that none of the compounds bind to NOP (this is known for DAMGO, and we show lack of FNZ and DFNZ binding to NOP in Fig. 1a and Extended Data Fig. 1b). Nevertheless, to be certain, we obtained a new plasmid, confirmed via DNA sequencing that it contained only RLuc8-fused hMOR, and repeated the experiments. The new results fully replicated the original findings. Below we show the previous data on the left and the new (current) data on the right. We include the new data in revised Fig. 2c, 2e, 2f and Extended Data Fig. 4b (raw net-BRET values).

Sincerely,

Michael Michaelides, Georgios Skiniotis, and Kenner Rice (on behalf of all coauthors)